# TBK1-Zyxin signaling controls tumor-associated macrophage recruitment to mitigate antitumor immunity

Ruyuan Zhou[1,2,3,10], Mengqiu Wang[1,10], Xiao Li[1,10], Yutong Liu[1,2,3], Yihan Yao[4], Ailian Wang[1,2,3], Chen Chen[1,2,3], Qian Zhang[1], Qirou Wu [1,2,3], Qi Zhang[2], Dante Neculai [5], Bing Xia[6], Jian-Zhong Shao[7], Xin-Hua Feng [1,8], Tingbo Liang[2,8], Jian Zou[9], Xiaojian Wang [4] & Pinglong Xu [1,2,3,8✉]

## Abstract

**Mechanical control is fundamental for cellular localization within a tissue, including for tumor-associated macrophages (TAMs). While the innate immune sensing pathways cGAS-STING and RLR-MAVS impact the pathogenesis and therapeutics of malignant diseases, their effects on cell residency and motility remain incompletely understood. Here, we uncovered that TBK1 kinase, activated by cGAS-STING or RLR-MAVS signaling in macrophages, directly phosphorylates and mobilizes Zyxin, a key regulator of actin dynamics. Under pathological conditions and in STING or MAVS signalosomes, TBK1-mediated Zyxin phosphorylation at S143 facilitates rapid recruitment of phospho-Zyxin to focal adhesions, leading to subsequent F-actin reorganization and reduced macrophage migration. Intratumoral STING-TBK1-Zyxin signaling was evident in TAMs and critical in antitumor immunity. Furthermore, myeloid-specific or global disruption of this signaling decreased the population of CD11b⁺ F4/80⁺ TAMs and promoted PD-1-mediated antitumor immunotherapy. Thus, our findings identify a new biological function of innate immune sensing pathways by regulating macrophage tissue localization, thus providing insights into context-dependent mitigation of antitumor immunity.**

**Keywords** cGAS-STING; Tumor-associated Macrophages; TBK1; Cell Motility; Antitumor Immunity
**Subject Categories** Cancer; Cell Adhesion, Polarity & Cytoskeleton; Immunology

## Introduction

Pattern recognition receptors (PRRs) monitor pathogen-associated molecular patterns (PAMPs) and damage-associated molecular patterns (DAMPs) to surveil tissue abnormalities and pathogen invasion. When double-stranded DNA (dsDNA) is detected in the cytosol, cyclic GMP-AMP synthase (cGAS) synthesizes 2'3'-cyclic GMP-AMP (cGAMP), activating STING located in the endoplasmic reticulum (ER) (Gao et al, 2013; Sun et al, 2013; Wu et al, 2013). The cGAS-STING pathway controls multiple cellular functions, including mRNA translation (Zhang et al, 2022a), autophagy (Gui et al, 2019; Liu et al, 2019), phase separation (Meng et al, 2021; Yu et al, 2021), and the activation of transcription factors interferon regulatory factor 3 (IRF3) and nuclear factor kappa B (NF-κB), which transcribe interferons and pro-inflammatory cytokines to initiate innate immune responses (Liu et al, 2015; Roers et al, 2016; Zhang et al, 2019). Similarly, RIG-I-like receptor (RLR)-MAVS signaling activates TBK1/IKKε kinases in mitochondria-based MAVS signalosomes to phosphorylate IRF3, thus initiating innate immune responses against RNA viruses (Rehwinkel and Gack, 2020). Dysfunction of cGAS-STING signaling or RLR-MAVS signaling contributes to some severe human diseases, including infectious diseases (Chen et al, 2016b; Di Domizio et al, 2022; Roers et al, 2016), autoimmune disorders (Crampton and Bolland, 2013; Gao et al, 2015; Liu et al, 2014; Sisirak et al, 2016), neurodegenerative diseases (Roy et al, 2020; Sliter et al, 2018; Wang et al, 2024; Yu et al, 2020), cardiovascular diseases (King et al, 2017; Luo et al, 2020), organ fibrosis (Chung et al, 2019; Wu et al, 2024; Zhang et al, 2022a), and cancers (Demaria et al, 2015; Deng et al, 2014; Fu et al, 2015; Kwon and Bakhoum, 2020). Therapeutic agents targeting cGAS-STING signaling, alone or in combination therapies (Zhang et al, 2022b), have proven valuable in murine models in treating cancers (Demaria et al, 2015; Fu et al, 2015) and inflammatory diseases

[1]MOE Laboratory of Biosystems Homeostasis and Protection, Zhejiang Provincial Key Laboratory for Cancer Molecular Cell Biology, Life Sciences Institute, Zhejiang University, Hangzhou 310058, China. [2]Institute of Intelligent Medicine, ZJU-Hangzhou Global Scientific and Technological Innovation Center, Hangzhou 310058, China. [3]Department of Hepatobiliary and Pancreatic Surgery and Zhejiang Provincial Key Laboratory of Pancreatic Disease, The First Affiliated Hospital, University School of Medicine, Zhejiang University, Hangzhou 310058, China. [4]Institute of Immunology, Zhejiang University School of Medicine, Hangzhou 310058, China. [5]Department of Cell Biology, Zhejiang University School of Medicine, Hangzhou 310058, P. R. China. [6]Department of Thoracic Cancer, Affiliated Hangzhou Cancer Hospital, Westlake University, Hangzhou 310030, China. [7]College of Life Sciences, Key Laboratory for Cell and Gene Engineering of Zhejiang Province, Zhejiang University, Hangzhou 310058, China. [8]Cancer Center, Zhejiang University, Hangzhou 310058, China. [9]Eye Center of the Second Affiliated Hospital School of Medicine, Institute of Translational Medicine, Zhejiang University, Hangzhou 310058, China. [10]These authors contributed equally: Ruyuan Zhou, Mengqiu Wang, Xiao Li. ✉E-mail: xupl@zju.edu.cn

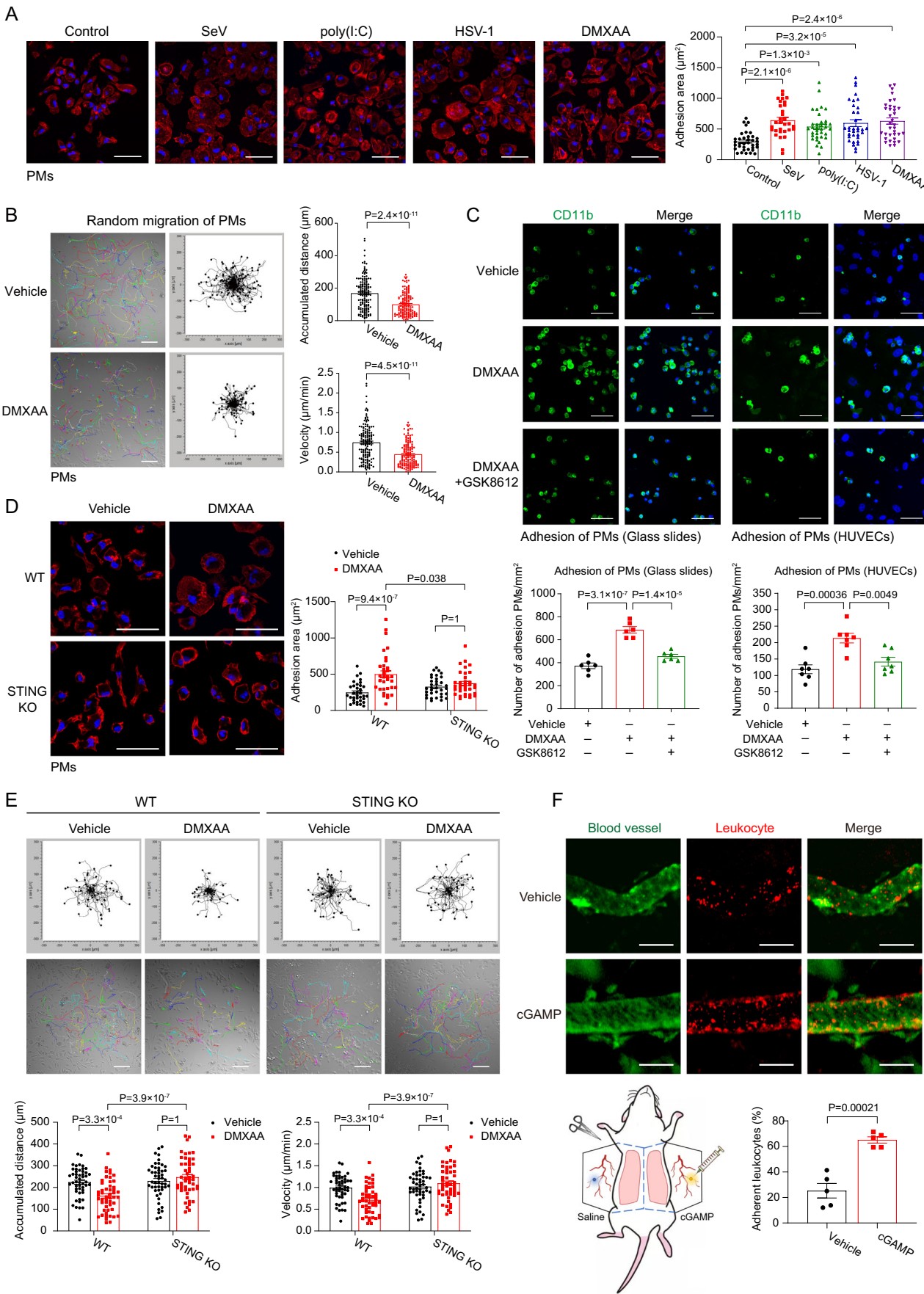

Figure 1. Nucleic acid sensing regulates the adhesion and motility of macrophages.

(A) Immunofluorescence imaging and statistics revealed enlarged adhesion areas of primary murine peritoneal macrophages (PMs) upon treatments of SeV (RNA virus, 9 h), poly(I:C) (RNA analogs, 3 h), HSV-1 (DNA virus, 9 h), or DMXAA (murine STING agonist, 1 h), which activated RLR-MAVS-mediated RNA sensing or cGAS-STING-mediated DNA sensing, respectively. DAPI (Blue) and TRITC-conjugated phalloidin (Red) labeled the nucleus and F-actin. Scale bar, 50 μm. Control group, $n = 36$; SeV group, $n = 31$; poly(I:C) group, $n = 35$; HSV-1 group, $n = 34$; DMXAA group, $n = 35$. (B) Representative migration plots indicated the movement of PMs upon the treatment of vehicle or DMXAA (10 μg/mL) for 4 h, monitoring by live-cell microscopy (left panel). Statistics of the accumulated distance of cell movement revealed (right panel). Scale bar, 100 μm. $n = 150$ cells per group. (C) PMs pretreated for 2 h with vehicle or DMXAA (10 μg/mL) in the absence or presence of TBK1 inhibitor GSK8612 (10 μM) were incubated on glass slides (left panel) or HUVECs (endothelial cell line, right panel) for 0.5 h. After washing off non-adherent cells, the remaining macrophages were labeled by anti-CD11b (green). Statistics of PMs attached to the glass slides ($n = 6$ per group) or HUVECs ($n = 7$ per group) were displayed. Scale bar, 50 μm. (D) Immunofluorescence imaging and statistics of cell adhesion area were shown in PMs from $Sting1^{+/+}$ or $Sting1^{-/-}$ (STING KO) mice and upon treatment with DMXAA (10 μg/mL) for 2 h. Scale bar, 50 μm. $n = 34$ cells per group. (E) Genetic ablation of STING enhanced macrophage motility, as evidenced by representative migration plots and statistics under live-cell microscopy Scale bar, 100 μm. $n = 50$ cells per group. (F) The murine vessels were visualized with FITC-Dextran, and the leukocytes (rhodamine-6G$^+$) that adhered to the blood vessels were imaged and quantitated in mice injected with saline or cGAMP (5 μg) for 2 h. Scale bar, 100 μm. $n = 5$ vessels in the injection site for imaged per group. Data information: unless otherwise indicated, $n = 3$ independent biological experiments (mean ± SEM). *$P < 0.05$, **$P < 0.01$, and ***$P < 0.001$, compared with the control condition (One-way ANOVA test and Bonferroni correction). Source data are available online for this figure.

(Decout et al, 2021; Haag et al, 2018). Characterizing cellular functions is vital for a more comprehensive understanding of these fundamental signaling pathways (Chen and Xu, 2023; Zhang et al, 2022b). However, our understanding of how these innate immune pathways influence adhesion and motility—key characteristics of individual cell sets—is notably limited.

Cell motility is a complex process involving functional and structural interactions between cells and their surrounding microenvironment. Macrophages, known for their inherent motility, reside in all types of tissues, exhibit a range of morphologies, and execute various functions from immune surveillance to tissue development (Davies et al, 2013; Jung, 2014). TAMs promote tumor progression by stimulating tumor cell survival and motility, angiogenesis, and suppressing attacks by natural killer cells and cytotoxic T cells, representing significant obstacles to tumor therapeutics in suppressive tumor microenvironments (Gao et al, 2021). Therefore, targeting the adhesion and movement processes of TAMs in the tumor microenvironment (TME) could potentially create new avenues for antitumor immunotherapy. However, our current understanding of innate immune control of macrophage adhesion is scarce.

Macrophages generally maintain organism homeostasis and have a more detrimental role in antitumor immunity (Colegio et al, 2014; Pan et al, 2020). While tissue-resident and circulating macrophages are crucial in organismal injury repair and cancer biology, the correlation between macrophage adhesion and function has been barely examined (Davies et al, 2013; Wynn et al, 2013; Wynn and Vannella, 2016). Investigating how the motility and adhesion of TAMs are signalingly regulated will broaden our understanding of the immune landscape. In this study, we report for the first time how the biological cGAS-STING signaling and RLR-MAVS signaling are transformed into a mechanical role that limits the motility of macrophages via an unrecognized TBK1-Zyxin axis. By employing genetic and immunological assays of STING-, TBK1-, Zyxin-, and IRF3-knockout cells and murine models, we found that this TBK1-Zyxin mechanism is robustly active in TAMs within tumors. Genetically and line-specifically disrupting this signaling enhanced antitumor immunity in murine models. Consequently, we have identified a prominent function of cGAS-STING and RLR-MAVS signaling in controlling the adhesion and motility of macrophages and its suppressive nature in antitumor immunity.

## Results

### Nucleic acid sensing regulates the adhesion and motility of macrophages

Remodeling the actin cytoskeleton and focal adhesions is necessary for macrophage migration (Friedl and Weigelin, 2008). To examine the effect of innate immune signaling on macrophage spreading and adhesion, we activated nucleic acid sensing signaling, including innate RNA sensing by RLR-MAVS signaling and innate DNA sensing by cGAS-STING signaling, by employing SeV (RNA virus), poly(I:C) (dsRNA analogs), HSV-1 (DNA virus), or DMXAA and diABZI (STING agonists) in primary murine peritoneal macrophages (PMs) and activated human monocyte THP-1. Distinct time frames were chosen according to the differential dynamics of cellular responses to these innate immune stimulations. Phalloidin staining revealed that sensing of both RNA and DNA noticeably increased the adhesion area of macrophages (Figs. 1A and EV1A). Next, we performed time-lapse imaging to observe the random migration of PMs on glass slides and measure their direct impacts on macrophage motility. Migration paths of individual cells revealed a substantially reduced velocity and cumulative movement distance in PMs upon cGAS-STING or RLR-MAVS activation (Figs. 1B and EV1B). The adhesion of PMs and THP-1 cells on the glass slides or human umbilical vein endothelial cells (HUVECs) was enhanced upon DMXAA or diABZI treatment but diminished by TBK1 inhibition (Figs. 1C and EV1C), while PMs from STING KO mice showed an unaltered adhesion area and reverted motility defects upon DMXAA treatment (Fig. 1D,E).

To validate macrophage movement control in vivo, we performed an assay to measure leukocyte adhesion to the blood vessels. For this purpose, circulating leukocytes were labeled by rhodamine-6G injected in the retro-orbital venous plexus, and cGAS-STING signaling is locally activated by administering cGAMP subcutaneously in the murine abdomen. Intravital imaging analyses of mice abdomen subcutaneous vessels revealed that STING activation promoted the adhesion of immune cells to blood vessels (Fig. 1F). These observations in vitro and in vivo suggest an intriguing role of cGAS-STING or RLR-MAVS signaling in controlling the adhesion and motion of macrophage and leukocytes.

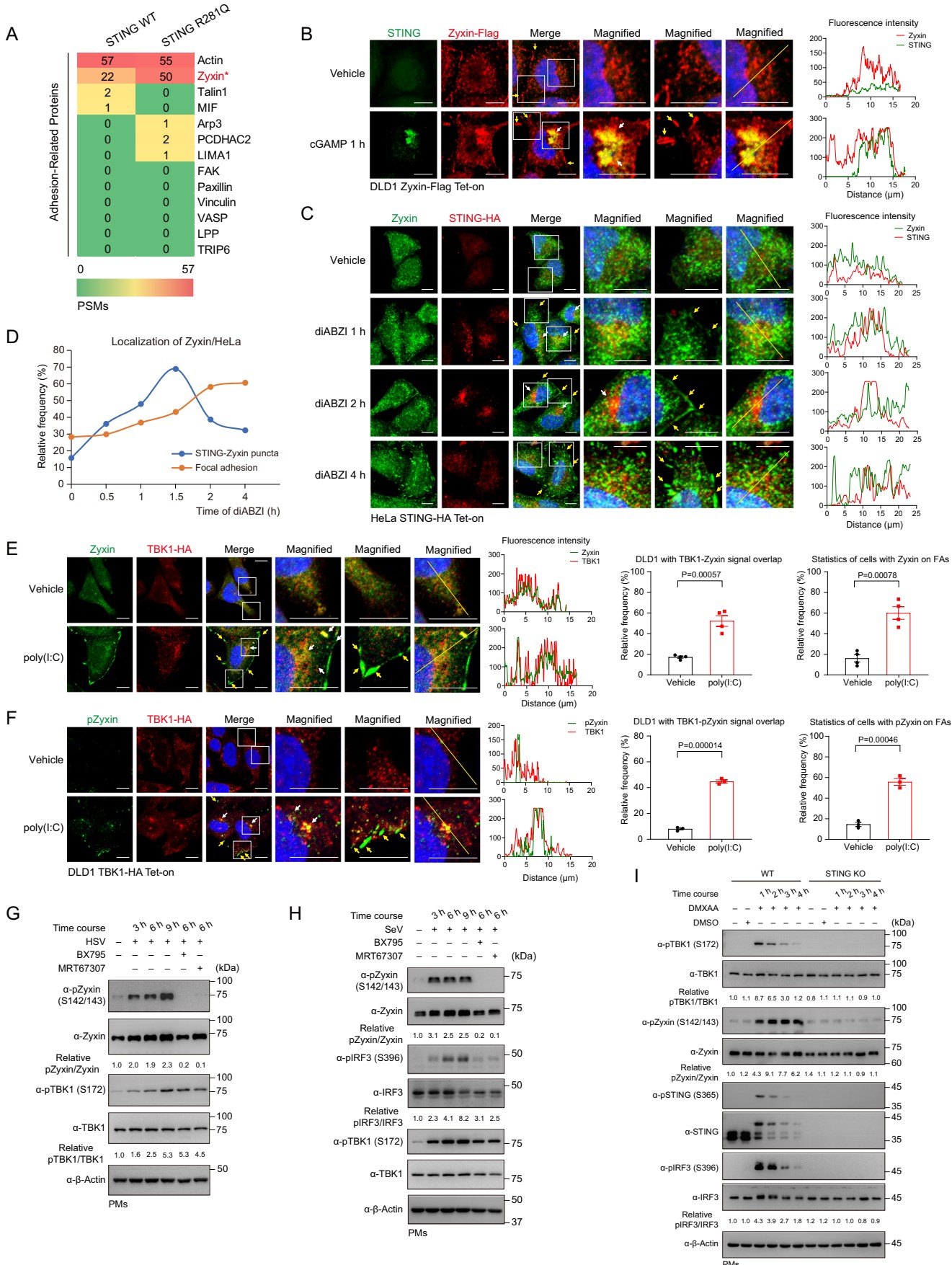

**Figure 2. STING or MAVS signalosomes recruit and phosphorylate Zyxin.**

(A) Heatmap represented the mass spectrometry analyses of enriched adhesion-related proteins associating with stably expressed STING, wild-type, or constitutively activated (caSTING, R281Q). Zyxin was an interacting protein of STING in mass spectrometry assays, with a higher affinity to caSTING. (B) Immunofluorescence imaging detected an overlap of the cellular distribution of stably expressed Zyxin in STING signalosomes formed upon cGAMP treatment. cGAMP drove a colocalization of Zyxin into STING aggregates (white arrowed) and focal adhesions (yellow arrowed). Scale bar, 10 μm. (C, D) STING agonist diABZI induced an association of endogenous Zyxin with STING signalosomes (white arrowed) and its sequential distribution into focal adhesions (yellow arrowed) in HeLa cells (C). Statistics for the percentage of total cells of three experiments with Zyxin in STING signalosomes or focal adhesions were shown (D). Scale bar, 10 μm. (E) Upon poly(I:C) stimulation (3 h), immunofluorescence and statistics revealed a signal overlap of endogenous Zyxin with stably expressed TBK1 (white arrowed) or on focal adhesions (yellow arrowed). $n = 4$ per group. (F) Immunofluorescence and statistics showed pZyxin, revealed by a phospho-Zyxin (S142/S143) antibody, was exclusively colocalized with TBK1 in the puncta (white arrowed) or on focal adhesions (yellow arrowed). Scale bar, 10 μm. $n = 3$ per group. (G, H) Nucleic acid sensing in PMs, induced by the infections of HSV-1 (G, DNA sensing) or SeV (H, RNA sensing), triggered robust Zyxin phosphorylation at S142/S143 residues. (I) Genetic ablation of STING in primary macrophages abrogated DMXAA-induced phosphorylation of TBK1, STING, IRF3, and Zyxin. Data information: unless otherwise indicated, $n = 3$ independent biological experiments (mean ± SEM). *$P < 0.05$, **$P < 0.01$, and ***$P < 0.001$, compared with the control condition (One-way ANOVA test and Bonferroni correction). Source data are available online for this figure.

## STING or MAVS signalosomes recruit and phosphorylate Zyxin

To decipher the underlying molecular basis of STING-regulated adhesion, we stably expressed wild-type and constitutively activated STING (R281Q) in human colorectal cancer cell line DLD1 cells and performed a high-throughput mass spectrometry analysis for STING interactomes. Zyxin, a zinc-binding protein and a critical regulator of F-actin polymerization (Smith et al, 2010; Smith et al, 2014), was revealed to interact with STING substantially among a variety of adhesion-related proteins, particularly with active STING (Fig. 2A). Next, we generated DLD1 cells stably expressing Flag-tagged Zyxin to directly visualize the complex of Zyxin and STING by immunofluorescence and overcome the species cross-reaction of commercially available primary antibodies targeting STING and Zyxin. Upon activation by its natural agonist 2'3'-cGAMP, STING translocated to the ERGIC/Golgi apparatus, where it aggregated into puncta (Figs. 2B and EV2A). Upon STING activation, we found some signal overlaps between Zyxin and STING puncta (indicated by white arrows), suggesting that Zyxin might be a component of STING signalosomes. Intriguingly, activation of cGAS-STING signaling promoted a substantial distribution of Zyxin on focal adhesions (indicated by yellow arrows) (Fig. 2B). Consistently, STING agonist diABZI induced the association of endogenous Zyxin proteins with STING signalosome and its sequential distribution into focal adhesions in HeLa cells and PMs (Figs. 2C,D and EV2B). A significant proportion of Zyxin was translocated into adhesions alongside the termination of STING signalosomes (Figs. 2C,D and EV2B). These observations support Zyxin as a component of the STING signaling complex.

TBK1 is a core component of both MAVS and STING signalosomes, where it phosphorylates both STING and IRF3 to assemble, activate, and execute the function of these signalosomes (Liu et al, 2015; Zhang et al, 2019). We found that activating either RNA sensing by poly(I:C) or DNA sensing by diABZI drove the pericellular focal adhesion localization of endogenous Zyxin proteins in DLD1 cells (yellow arrows) (Figs. 2E and EV2C) and observed the signal overlap between Zyxin and TBK1 in the perinuclear area (white arrows) (Figs. 2E and EV2C). These observations reveal that Zyxin, a component of STING or MAVS signalosomes, is driven to focal adhesion localization by cGAS-STING or RLR-MAVS signaling. We then monitored these effects on the phosphorylation of endogenous Zyxin in DLD1 cells upon TBK1 activation with a commercially available antibody against phospho-Zyxin (S142/143). To our surprise, we observed a robust pZyxin signal when nucleic acid sensing was activated, which colocalized with TBK1 (white arrows) or attached to focal adhesions (yellow arrows) (Fig. 2F). These unexpected visualizations suggest the previously unrecognized signaling axes of STING-TBK1-Zyxin and MAVS-TBK1-Zyxin.

We found substantial signals of HSV-1-, SeV-, or poly(I:C)-induced formation of endogenous pZyxin at S142/S143 residues in primary macrophages (Figs. 2G,H and EV2D). Inhibiting TBK1 by small molecular inhibitors BX795 or MRT67307, which abrogated TBK1 activity but not its S172 phosphorylation, eliminated Zyxin phosphorylation in resembling their elimination for IRF3 phosphorylation, indicating that both Zyxin and IRF3 as downstream molecules of TBK1 activation, directly or indirectly (Figs. 2G,H and EV2D). PMs obtained from global STING knockout mice lost DMXAA-induced phosphorylation of Zyxin, in agreement with their losses of pTBK1, pSTING, and pIRF3 (Fig. 2I), suggesting that Zyxin phosphorylation is an event within and of STING signalosomes in macrophages. In addition, PMs from global IRF3 knockout mice retained the signal of DMXAA-induced pZyxin, indicating that the TBK1-Zyxin axis is independent of the classical TBK1-IRF3 axis (Fig. EV2E). Besides, we found that interferons mIFNα and mIFNβ induced a moderate increase in PM adhesion area while reversible by an anti-INFAR1 neutralizing antibody (Fig. EV2F). These interferons, as well as the activation of cGAS-STING signaling, induced an increase in Zyxin mRNA and protein levels (Fig. EV2G–I), indicating an inherent connection between TBK1-IRF3-IFN signaling and TBK1-Zyxin signaling. These collective observations from primary macrophages suggest the presence of robust TBK1-Zyxin signaling in response to nucleic acid sensing.

## TBK1 directly phosphorylates S143 to facilitate Zyxin focal adhesion localization

Next, we attempted to validate TBK1-direct or indirect phosphorylation on Zyxin. An endogenous complex of TBK1-Zyxin was detected by coimmunoprecipitation assays in primary murine macrophages, facilitated upon nucleic acid sensing (Fig. 3A). Further investigation suggested that Zyxin interacted moderately with both active and inactive TBK1 (Fig. 3B). A mobility shift of

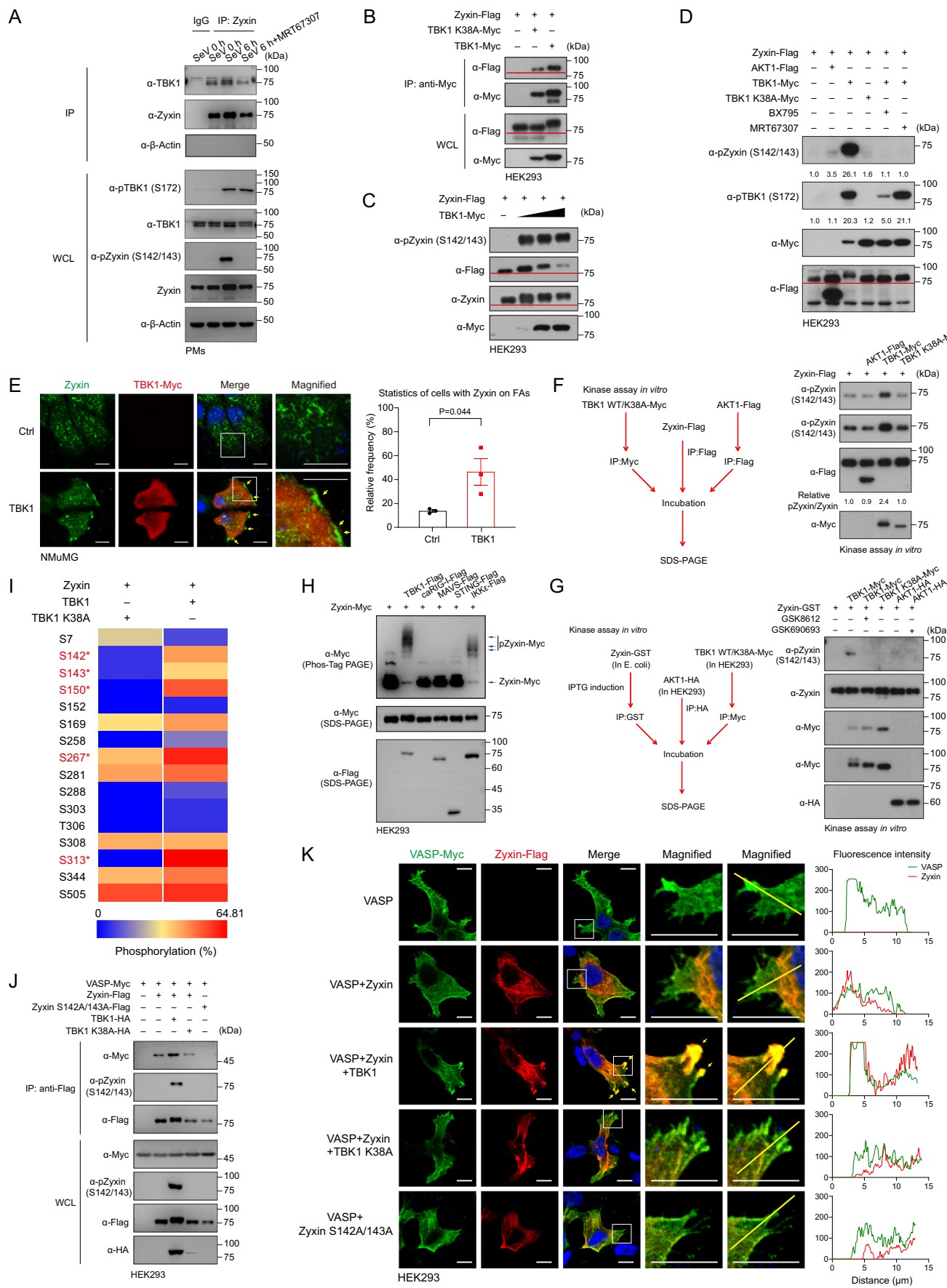

Figure 3. TBK1 directly phosphorylates S143 residue to facilitate Zyxin focal adhesion localization.

(A) Coimmunoprecipitation assay revealed an endogenous complex comprising Zyxin and TBK1 in PMs. Infection of SeV (6 h) enhanced the formation of this complex, while inhibition of TBK1 by MRT67307 (10 μM) attenuated it. (B) Coimmunoprecipitation assay revealed the association of Zyxin and TBK1, either in wild-type or kinase-dead (K38A) form. (C) Ectopic expression of TBK1 in HEK293 cells phosphorylated Zyxin proteins at residues S142/S143 in a dose-dependent manner. (D) Robust signals of pZyxin (S142/S143) were detected in HEK293 cells in the presence of TBK1, compared to AKT1, a kinase reported for phosphorylating Zyxin at S142. pZyxin was abrogated in the presence of specific TBK1 inhibitors BX795 (6 μM) or MRT67307 (10 μM). (E) Immunofluorescence imaging and statistics revealed a localization of Zyxin on focal adhesion driven by TBK1 (yellow arrowed). Scale bar, 10 μm. $n = 3$ per group. (F, G) An in vitro kinase assay was performed by separately expressed and purified Zyxin from HEK293 cells (F) or bacteria E. coli (G) and purified TBK1 and AKT1, revealing that TBK1 directly phosphorylated S142/S143 residues of Zyxin. TBK1 inhibitor GSK8612, but not AKT1 inhibitor GSK690693, blocked this Zyxin phosphorylation. (G) Phos-Tag electrophoresis showed robust phosphorylation of Zyxin when coexpressed with TBK1 or IKKε. (I) Mass spectrometry analyses showed that the phosphorylation of multiple residues on Zyxin proteins was upregulated by active TBK1 (labeled as *), including S142, S143, S150, S267, and S313. (J) Coimmunoprecipitation assay showed that TBK1 enhanced the association of Zyxin with VASP, a core component of focal adhesion. Zyxin S142A/S143A mutation failed to interact with VASP. (K) The subcellular localization of WT Zyxin and the S142A/143A mutant with VASP was visualized by immunofluorescence, promoted on focal adhesion (yellow arrowed) by active TBK1, but attenuated by the S142A/143A mutant. Scale bar, 10 μm. Data information: unless otherwise indicated, $n = 3$ independent biological experiments (mean ± SEM). *$P < 0.05$, **$P < 0.01$, and ***$P < 0.001$, compared with the control condition (One-way ANOVA test and Bonferroni correction). Source data are available online for this figure.

Zyxin (indicated by red lines) and phosphorylation on the S142/S143 residues were detected when coexpressed with exogenous TBK1 (Fig. 3B,C), suggesting significant phosphorylation on Zyxin driven by TBK1. Previously, AKT1 was reported to phosphorylate Zyxin at S142 residue to regulate its nuclear translocation (Chan et al, 2007), and we thus included AKT1 in a comparison. TBK1, rather than AKT1, induced an evident Zyxin signal, which was eliminated by TBK1 inhibitors (Fig. 3D). In addition, immunofluorescence imaging indicated that ectopic expression of TBK1, which was active due to autophosphorylation, directly drove the relocalization of Zyxin on the focal adhesions (yellow arrows) (Fig. 3E). To confirm whether TBK1 directly phosphorylates Zyxin at S142/S143 residues, we employed an in vitro kinase assay with AKT1 as a control. TBK1, depending on its enzymatic activity, phosphorylated Zyxin at S142/S143 in vitro, either on Zyxin proteins expressed and purified from HEK293 cells (Fig. 3F) or E. coli (Fig. 3G). The phosphorylation modification of Zyxin proteins by TBK1 was also evident in Phos-Tag PAGE (Fig. 3H). A high-throughput mass spectrometry analysis of Zyxin was performed in the presence of wild-type or kinase-dead TBK1, which revealed that multiple Zyxin residues, including S142, S143, and S150 residues on the same α-helix and S267 and S313 residues with flexible structure (Fig. EV3A, predicted by Alphafold), were substantially phosphorylated by TBK1 (Fig. 3I). These data collectively suggest TBK1-mediated direct phosphorylation of Zyxin in the TBK1-Zyxin complexes.

To precisely determine the biological effect of TBK1-induced phosphorylation on Zyxin at individual residue, we generated Zyxin KO HEK293 cells by CRISPR and reconstituted them with the phosphomimetic Zyxin (serine-to-aspartate point mutants) and analyzed by immunofluorescence. Zyxin mutants of S143D and S142D/S143D displayed an apparent focal adhesion distribution (Fig. EV3B). S143 is on an α-helix that Alphafold predicts interacting with the C-terminal region of Zyxin to form a closed structure (Fig. EV3A), suggesting that TBK1-mediated phosphorylation on S142, S143, and S150 may interfere with the head-tail closed structure of Zyxin (Call et al, 2011; Moody et al, 2009). We also found that the LIM domain of Zyxin resided in the nucleus, while the Zyxin truncation lacking LIM domains localized to focal adhesions (Fig. EV3C), supporting that TBK1-mediated phosphorylation influences the head-tail closed structure of Zyxin. Furthermore, the domain mapping analyses utilizing truncated TBK1 and Zyxin proteins indicated that the

N-terminal kinase domain and ULD (1–382) of TBK1 and the C-terminal LIM of Zyxin were responsible for their interaction (Fig. EV3D,E). In addition, TBK1 promoted the interaction between Zyxin and VASP, a core component of focal adhesions (Grange et al, 2013; Hoffman et al, 2006); by contrast, preventing TBK1-mediated Zyxin phosphorylation by S142A/143A mutations disrupted Zyixn recruitment on focal adhesions (Figs. 3J,K and EV3F). Signal overlap of endogenous Zyxin and VASP increased on focal adhesions (yellow arrowed) upon activating TBK1 with diABZI or poly(I:C) in DLD1 and decreased when STING or TBK1 were inhibited by H151 or GSK8612 (Fig. EV3G,H). These data suggest a TBK1-mediated direct phosphorylation of Zyxin on S143, which promotes Zyxin recruitment on focal adhesions, where Zyxin polymerizes F-actin (Smith et al, 2010; Smith et al, 2014).

## The TBK1-Zyxin cascade restricts macrophage motility

Zyxin is a critical regulator of cell migration, adhesion, and mechanical force sensitivity (Smith et al, 2010; Yoshigi et al, 2005), although previous reports mainly focus on its role in epithelial and endothelial cells. We found that cGAS-STING and RLR-MAVS signaling regulated the adhesion and motility of macrophages (Fig. 1). We analyzed total and phospho-specific endogenous Zyxin proteins in PMs by immunofluorescence. The evident spread of PMs on the glass slides was observed when nucleic acid sensing was activated, as well as the relocalization of a substantial proportion of Zyxin proteins from cellular distribution to adhesion plaques (yellow arrows) (Figs. 4A–D and EV4A,B). Phosphorylation of endogenous Zyxin at the S142/S143 residues was robustly triggered by cGAS-STING signaling or RLR-MAVS signaling and exclusively distributed into adhesion plaques (yellow arrows) (Fig. 4A,B,E,F), demonstrating robust TBK1-Zyxin signaling in macrophages. Either pharmacological inhibition of TBK1 (Figs. 4A,B and EV4A) or genetic deletion of STING (Figs. 4C–F and EV4B) abrogated the phosphorylation and adhesion redistribution of endogenous Zyxin and macrophage spreading morphology. These observations suggest TBK1-Zyxin signaling in macrophages downstream of RNA/DNA sensing to facilitate macrophage adhesion and spread.

We also found that Zyxin deletion in macrophages reduced their adhesion areas on the glass slides in response to DNA or RNA sensing (Fig. EV4C,D). Besides, Zyxin deletion restored random

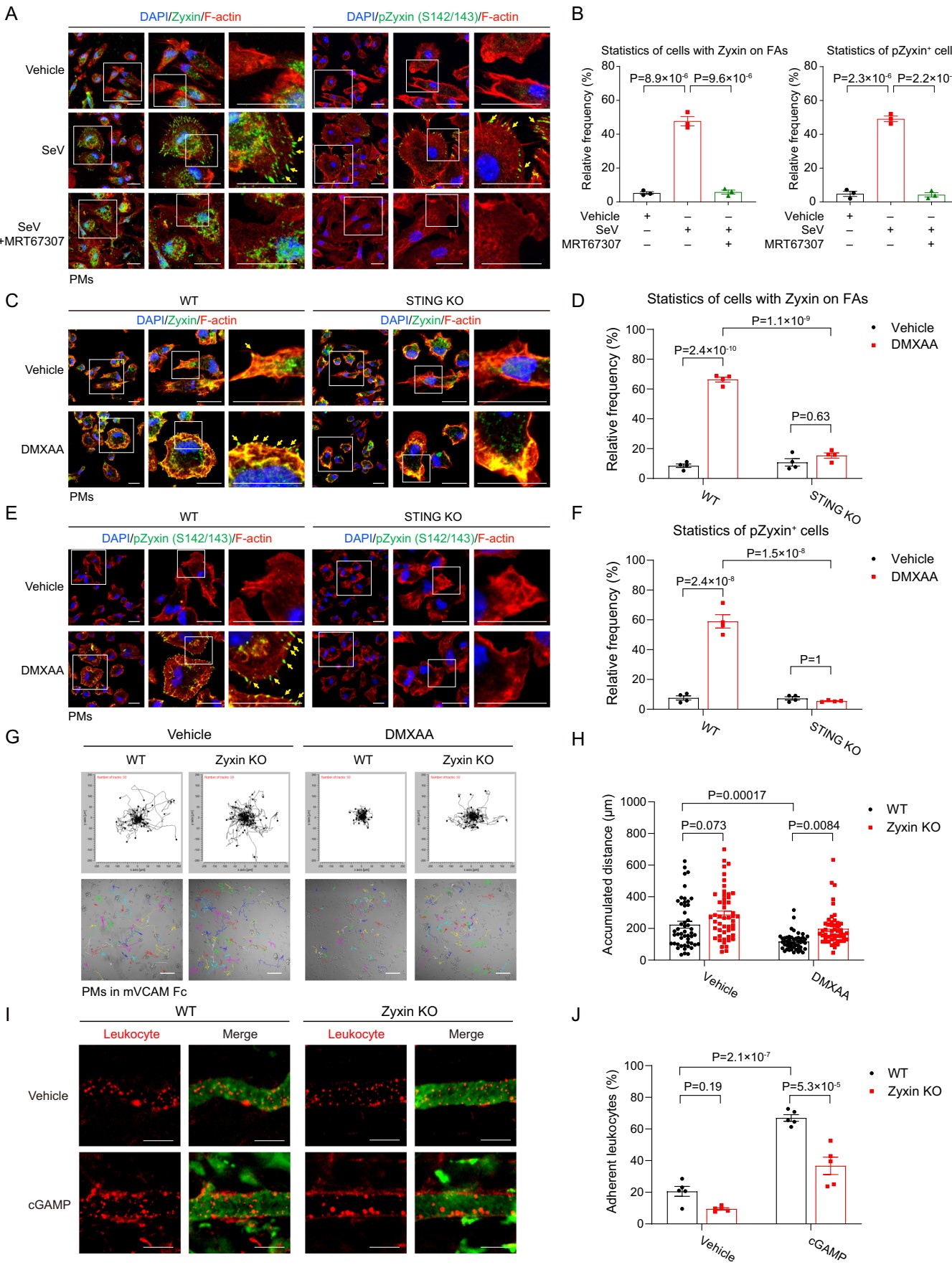

**Figure 4. The TBK1-Zyxin cascade restricts macrophage motility.**

(A, B) Immunofluorescence and statistics indicated that SeV infection triggered a focal adhesion location of endogenous Zyxin proteins in macrophages (indicated by the yellow arrows). The phospho-Zyxin (S142/S143) signal, induced by SeV infection, was translocated exclusively into focal adhesions, a process blocked by TBK1 inhibitor MRT67307 (10 μM, 6 h). Scale bar, 20 μm. (B) $n = 3$ per group. (C–F) Immunofluorescence and statistics revealed that DMXAA induced Zyxin aggregations (C, D) and phosphorylation (E, F) on focal adhesions (yellow arrowed), a process entirely blocked in PMs from STING KO mice. Scale bar, 20 μm. (D, F) $n = 4$ per group. (G, H) Genetic ablation of Zyxin enhanced macrophage motility on the surface of mVCAM Fc-coated glass slides, as evidenced by representative migration plots and statistics under live-cell microscopy in the absence or presence of DMXAA (10 μg/mL, 6 h). Scale bar, 100 μm. (H) $n = 50$ cells. (I, J) $Zyxin^{+/+}$ or $Zyxin^{-/-}$ mice were injected with saline or cGAMP (5 μg) for 2 h, and their blood vessels were visualized with FITC-Dextran. Leukocytes (rhodamine-6G$^+$) that adhered to the blood vessels were imaged and quantitated. Scale bar, 100 μm. (J) $n = 5$ vessels in the injection site for imaged per group. Data information: unless otherwise indicated, $n = 3$ independent biological experiments (mean ± SEM). *$P < 0.05$, **$P < 0.01$, and ***$P < 0.001$, compared with the control condition (One-way ANOVA test and Bonferroni correction). Source data are available online for this figure.

macrophage migration, which was otherwise inhibited by STING/MAVS signaling (Figs. 4G,H and EV4E). Similarly, the distribution of Zyxin on focal adhesions (yellow arrowed) in response to RNA sensing was wholly attenuated in Zyxin-deleted cells (Fig. EV4F,G). Zyxin deletion also inhibited the enhanced adhesion by poly(I:C) of PMs on both glass slides and HUVECs (Fig. EV4H,I). In vivo, Zyxin deletion profoundly decreased the adhesion of immune cells to blood vessels, a process promoted by cGAS-STING signaling (Fig. 4I,J). These cellular and animal observations suggest that TBK1-Zyxin signaling is responsible for macrophage adhesion and motility regulation, at least partially.

## Zyxin deficiency enhances inflammatory responses

Macrophages are critical in initiating, maintaining, and resolving inflammation. We thus analyzed Zyxin KO mice to investigate the connection between Zyxin and inflammatory responses (Fig. EV5A). Zyxin KO mice were mainly normal. However, over half of the mice had visible skin lesions and inflammation at eight months of age, indicating the presence of hyperactive inflammatory or autoimmune responses (Fig. EV5B). Zyxin deficiency appeared to promote cGAMP- or diABZI-induced cGAS-STING signaling in PMs moderately (Fig. 5A,B), probably by stabilizing STING signalosomes, as indicated by an extended aggregation of TBK1 proteins (Fig. 5C). Furthermore, Zyxin deletion partially promoted poly(I:C)-induced RNA sensing and TLR4 signaling, as evidenced by somewhat elevated levels of pTBK1 and pIRF3 (Fig. EV5C,D). We found that Zyxin interfered with the TBK1-IRF3 interaction, suggesting the presence of a possible substrate competition between Zyxin and IRF3 (Figs. 5D and EV5E). Noticeably, Zyxin depletion caused more severe lung inflammation induced by cGAMP, disrupting the alveolar architecture and infiltrating granulocytes in hematoxylin and eosin (H&E) staining (Fig. 5E,F). These observations suggest an inflammation-inhibitory role of Zyxin in vivo, possibly through macrophage regulation.

## Intervening in STING-TBK1-Zyxin signaling improves antitumor immunity and synergizes with PD-1 immunotherapy

TAMs are frequently immunosuppressive and involved in the stages of tumorigenesis, cancer progression, invasion, and metastasis, and their infiltration is commonly associated with a poorer prognosis (Pittet et al, 2022; Quail and Joyce, 2013).

By contrast, cGAS-STING signaling has validated functions in boosting antitumor immunity by enhancing the secreting of interferons that recruit and mature various T cells and NK cells (Corrales et al, 2015; Li et al, 2020; Marcus et al, 2018). Can STING-TBK1-Zyxin signaling detain TAMs in tumor microenvironments and thus suppress antitumor immunity? To investigate this possibility, we employed syngeneic models of B16-F10 melanoma and MC38 colon adenocarcinoma in Zyxin KO mice (Fig. 6A). Activation of STING-TBK1-Zyxin signaling was evident in B16-F10 melanoma (Fig. EV6A), and phospho-Zyxin induced upon cGAMP administration were mainly in F4/80$^+$ macrophages and, to a lesser extent, in CD4$^+$ T cells (Fig. EV6B,C). Noticeably, cGAMP promoted the aggregation of F4/80$^+$ cells at the edge of B16-F10 tumors (Fig. EV6B), while Zyxin deletion inhibited tumor growth at a level comparable to cGAS-STING activation (Fig. 6B–D). A substantial reduction of CD11b$^+$F4/80$^+$ TAMs in B16-F10 tumors was observed in Zyxin KO mice by immunohistochemistry and flow cytometry analyses, alongside an enhanced proportion of CD4$^+$ T and CD8$^+$ cytotoxic T lymphocytes. Stimulation in Zyxin KO mice by cGAMP did not further enhance infiltrated T cells (Figs. 6E,F and EV6D). Zyxin knockout similarly compromised the growth of syngeneic MC38 colon adenocarcinoma (Fig. EV6E–G), where the CD11b$^+$ myeloid cells contained a high proportion of visible phospho-Zyxin labeling, while $Sting^{-/-}$ mice showed a decreased proportion of CD11b$^+$pZyxin$^+$ cells (Fig. EV6H,I). These observations suggest an intriguing role of STING-TBK1-Zyxin signaling in facilitating intratumoral macrophage residency that suppresses antitumor immunity. Besides, we observed that Zyxin deletion decreased the levels of CD206$^+$ M2-type macrophages but somewhat increased CD86$^+$ M1-type macrophages within tumors by unknown mechanisms (Fig. 6G,H), which probably contributed to improved antitumor immunity.

We additionally examined the effect of neutralizing anti-PD-1 antibodies in Zyxin KO mice, which otherwise had a marginal effect on melanoma because of unfavorable tumor microenvironments (Kleffel et al, 2015). Anti-PD-1 administration reduced the subcutaneous growth of melanoma-derived B16-F10 cells in Zyxin KO mice (Fig. 6I–K), with significantly more CD8$^+$ T lymphocyte infiltration (Fig. EV6J). The Zyxin KO+anti-PD-1 group exhibited significantly synergic and more potent antitumor effects than any single condition (Figs. 6I–K and EV6J), possibly as the anti-PD-1 strategy further improved activation and efficacy of these infiltrated T lymphocytes.

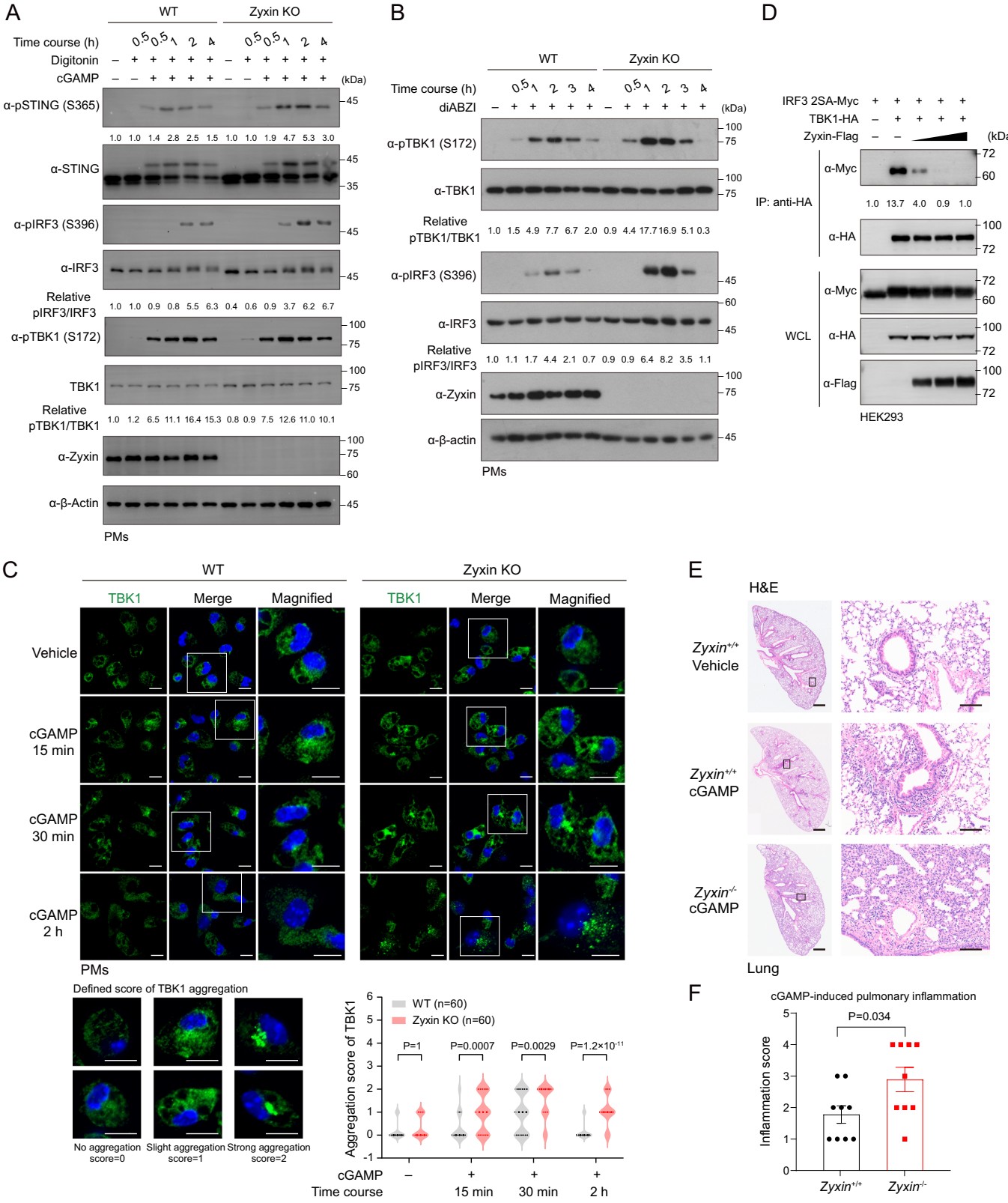

**Figure 5. Zyxin deficiency enhances inflammatory responses.**

(A–C) Zyxin deficiency promoted cGAMP or diABZI-induced cGAS-STING signaling in PMs moderately (A, B) and stabilized endogenous TBK1 aggregation (C). Scale bar, 10 μm. (C) WT Vehicle group, $n = 54$; WT cGAMP 15 min group, $n = 42$; WT cGAMP 30 min group, $n = 59$; WT cGAMP 2 h group, $n = 59$; Zyxin KO Vehicle group, $n = 40$; Zyxin KO cGAMP 15 min group, $n = 52$; Zyxin KO cGAMP 30 min group, $n = 60$; Zyxin KO cGAMP 2 h group, $n = 58$. (D) Coimmunoprecipitation assays to detect the TBK1-IRF3 interaction in the absence or presence of Zyxin, revealing a competitive inhibition of Zyxin on the association of TBK1-IRF3 2SA (a documented IRF3 mutant used to detect subtle TBK1-IRF3 interaction). (E, F) Representative images and statistics showed that Zyxin-deficiency mice with severe symptoms of cGAMP-induced pulmonary inflammation, including the disruption of the alveolar architecture and infiltration of granulocytes. (F) $n = 9$ per group. Data information: unless otherwise indicated, $n = 3$ independent biological experiments (mean ± SEM). *$P < 0.05$, **$P < 0.01$, and ***$P < 0.001$, compared with the control condition (One-way ANOVA test and Bonferroni correction). Source data are available online for this figure.

## STING-TBK1-Zyxin signaling retains TAM residency to suppress antitumor immunity

To precisely determine the route of STING-TBK1-Zyxin signaling in antitumor immunity, we first applied clodronate liposomes in WT and Zyxin KO mice, the reagent that specifically eliminated macrophage in tissues (Nguyen et al, 2021) (Fig. 7A). Clodronate liposome administration reduced the levels of TAMs in B16 tumors, alongside a decrease of phospho-Zyxin (Fig. EV7A–C). Though insignificant, clodronate liposomes somewhat improved tumor growth in Zyxin KO mice. Consistent with observations in global Zyxin KO, we observed a marked inhibition of tumor growth in Zyxin KO mice only in the control group rather than in groups with macrophage depletion (Fig. 7B,C). Next, we employed syngeneic B16-F10 models in immunodeficient NSG mice. Tumor growth curves and weights were comparable between WT and Zyxin KO B16-F10 tumors (Fig. 7D–F), further supporting that Zyxin depletion does not impact tumor growth but probably influences antitumor immunity. As anticipated, intratumoral cGAMP injection only modestly reduced B16-F10 tumor growth and weight in NSG mice, significantly less than observed in C57BL/6 mice (Figs. 6B–D and 7D–I). To genetically determine the presence of STING-TBK1-Zyxin signaling in macrophages, we generated $Cx3cr1^{cre};Sting1^{flox/flox}$ mice and induced macrophage-specific STING deletion with tamoxifen. As expected, pTBK1 and pZyxin signals were abrogated in F4/80$^+$ macrophages from these mice (Fig. EV7D–F). These findings suggest that STING-TBK1-Zyxin signaling is active in TAMs and is crucial in modulating antitumor immunity.

To investigate whether the effects of STING-TBK1-Zyxin signaling were relied on IRF3, we inoculated B16-F10 melanoma cells into WT, $Sting1^{-/-}$, $Irf3^{-/-}$, or $Zyxin^{-/-};Irf3^{-/-}$ mice. We found that cGAMP-induced antitumor immunity and increased pZyxin levels were STING-dependent (Figs. 7G–I and EV6A). Inhibition of TBK1, but not global IRF3 knockout, reduced tumor growth, F4/80$^+$ macrophage infiltration, and pZyxin$^+$ cells proportion (Fig. 7J–N). The decreased F4/80$^+$ macrophages and reduced tumor growth in $Zyxin^{-/-};Irf3^{-/-}$ mice (Fig. 7L–N) suggested that the enhanced antitumor immunity observed upon Zyxin depletion was at least partially independent of the TBK1-IRF3 axis. These pharmacological and genetic evidence lines suggest that STING-TBK1-Zyxin signaling is significantly present in TAMs and critical in antitumor immunity.

## Discussion

Innate nucleic sensing pathways such as cGAS-STING monitor pathogenic and aberrant DNAs to guard against pathogen infection and maintain tissue homeostasis. Besides producing a range of interferons and cytokines, the non-canonical roles of nucleic acid sensing signaling in autophagy (Gui et al, 2019; Liu et al, 2019), translation (Zhang et al, 2022a), organelle dynamics (Chen et al, 2020), phase separation (Meng et al, 2021), senescence (Dou et al, 2017; Gluck et al, 2017; Wu et al, 2024), and epithelial-to-mesenchymal transition (Xu et al, 2014), have been increasingly recognized. Dysregulation of nucleic acid sensing frequently leads to infectious diseases, inflammatory diseases, autoimmune diseases, neurodegenerative disorders, cardiovascular diseases, fibrosis, and cancers (Chen et al, 2016b; Crampton and Bolland, 2013; Demaria et al, 2015; Deng et al, 2014; Fu et al, 2015; Gao et al, 2015; Roers et al, 2016; Zhang et al, 2022a). Conversely, this study uncovers an intriguing role of nucleic acid sensing signaling in governing cell motility by directly converting biological signals into mechanical cues. Upon activation of nucleic acid sensing, we discovered that TBK1 phosphorylates of Zyxin, a crucial orchestrator of the actin cytoskeleton (Smith et al, 2010; Smith et al, 2014), which confers an innate immune-induced restraining of macrophage motility. This unanticipated STING-TBK1-Zyxin pathway is prominently activated in macrophages, and its intervention relieves the suppressive immune effects against tumors. Therefore, targeting TBK1-Zyxin signaling might offer mechanistic insights and therapeutic potential in enhancing antitumor immunity and suppressing excessive inflammatory responses.

## STING-TBK1-Zyxin signaling detains TAMs to modulate antitumor immunity

A STING-dependent adhesion and residency of TAMs in the tumor microenvironment was confirmed through genetic deletions of Zyxin, STING, and IRF3. The activation of cGAS-STING signaling significantly restrains macrophage movement both in vitro and in vivo, achieved through STING-induced phosphorylation and recruitment of Zyxin into the focal adhesions that regulate the cytoskeleton. Roles of cGAS-STING signaling in tumor biology are complex, particularly with integrating antitumor immunity from NK cells and CD8$^+$ T lymphocytes, TAM-mediated immune suppression, and effects of STING-induced senescence and microenvironmental regulation (Chen and Xu, 2023). cGAS-STING signaling induces robust expression of type I interferons and ISGs, which recruits and activates CD8$^+$-mediated antitumor immunity (Wu et al, 2019). This study observed that activating cGAS-STING signaling by cGAMP restrains intratumoral macrophage residency through STING-TBK1-Zyxin signaling. Interestingly, an M2-to-M1 macrophage polarization was also seen during the activation of cGAS-STING signaling and in the absence of

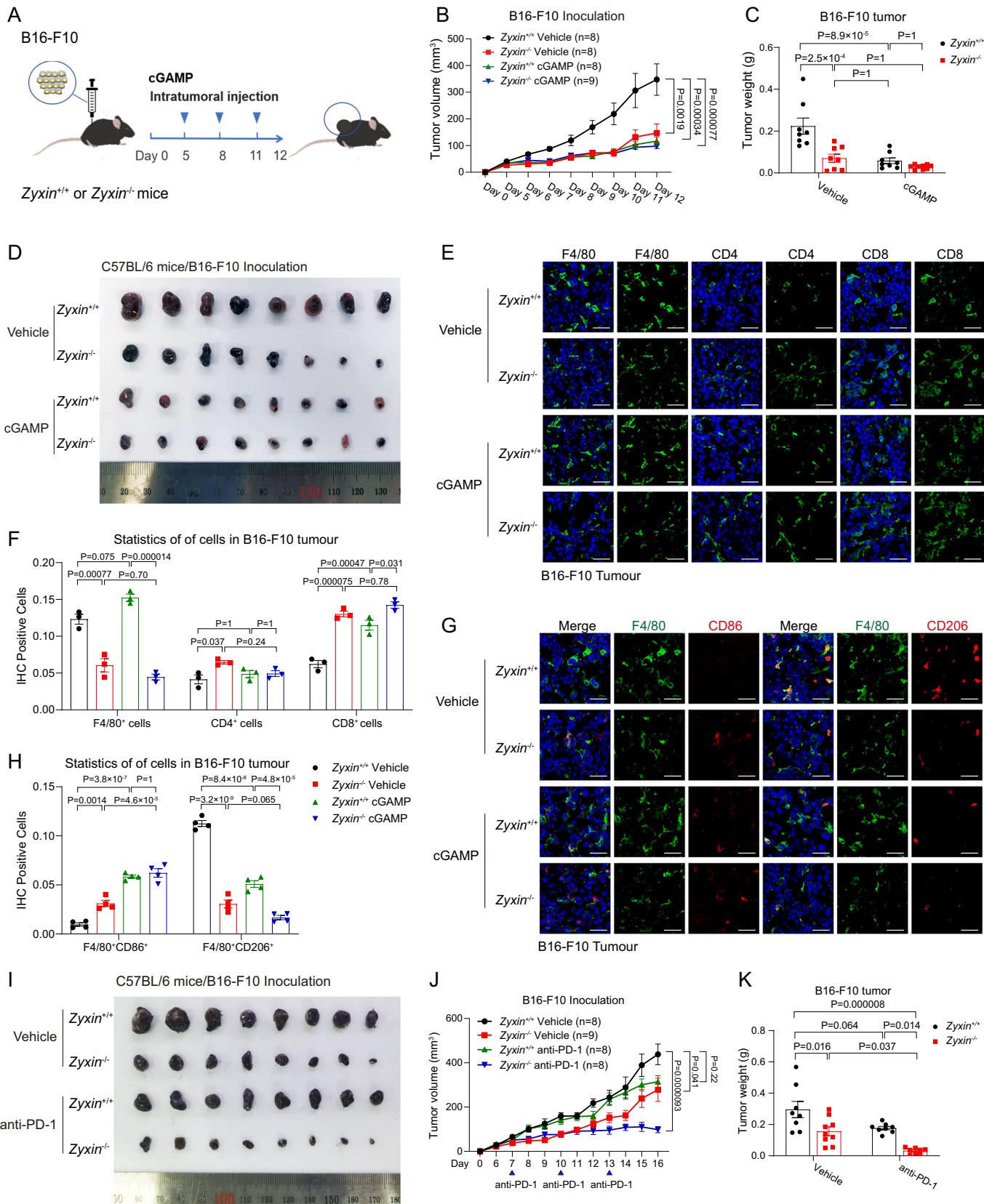

**Figure 6.  Intervening in STING-TBK1-Zyxin signaling improves antitumor immunity and synergizes with PD-1 immunotherapy.**

(A–D) B16-F10 melanoma cells were implanted subcutaneously into Zyxin KO or WT C57BL/6 mice with intratumoral injection of cGAMP every 3 days (A). Volumes (B), weights (C), and photos (D) of melanoma revealed that Zyxin deficiency substantially suppressed tumor growth in the B16-F10 melanoma syngeneic model. (B, C) $Zyxin^{+/+}$ Vehicle group, $n = 8$; $Zyxin^{-/-}$ Vehicle group, $n = 8$; $Zyxin^{+/+}$ cGAMP group, $n = 8$; $Zyxin^{-/-}$ cGAMP group, $n = 9$. (E, F) Infiltration of CD4$^+$ and CD8$^+$ T lymphocytes and macrophages (F4/80$^+$) were imaged and evaluated in B16-F10 tumors; Zyxin deficiency diminished the proportion of TAMs but enhanced CD8$^+$ T lymphocytes in melanoma. Scale bars, 20 μm. (F) $n = 3$ per group. (G, H) Intratumoral CD86$^+$ and CD206$^+$ macrophages (F4/80$^+$) were imaged and evaluated in B16-F10 tumors; Zyxin deletion in mice decreased CD206$^+$ M2-type macrophages but increased CD86$^+$ M1-type macrophages in tumors, which potentially promoted antitumor immunity. Scale bars, 20 μm. (H) $n = 4$ per group. (I–K) Wild-type B16-F10 melanoma cells were implanted subcutaneously into WT or Zyxin KO mice, which were intraperitoneally injected with anti-PD-1 antibodies or vehicle every 3 days. Photos (I), volumes (J), and weights (K) of melanoma showed that anti-PD-1 therapy had a marginal effect on the B16-F10 melanoma syngeneic model in wild-type mice but significantly suppressed tumor growth in Zyxin KO mice. (J, K) $Zyxin^{+/+}$ Vehicle group, $n = 8$; $Zyxin^{-/-}$ Vehicle group, $n = 9$; $Zyxin^{+/+}$ anti-PD-1 group, $n = 8$; $Zyxin^{-/-}$ anti-PD-1 group, $n = 8$. Data information: unless otherwise indicated, $n = 3$ independent biological experiments (mean ± SEM). *$P < 0.05$, **$P < 0.01$, and ***$P < 0.001$, compared with the control condition (One-way ANOVA test and Bonferroni correction). Source data are available online for this figure.

Zyxin, which further modules cytokine production and exerts immunostimulatory functions.

Intervening in TBK1-Zyxin signaling led to a noticeable enhancement of cGAS-STING-mediated antitumor immunity, associated with a significant decrease of intratumoral TAMs and an enhanced infiltration of CD8$^+$ T lymphocytes. The diminished level of TAMs is correlated with an augmented antitumor immunity, in line with previous reports on TAM function (Colegio et al, 2014; Pan et al, 2020; Pittet et al, 2022). As also evidenced in immuno-deficiency mice, Zyxin KO probably relieves TAM-mediated immune suppression, from which the CD8$^+$ T-mediated antitumor effect was compromised. Besides, Zyxin deletion in IRF3 knockout KO mice still displayed a compromised intratumoral TAMs residency and growth inhibition of B16-F10 tumors, suggesting STING-TBK1-Zyxin signaling is partially independent with IRF3. The roles of cGAS-STING signaling in tumors are currently somewhat controversial (Samson and Ablasser, 2022). Rapid activation of cGAS-STING signaling could robustly inhibit tumor growth; however, tumor-derived cGAMP and chronic inflammation caused by DNA-damage response (DDR), chromosomal instability (CIN), mtDNA release, and dying cells may promote tumor progression (Bakhoum et al, 2018; Hou et al, 2018). These discrepancies could be due to different target cells and downstream effectors of cGAS-STING. We propose that cGAMP-induced STING-TBK1-Zyxin signaling maintains TAMs' residency within tumors that suppress antitumor immunity while the induction of robust IFN responses by STING-TBK1-IRF3 signaling promotes CD8$^+$ and NK cells functionality and TAMs pro-antitumoral polarization. Though these contradictory effects look paradoxical, we believe that they fine-tune the complex interactions of tumors and TME. The lack of additional cGAMP effects in Zyxin KO mice highlights the role of TAM residency in antitumor immunity. Zyxin may also influence other immune and tumor cells, which warrants further investigation.

These intriguing observations also indicate the widespread presence of TBK1-Zyxin signaling in macrophages under various physiological and inflammatory conditions. We hypothesize that the previously unrecognized signaling axis can negatively regulate antitumor immunity by controlling the tissue residency of macrophages. Given its marked suppressive effect on antitumor immunity, targeting this pathway might serve as an alternate strategy to enhance STING-mediated antitumor therapeutics, such as achieved by specifically blocking TBK1-mediated phosphorylation of Zyxin or proteins responsible for Zyxin translocation. We

also confirmed that combining Zyxin knockout with anti-PD-1 effectively reduced the growth of subcutaneously grown melanoma-derived cells in syngeneic murine models. The anti-PD-1 strategy is generally less effective in tumors with lower T-cell infiltration. However, the knockout of Zyxin creates a more favorable immune microenvironment of PD-1 immunotherapy by reducing TAM residency and increasing T-cell infiltration, thus offering a novel strategy for addressing tumors with minimal immune infiltration.

## TBK1-mediated Zyxin phosphorylation transmits nucleic acid sensing signals into mechanical cues

TBK1 stands in the signaling hubs controlling immunity, stress response, metabolism, and development (Zhou et al, 2020). Recent advances have defined a range of cellular event regulators as TBK1 targets, including IRF3 (Chen et al, 2016b; Liu et al, 2015), STING (Zhang et al, 2019; Zhao et al, 2019), AMPK (Zhao et al, 2018), AKT (Ou et al, 2011), OPTN and p62 (Pilli et al, 2012; Wild et al, 2011), DRP1 (Chen et al, 2020), and Rab7 (Ritter et al, 2020). Over the past two decades, a regulatory signal network centered around TBK1 has been gradually established and continually expanded. Research indicates that inhibiting TBK1 can enhance antitumor immunity (Jenkins et al, 2018; Sun et al, 2023; Xiao et al, 2017), but more comprehensive studies are needed to understand these potential mechanisms. This report presents multiple layers of evidence demonstrating that Zyxin is a substrate of TBK1. TBK1-mediated Zyxin phosphorylation was confirmed by a pronounced mobility shift of Zyxin in Phos-Tag electrophoresis, mass spectrometry analyses, and phospho-specific antibodies against Zyxin S142/S143 residues. Compared to the previously identified Zyxin kinase AKT1 (Chan et al, 2007; Kato et al, 2005), TBK1 generates substantial Zyxin modifications on various residues, with extremely high efficacy.

TBK1-mediated phosphorylation substantially altered the cellular distribution of Zyxin, a process significantly related to Zyxin's functions (Grange et al, 2013; Hoffman et al, 2006; Moody et al, 2009; Smith et al, 2010; Yoshigi et al, 2005). TBK1 is a critical stress kinase activated in various cellular stress scenarios. We believe whether Zyxin could participate in the signalosome is the key to TBK1-Zyxin interaction, such as in the case of STING signalosomes and MAVS signalosomes. TBK1-phosphorylated Zyxin at S143 transports Zyxin into focal adhesions, where it forms a large complex with VASP and α-actinin, guiding F-actin to organize into fibers (Moody et al, 2009; Smith et al, 2010; Yoshigi et al, 2005).

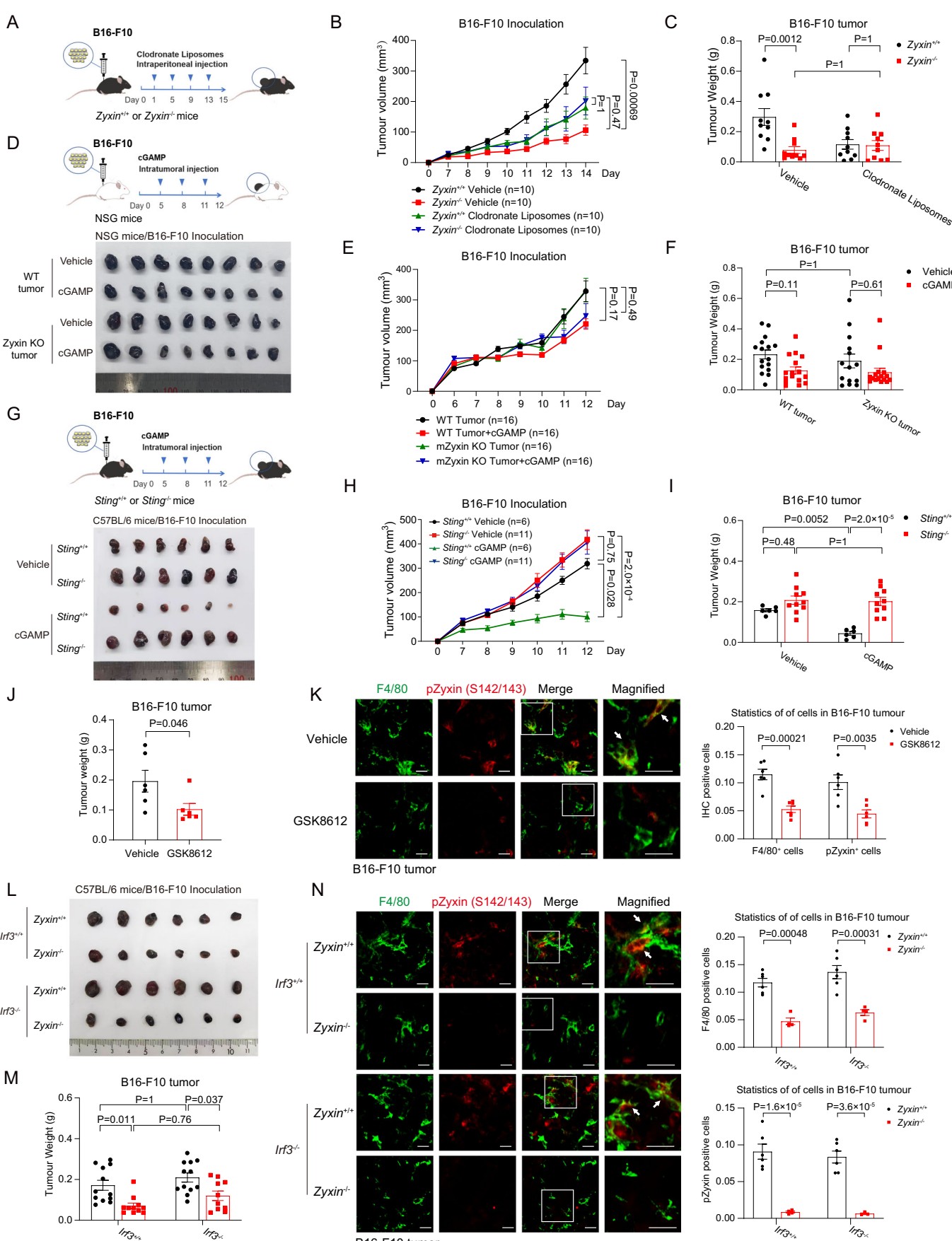

**Figure 7. STING-TBK1-Zyxin signaling retains TAM residency to suppress antitumor immunity.**

(A) A clodronate liposome-mediated macrophage depletion strategy was employed in the B16-F10 syngeneic antitumor model. (B, C) B16-F10 melanoma cells were implanted subcutaneously into Zyxin KO or WT C57BL/6 mice and treated with vehicle or clodronate liposomes that depleted macrophage in vivo. The volumes (B) and weights (C) of B16-F10 tumors indicated that macrophage depletion rescued tumor growth arrest phenotypes in Zyxin KO mice. (B, C) $n = 10$ per group. (D–F) Zyxin deletion in B16-F10 tumors failed to influence tumor growth in immunodeficient NSG mice. The photos (D), volumes (E), and weights (F) showed no difference in the growth of WT and Zyxin-deletion B16-F10 in NSG mice, and intratumoral cGAMP injection modestly reduced B16-F10 tumor growth and weight in NSG mice. (E) $n = 16$ per group. (F) WT Tumor group, $n = 16$; WT Tumor+cGAMP group, $n = 16$; mZyxin KO Tumor group, $n = 14$; mZyxin KO Tumor+cGAMP group, $n = 16$. (G–I) B16-F10 melanoma cells were implanted subcutaneously into $Sting1^{+/+}$ or $Sting1^{-/-}$ mice with intratumoral injection of cGAMP every 3 days. The photos (G), volumes (H), and weights (I) of B16-F10 tumors indicated that cGAMP-induced antitumor immunity depended on STING. (H) $Sting^{+/+}$ Vehicle group, $n = 6$; $Sting^{-/-}$ Vehicle group, $n = 11$; $Sting^{+/+}$ cGAMP group, $n = 6$; $Sting^{-/-}$ cGAMP group, $n = 11$. (I) $Sting^{+/+}$ Vehicle group, $n = 6$; $Sting^{-/-}$ Vehicle group, $n = 10$; $Sting^{+/+}$ cGAMP group, $n = 6$; $Sting^{-/-}$ cGAMP group, $n = 10$. (J, K) Statistics of tumor weight (J) and immunofluorescence (K) of B16-F10 melanoma from wild-type mice were shown, with or without TBK1 inhibitor (GSK8612). White arrows indicated the pZyxin+F4/80+ macrophages. Inhibition of TBK1 reduced tumor growth, F4/80+ macrophage infiltration, and pZyxin+ cell proportion. Scale bars, 20 µm. (J, K) $n = 6$ per group. (L–N) Statistics showed tumor photos (L), weight (M), and immunofluorescence (N) of B16-F10 melanoma from $Irf3^{+/+};Zyxin^{+/+}$, $Irf3^{+/+};Zyxin^{-/-}$, $Irf3^{-/-};Zyxin^{+/+}$, or $Irf3^{-/-};Zyxin^{-/-}$ mice. Zyxin deletion induced tumor growth arrest in IRF3 KO mice (L, M). Representative images showed that Zyxin deletion promoted antitumor immunity through downregulated F4/80+ macrophage proportion, a process independent of IRF3 (N). White arrows indicated the pZyxin+F4/80+ macrophages. (M) $Irf3^{+/+};Zyxin^{+/+}$ group, $n = 12$; $Irf3^{+/+};Zyxin^{-/-}$ group, $n = 11$; $Irf3^{-/-};Zyxin^{+/+}$ group, $n = 12$; $Irf3^{-/-};Zyxin^{-/-}$ group, $n = 10$. (N) $Irf3^{+/+};Zyxin^{+/+}$ group, $n = 6$; $Irf3^{+/+};Zyxin^{-/-}$ group, $n = 4$; $Irf3^{-/-};Zyxin^{+/+}$ group, $n = 6$; $Irf3^{-/-};Zyxin^{-/-}$ group, $n = 4$. Data information: unless otherwise indicated, $n = 3$ independent biological experiments (mean ± SEM). $*P < 0.05$, $**P < 0.01$, and $***P < 0.001$, compared with the control condition (One-way ANOVA test and Bonferroni correction). Source data are available online for this figure.

The proteins that sense this phosphorylation and aid Zyxin transportation remain unknown but could potentially involve VASP and α-actinin, which are associated with Zyxin (Grange et al, 2013; Hoffman et al, 2006). TBK1-mediated Zyxin phosphorylation upon DNA/RNA sensing is observed in various cells, especially macrophages. By studying the in vivo function of TBK1-Zyxin signaling in TAMs, we proposed that this non-canonical signaling has biological relevance, at least in macrophages. Activation of cGAS-STING signaling was reported to promote tumor metastasis (Bakhoum et al, 2018; Chen et al, 2016a), which, however, is distinct from its roles in regulating adhesion and migration in macrophages. Given that Zyxin is a well-defined mediator of mechanotransduction (Smith et al, 2010; Yoshigi et al, 2005) and a critical regulator of the Hippo-YAP signaling pathway (Harvey, 2015; Ma et al, 2016; Rauskolb et al, 2011; Zhou et al, 2018), we speculate that cGAS-STING signaling might regulate mechanosensitive events, in turn regulating the spatial perception and tissue residency of cells. In addition, Zyxin is known for signaling-triggered nuclear localization, where it transcribes mRNA expression (Ghosh et al, 2015; Nix et al, 2001; Wojtowicz et al, 2010; Zhao et al, 2022), such as regulating endothelial von Willebrand factor secretion and vascular repair (Han et al, 2017; Kang et al, 2021). However, TBK1-induced Zyxin phosphorylation appears not to induce a nuclear localization of Zyxin, contrasting its profound localization on focal adhesion.

This research provides a series of analyses highlighting the significance of TBK1-Zyxin signaling in response to innate nucleic acid sensing. Activating the cGAS-STING pathway restricted the mobility of TAMs, depending on the presence or activity of STING, TBK1, or Zyxin. Similarly, activating TBK1-Zyxin signaling increased macrophage adhesion to hard surfaces and the endothelial surfaces of blood vessels. While the physiological significance of these findings is not yet entirely clear, this study establishes a novel cellular function for cGAS-STING signaling and RLR-MAVS signaling—the regulation of macrophage adhesion and motility. The fact that this uncovered signaling is abundantly present in TAMs and typically plays adverse roles in STING-mediated immunity enhances our understanding of the complexity of signaling during antitumor immunity.

# Methods

**Reagents and tools table**

| Reagent/Resource | Reference or Source | Identifier or Catalog Number |
|---|---|---|
| **Experimental models** | | |
| HEK293T | ATCC | Cat#CRL-3216 |
| DLD1 | ATCC | Cat#CCL-221 |
| NMuMG | ATCC | Cat#CRL-1636 |
| B16-F10 | ATCC | Cat#CRL-6475 |
| HeLa | ATCC | Cat#CCL-2 |
| THP-1 | ATCC | Cat#TIB-202 |
| HUVEC | Wenzhou University | N/A |
| MC38 | Prof. Xiaojian Wang | N/A |
| Peritoneal Macrophages | This study | N/A |
| C57BL/6 mice wild-type (*M. musculus*) | SLAC Laboratory Animal | N/A |
| C57BL/6 mice *Zyxin*$^{-/-}$ (*M. musculus*) | Cyagen Biosciences | N/A |
| C57BL/6 mice *Sting*$^{-/-}$ (*M. musculus*) | Cyagen Biosciences | N/A |
| C57BL/6 mice *Irf3*$^{-/-}$ (*M. musculus*) | RIKEN BioResource Research Center | N/A |
| C57BL/6 mice *Sting*$^{flox/flox}$ (*M. musculus*) | Shanghai Model Organisms | N/A |
| C57BL/6 mice *Cx3cr1*$^{cre}$ (*M. musculus*) | The Jackson Laboratory | N/A |
| NSG mice (*M. musculus*) | SLAC Laboratory Animal | N/A |
| **Recombinant DNA** | | |
| | This study | Appendix Table S1 |

| Reagent/Resource | Reference or Source | Identifier or Catalog Number |
|---|---|---|
| **Antibodies** | | |
| Rhodamine phalloidin | Invitrogen | R415 |
| Anti-CD11b | Cell Signaling Technology | 46512S |
| Anti-pSTING (S365) | Cell Signaling Technology | 72971S |
| Anti-pSTING (S365) | Cell Signaling Technology | 62912S |
| Anti-STING | Cell Signaling Technology | 50494S |
| Anti-STING | Abcam | ab181125 |
| Anti-Zyxin | Abcam | ab109316 |
| Alexa Fluor® 488 Anti-Zyxin | Abcam | ab237072 |
| Anti-pZyxin (S142/143) | Cell Signaling Technology | 8467S |
| Anti-VASP | Cell Signaling Technology | 3132 |
| Anti-pTBK1 (S172) | Cell Signaling Technology | 5483S |
| Anti-TBK1 | Cell Signaling Technology | 3504S |
| Anti-TBK1 | Cell Signaling Technology | 38066S |
| Anti-TBK1 | Abcam | ab40676 |
| Anti-pIRF3 (S396) | Cell Signaling Technology | 4947S |
| Anti-pIRF3 (S396) | Cell Signaling Technology | 29047S |
| Anti-IRF3 | Cell Signaling Technology | 4302S |
| Anti-IRF3 | Abcam | ab76493 |
| Anti-CD4 | eBioscience | 14976680 |
| Anti-CD8 | eBioscience | 14080880 |
| Anti-F4/80 | Bio-Rad | MCA497RT |
| Anti-Flag (M2) | Sigma | F3165 |
| Anti-Flag M2 agarose | Sigma | A2220 |
| Anti-HA | Cell Signaling Technology | 3724S |
| Anti-HA | Sigma | H9658 |
| Anti-Myc | Cell Signaling Technology | 2276 |
| Anti-β-Actin | Sigma | A5441 |
| FITC anti-mouse CD45 | Biolegend | 147710 |
| BV421 anti-mouse F4/80 | Biolegend | 123131 |
| PE anti-mouse/human CD11b | Biolegend | 101208 |
| APC anti-mouse CD11c | Biolegend | 117310 |
| PE anti-mouse CD45 | Biolegend | 103106 |
| APC anti-mouse CD3 | Biolegend | 100236 |
| Pacific Blue™ anti-mouse CD4 | Biolegend | 116008 |
| PerCP/Cyanine5.5 anti-mouse CD8a | Biolegend | 100734 |
| TruStain FcX™ (anti-mouse CD16/32) | Biolegend | 101319 |
| FITC-AffiniPure Goat Anti-Rabbit IgG (H + L) | Jackson | 111-095-003 |
| FITC-AffiniPure Goat Anti-Mouse IgG (H + L) | Jackson | 115-095-003 |
| TRITC-AffiniPure Goat Anti-Rabbit IgG (H + L) | Jackson | 111-025-003 |
| TRITC-AffiniPure Goat Anti-Mouse IgG (H + L) | Jackson | 115-025-003 |
| **Oligonucleotides and other sequence-based reagents** | | |
| | This study | Appendix Table S3 |
| **Chemicals, Enzymes and other reagents** | | |
| Poly(I:C) LMW | Invivogen | Cat#tlrl-picw |
| cGAMP | Invivogen | Cat#tlrl-nacga23-02 |
| DMXAA | Selleck | Cat#S1537 |
| diABZI | Selleck | Cat#S8796 |
| MRT67307 | Selleck | Cat#S7948 |
| BX795 | Selleck | Cat#S1274 |
| GSK8612 | Selleck | Cat#S8872 |
| GSK690693 | Selleck | Cat#S1113 |
| mIFNα | PBL Assay Science | Cat#12100-1 |
| mIFNβ | PBL Assay Science | Cat#12401-1 |
| PMA/TPA | Beyotime Biotechnology | Cat# S1819 |
| Anti-IFNAR1 | Bio X Cell | Cat#BE0241 |
| Anti-PD-1 | Prof. Xiaojian Wang | N/A |
| Sendai Virus (SeV) | Charles River laboratories | Cat#VR-907 |
| Herpes Simplex Virus-1 (HSV-1) | Prof. Zhengfan Jiang | N/A |
| Doxycycline hydrochloride | Sangon Biotech | A600889; CAS: 24390-14-5 |
| Puromycin Dihydrochloride | Yeasen | 60210ES25; CAS: 58-58-2 |
| G418 Sulfate (Geneticin) | Yeasen | 60220ES03; CAS: 108321-42-2 |
| CMFDA | Yeasen | 40721ES50; CAS: 136832-63-8 |
| Q5® High-Fidelity 2X Master Mix | NEW ENGLAND BioLabs | Cat#M0492L |
| KOD Hot Start DNA Polymerase | Merck Millipore | Cat#71086 |
| AxyPrep Multisource Total RNA Miniprep Kit | Axygen | Cat#AP-MN-MS-RNA-50 |

| Reagent/Resource | Reference or Source | Identifier or Catalog Number |
|---|---|---|
| All-in-One cDNA Synthesis SuperMix | Bimake | Cat#24408 |
| EvaGreen qPCR MasterMix | Abm | Cat#MasterMix-R |
| **Software** | | |
| GraphPad Prism 8.0 | GraphPad | https://www.graphpad.com/scientific-software/prism/ |
| Origin 9.0 | OriginLab | https://www.originlab.com/index.aspx?go=PRODUCTS/Origin |
| ImageJ | ImageJ | https://imagej.nih.gov/ij/ |
| **Other** | | |

## Expression plasmids, reagents, and antibodies

Expression plasmids encoding Flag-, Myc-, or HA-tagged wild-type or mutants of human TBK1, AKT1, IRF3 2SA, and STING have been described previously (Meng et al, 2021; Zhang et al, 2017). Wild type or mutants Zyxin and VASP were constructed on the pRK5 vector. All coding sequences were verified by DNA sequencing. The list of recombinant DNA is provided in the Appendix Table S1.

The pharmacological reagents 2′3′-cGAMP (Invivogen), poly(I:C) (Invivogen), Dox (Sangon Biotech), diABZI (Selleck), puromycin (Yeasen, Shanghai, China), and G418 (Yeasen) were purchased. SeV (Cantell strain) was from Charles River Laboratories, and HSV-1 was from Dr. Zhengfan Jiang (Peking University, Beijing). Detailed information on all the antibodies applied in immunoblotting, immunoprecipitation, immunofluorescence, and immunohistochemistry is provided in Appendix Table S2. Monoclonal anti-pZyxin (S142/143), anti-TBK1, anti-pTBK1(S172), anti-IRF3, anti-pIRF3(S396), anti-pSTING (S365), anti-STING, anti-CD11b, anti-VASP and anti-Myc were purchased from Cell Signaling Technology. Anti-Zyxin was purchased from Abcam, and anti-HA, anti-α-tubulin, anti-β-Actin, and anti-Flag (M2) were purchased from Sigma-Aldrich. Anti-F4/80 antibodies were purchased from Bio-Rad, and anti-CD4 and anti-CD8 antibodies were purchased from Thermo Fisher Scientific. Detailed information on all the antibodies applied in immunoblotting, immunoprecipitation, immunofluorescence, and immunohistochemistry is provided in Appendix Table S2.

## Cell culture, transfections, and infections

HEK293, DLD1, HeLa, B16-F10, THP-1, and NMuMG cells were from ATCC. MC38 was from Dr. Xiaojian Wang (Zhejiang University, Hangzhou); no cell lines used in this study were found in the database of commonly misidentified cell lines maintained by ICLAC and NCBI Biosample. Cell lines were frequently checked for morphology under a microscope and tested for mycoplasma contamination but were not authenticated. HEK293, DLD1, HeLa, and NMuMG cells were cultured in DMEM medium with 10% fetal bovine serum (FBS) at 37 °C in 5% $CO_2$ (v/v), and B16-F10, MC38,

THP-1 cells were cultured in RPMI 1640 medium with 10% FBS. 100 ng/mL PMA (Beyotime) for 2 days induced THP-1 differentiation into macrophages. The Zyxin-Flag or TBK1-HA inducible expressing DLD1, STING-HA inducible expressing HeLa cells were generated by a lentiviral vector containing the inducible Tet-On system followed by ORF of Zyxin, TBK1 or STING and selected by G418 antibiotic at a concentration of 1500 μg/ml for seven days. LipofectAmine 3000 (Invitrogen, Thermo Fisher Scientific, Waltham, Massachusetts, USA) or polyethyleneimine (PEI, Polysciences, Warrington, Pennsylvania, USA) transfection reagents were used for plasmid transfection.

Primary peritoneal macrophages were obtained from C57BL/6 male mice at 6–8 weeks of age by the Brewer thioglycollate medium (Sigma-Aldrich) induced approach by standard protocol (Meng et al, 2021). Three days after the intraperitoneal injection of 3 mL of 3% thioglycollate medium, peritoneal macrophages were isolated and cultured in RPMI 1640 medium.

## Immunofluorescence and microscopy

Zyxin, STING, TBK1, or their mutants in PMs, DLD1, HeLa, HEK293, and NMuMG cells were treated as indicated or transfected with specified plasmids for 24 h before harvest to visualize the cytoskeleton and subcellular localization of endogenous Zyxin. The cells were fixed in 4% paraformaldehyde, blocked in 2% bovine serum albumin (Sigma-Aldrich) in PBS for 1 h, and incubated sequentially with primary antibodies anti-CD11b, anti-pZyxin (S142/143), anti-Zyxin, anti-VASP, anti-pSTING (S365), anti-STING, anti-pTBK1 (S172), anti-TBK1, anti-HA, anti-Myc, or anti-Flag (M2) and Alexa-labeled secondary antibodies (Jackson ImmunoResearch, West Grove, Pennsylvania, USA, 111-095-003; 115-095-003; 111-025-003; 115-025-003, 1:500 dilution) with extensive washing. F-actin was labeled with TRITC-phalloidin (Thermo Fisher Scientific). Slides were then mounted with VectaShield and stained with DAPI (Vector Laboratories, Burlingame, California, USA). Immunofluorescence images were obtained and analyzed using the Nikon Eclipse Ti inverted microscope or the Zeiss LSM710 and LSM880 confocal microscope.

## Time-lapse imaging and analysis of cell motility

After incubating primary PMs for 24 h, live-cell images were captured and analyzed using a Zeiss LSM880 confocal microscope and monitored for 4–6 h. At least 50 cells were analyzed under the indicated conditions. ImageJ, Chemotaxis, and Migration Tool analyzed and calculated the accumulation distance and velocity.

## Adhesion assay

HUVECs were cultured in 24-well plates until confluent. Following a 2 h pretreatment with the vehicle, DMXAA (S1537, Selleck) or GSK8612 (S8872, Selleck), $2 \times 10^5$ PMs were seeded onto either glass coverslips or the HUVECs monolayer. After a designated adherence period, non-adherent PMs were removed by washing three times with PBS. The remaining adherent PMs were labeled with anti-CD11b antibody, imaged, and quantified using a Zeiss LSM880 confocal microscope. This approach ensured that only firmly adhered PMs, identified by CD11b immunofluorescence, were quantified.

## Intravital imaging of adherent leukocytes in blood vessels

A skin flap model was employed to visualize leukocyte adhesion to blood vessels. Abdominal hair was removed with depilatory cream, and injection sites were marked on both sides of the abdominal skin. At the marked locations, equal volumes of vehicle or cGAMP (5 μg) were injected adjacent to blood vessels. After 2 h, FITC-Dextran (100 μg, 2000 kDa, Sigma-Aldrich) and rhodamine-6G (20 μg, Sigma-Aldrich) were administered via retro-orbital injection. Subsequently, abdominal skin flaps were harvested, mechanically fixed, and imaged using a Zeiss LSM880 confocal microscope to capture videos and images of leukocyte movement within subcutaneous vessels.

## CRISPR/Cas9-mediated generation of Zyxin KO cells

CRISPR/Cas9 genomic editing for gene deletion was performed as described (Ran et al, 2013). Guide RNA (gRNA) sequences targeting Zyxin exon were cloned into the pX330 plasmids. Together with the puromycin vector pRK7-puromycin, these constructs were transfected into HEK293, NMuMG, or F16-F10 at a ratio of 15:1 using LipofectAmine 3000 transfection reagent. Twenty-four hours after transfection, cells were selected by puromycin (1.5 μg/ml) for 72 h, and single clones were obtained by serial dilution and amplification. Clones were identified by immunoblotting with an anti-Zyxin antibody. gRNAs used in the experiments are also listed in the Appendix Table S3.

## Coimmunoprecipitations and immunoblottings

Liver lysates upon PMs and HEK293 cells with indicated treatments were lysed using a modified Myc lysis buffer (MLB) (20 mM Tris-HCl, 200 mM NaCl, 10 mM NaF, 1 mM NaV$_2$O$_4$, 1% NP-40, 20 mM β-glycerophosphate, and protease inhibitor (pH 7.5)). Cell lysates were then subjected to immunoprecipitation using antibodies of anti-Flag (M2), anti-Myc, or anti-HA for transfected proteins or using an anti-Zyxin antibody for endogenous proteins. After 3 to 4 washes with MLB, adsorbed proteins were resolved by SDS-PAGE (Bio-Rad) and immunoblotting with the indicated antibodies. Cell lysates were also analyzed using SDS-PAGE and immunoblotting to control the protein abundance.

## In vitro kinase assay

The in vitro kinase assay procedure was described previously (Wu et al, 2019). Recombinant protein Zyxin-GST was purified through the lac operon system in E. coli. The pGEX-6P-1 plasmid, which encodes Zyxin, was successfully transformed into BL21-competent cells. Recombinant protein expression was induced overnight with IPTG (1 mM) to achieve optimal expression levels. The E. coli cells expressing Zyxin were sonicated for efficient lysis, and the resulting lysate was then processed for purification using glutathione beads. HEK293T cells were transfected with plasmids to express indicated proteins, including Zyxin-Flag, TBK1-Myc, TBK1 K38A-Myc, AKT1-HA, and AKT1-Flag. Cells were lysed in modified MLB lysis buffer after 36 h of transfection, and immunoprecipitations were performed using anti-Flag (M2) or anti-Myc antibodies. With twice washes in MLB and two washes in kinase assay buffer (20 μM

ATP, 20 mM Tris-HCl, 1 mM EGTA, 5 mM MgCl$_2$, 0.02% 2-mercapto-ethanol, 0.03% Brij-35, and 0.2 mg/ml BSA, pH 7.4), the immunoprecipitated Zyxin-Flag and TBK1 WT/K38A-Myc or AKT1-Flag were incubated in the kinase assay buffer at 30 °C for 60 min on a THERMO-SHAKER. The reaction was stopped by adding a 2×SDS loading buffer, and the samples were subjected to SDS-PAGE and specified immunoblotting.

## Nano-liquid chromatography-tandem mass spectrometry analysis

Protein identification, characterization, and label-free quantification were performed by the Phoenix National Proteomics Core using nano-liquid chromatography coupled with tandem mass spectrometry (nano LC/MS/MS). Tryptic peptides were separated on a C18 column and analyzed using an LTQ-Orbitrap Velos mass spectrometer (Thermo Fisher Scientific). Protein identification was achieved by searching against the human or mouse RefSeq protein databases using the National Center for Biotechnology Information search engine. The identified STING-interacting proteins and Zyxin modifications are detailed in the Dataset EV1 and Dataset EV2, respectively.

## Animal studies

Zyxin$^{-/-}$ and Sting1$^{-/-}$ mice were on C57BL/6 background generated by Cyagen Biosciences, Irf3$^{-/-}$ mice were on C57BL/6 background generated by RIKEN BioResource Research Center, Sting1$^{flox/flox}$ mice were on C57BL/6 background generated by Shanghai Model Organisms, Cx3cr1$^{cre}$ mice on C57BL/6 background were purchased from The Jackson Laboratory. Both male and female littermates were used in all the experiments. The mice were assigned according to the double-blind principle of randomization. All the mice were bred and maintained in a pathogen-free animal facility at the laboratory animal center of Zhejiang University. The care of experimental animals was approved by the committee of Zhejiang University and followed Zhejiang University guidelines. The committee protocol approved under the corresponding author used in this study is ZJU20170658.

## Murine allograft growth of B16 melanoma or MC38 colon carcinoma

C57BL/6 mice and NSG mice designated for tumor injection were housed under specific pathogen-free (SPF) conditions. Six- to eight-week-old mice received subcutaneous injections of either $2 \times 10^5$ B16-F10 melanoma cells or $5 \times 10^5$ MC38 colon carcinoma cells. Tumor growth was monitored daily from the sixth-day post-inoculation (dpti), following protocols approved by the Zhejiang University Institutional Animal Care and Use Committee (IACUC). Tumor size was measured using calipers and calculated as the product of two perpendicular diameters (mm$^2$). Mice were euthanized before tumors reached 15 mm in any dimension, adhering to local ethical guidelines. Treatment regimens included: cGAMP, 10 μg intratumoral injection every 3 days, starting on day 5 post-implantation; Anti-PD-1 antibody, 20 μg intraperitoneal injection every 3 days for 7 days, starting the day after implantation; Clodronate liposomes, 150 μL intraperitoneal injection every 4 days, starting the day after implantation; and GSK8612

(TBK1 inhibitor), 5 mg/kg oral gavage daily, starting on day 3 post-inoculation.

## Murine models of pulmonary inflammation

Pulmonary inflammation was induced in C57BL/6 mice via intratracheal instillation of either cGAMP (10 µg/mouse every 4 days) or an equivalent volume of 0.9% saline under anesthesia. Mice were euthanized at 10 days post-instillation (dpi), and lung tissue was collected after perfusion.

## Tissue dissociation and flow cytometry

Experimental mice were euthanized at the indicated times, stripped of their loads of B16 melanoma or MC38 colon carcinoma tumors, and lungs were removed from mice with perfusion of PBS. B16 tumors were prepared into single-cell suspensions using mechanical grinding, and MC38 tumors and lung tissue were cut up using scissors and rotated at 37 °C for 15 min using a digestion solution (Collagenase type IV, 1 mg/ml, Worthington Biochemical Corporation; Dnase I, 100 µg/ml, Solarbio). The tissue suspensions were filtered using 35-µm filters. Single-cell was subjected to flow cytometry using CytoFlex (Beckman Coulter) and the following fluorescence-labeled antibodies from Biolegend: FITC anti-mouse CD45, BV421 anti-mouse F4/80, PE anti-mouse/human CD11b, PE anti-mouse CD45, APC anti-mouse CD3, Pacific Blue™ anti-mouse CD4, PerCP/Cyanine5.5 anti-mouse CD8a, Zombie Aqua™ Fixable Viability Kit. For pZyxin staining, cells were fixed on ice using 4% PFA for 10 min and permeabilized using 4 °C methanol to pZyxin (S142/143) and labeled with FITC secondary antibody, and IgG was used as a negative control.

## Histology and immunohistochemistry

For the immunohistochemistry assays, the tumor samples were dissected, fixed in 4% paraformaldehyde for 12 h at 4 °C, dehydrated overnight with 30% sucrose at 4 °C, embedded in optimal cutting temperature compound, and immediately frozen at −80 °C. Samples sectioned at a thickness of 10 µm were washed twice with PBS, permeabilized with 0.5% Triton X-100, blocked in 3% BSA in PBS for 30 min, and incubated sequentially with the primary antibodies: anti-CD11b (Cell Signaling Technology, 46512S; 1:100 dilution), anti-CD4 (eBioscience, 14976680; 1:100 dilution), anti-CD8 (eBioscience, 14080880; 1:100 dilution) and anti-F4/80 (Bio-Rad, MCA497RT; 1:100 dilution). After the sections were incubated with Alexa-labeled secondary antibodies (Jackson Laboratories, 111-095-003; 1:500 dilution) and extensively washed, they were mounted with Vectorshield and stained with DAPI (Vector Laboratories). Immunofluorescence images were obtained and analyzed using a Nikon Eclipse Ti inverted microscope or a Zeiss LSM 880 confocal microscope.

## Statistics and reproducibility

Quantitative data are presented as mean ± standard error of the mean (SEM) from at least three independent experiments. Statistical differences between multiple comparisons were analyzed using the one-way ANOVA test with Bonferroni correction when appropriate. Differences were considered significant at $P < 0.05$. If preserved and adequately processed, all samples were included in the analyses, and no samples or animals were excluded, except for mice with conventional surgery damage. No statistical method was used to predetermine sample size, and all experiments except those involving animals were not randomized. Immunoblotting and qRT-PCR experiments were repeated to a minimum of three independent experiments to ensure reproducibility. The investigators were not blinded to allocation during experiments and outcome assessment.

## Data availability

The mass spectrometry raw files have been deposited into the Mass Spectrometry Interactive Virtual Environment (MassIVE MSV000095470). FTP Download Link: ftp://massive.ucsd.edu/v08/MSV000095470/. The source data has been uploaded into BioStudies and is publicly available. The accession ID for the BioStudies database S-BSST1517 and the storage links: https://www.ebi.ac.uk/biostudies/studies/S-BSST1517.

The source data of this paper are collected in the following database record: biostudies:S-SCDT-10_1038-S44318-024-00244-9.

## Peer review information

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

## Acknowledgements

This research was sponsored by the National Key Research and Development Program of China (2021YFA1301401 to PX), the NSFC Projects (32321002, 31830052, and 31725017 to PX, 82201920 to RZ, and 81902915 to QZ), and the Fundamental Research Funds for the Central Universities. Thanks also to technical assistance by the Life Sciences Institute core facilities, Zhejiang University.

## Author contributions

**Ruyuan Zhou**: Conceptualization; Data curation; Formal analysis; Investigation; Methodology; Writing—original draft; Writing—review and editing. **Mengqiu Wang**: Data curation; Investigation; Methodology. **Xiao Li**: Investigation; Methodology. **Yutong Liu**: Investigation; Methodology. **Yihan Yao**: Data curation; Formal analysis; Methodology. **Ailian Wang**: Investigation. **Chen Chen**: Data curation; Investigation. **Qian Zhang**: Data curation; Investigation. **Qirou Wu**: Investigation. **Qi Zhang**: Resources. **Dante Neculai**: Formal analysis. **Bing Xia**: Resources. **Jian-Zhong Shao**: Resources. **Xin-Hua Feng**: Resources. **Tingbo Liang**: Resources. **Jian Zou**: Resources; Formal analysis. **Xiaojian Wang**: Resources; Formal analysis; Methodology. **Pinglong Xu**: Conceptualization; Resources; Data curation; Formal analysis; Supervision; Funding acquisition; Validation; Writing—original draft; Project administration; Writing—review and editing.

Source data underlying figure panels in this paper may have individual authorship assigned. Where available, figure panel/source data authorship is listed in the following database record: biostudies:S-SCDT-10_1038-S44318-024-00244-9.

## Disclosure and competing interests statement

The authors declare no competing interests.

# Expanded View Figures

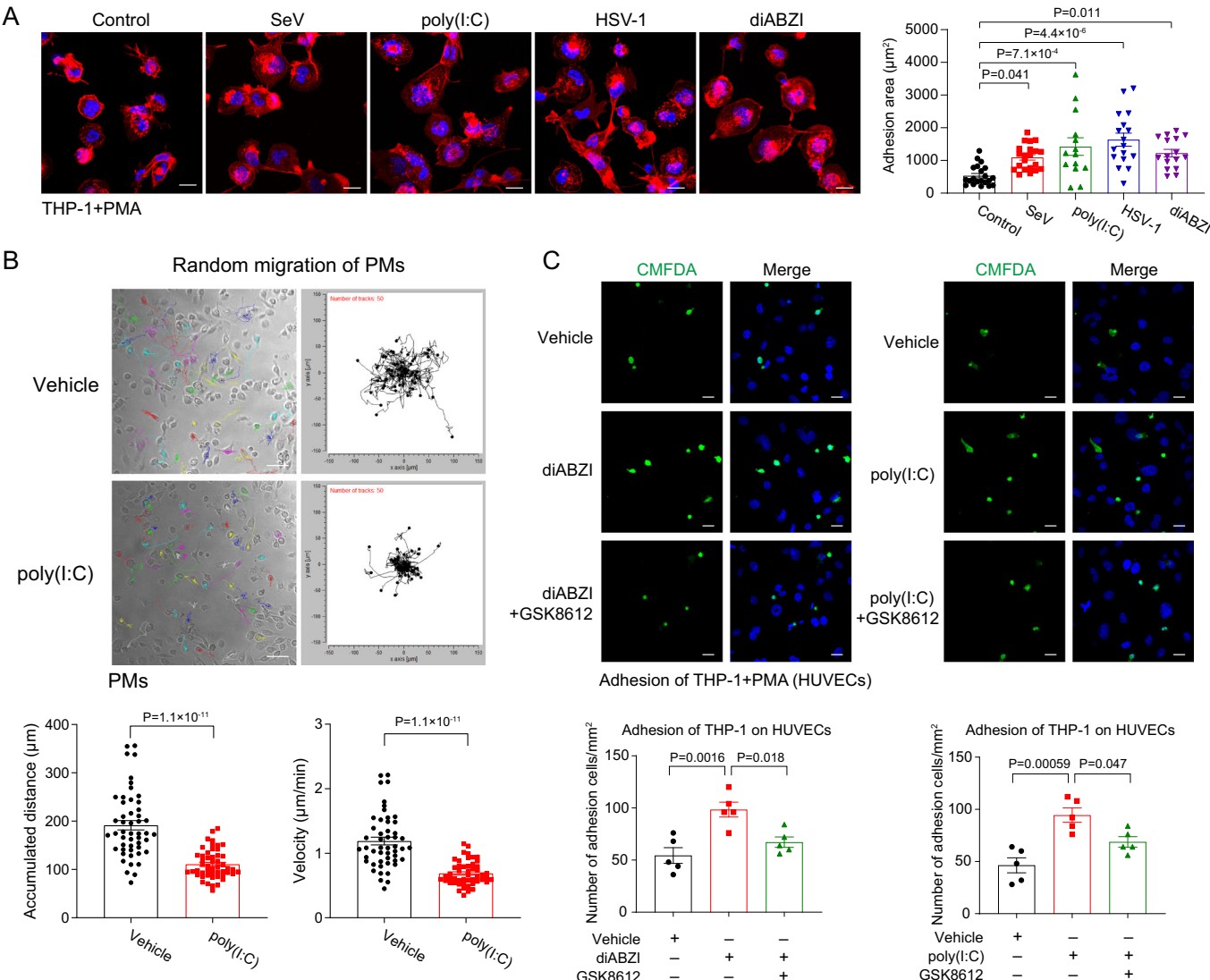

**Figure EV1. Nucleic acid sensing regulates the adhesion and motility of macrophages.**

(**A**) THP-1 cells, human monocytes, were differentiated into macrophages with PMA treatment, and their adhesion and migration were analyzed under the effects of SeV (RNA virus), poly(I:C) (RNA analog), HSV-1 (DNA virus), or diABZI (STING agonist). Scale bar, 20 µm. Control group, $n = 21$; SeV group, $n = 21$; poly(I:C) group, $n = 14$; HSV-1 group, $n = 17$; diABZI group, $n = 16$. (**B**) Representative migration plots indicated the movement of PMs upon the treatment of vehicle or poly(I:C). Statistics of the accumulated distance and velocity of cell movement were revealed. $n = 50$ cells per group. Scale bar, 100 µm. (**C**) diABZI or poly(I:C) promoted THP-1 adhesion to HUVECs, which was compromised by GSK8612, a TBK1 inhibitor. THP-1 cells were labeled by 5-chloromethyl fluorescein diacetate (CMFDA, green). Scale bar, 20 µm. $n = 5$ per group. Data information: unless otherwise indicated, $n = 3$ independent biological experiments (mean ± SEM). *$P < 0.05$, **$P < 0.01$, and ***$P < 0.001$, compared with the control condition (One-way ANOVA test and Bonferroni correction).

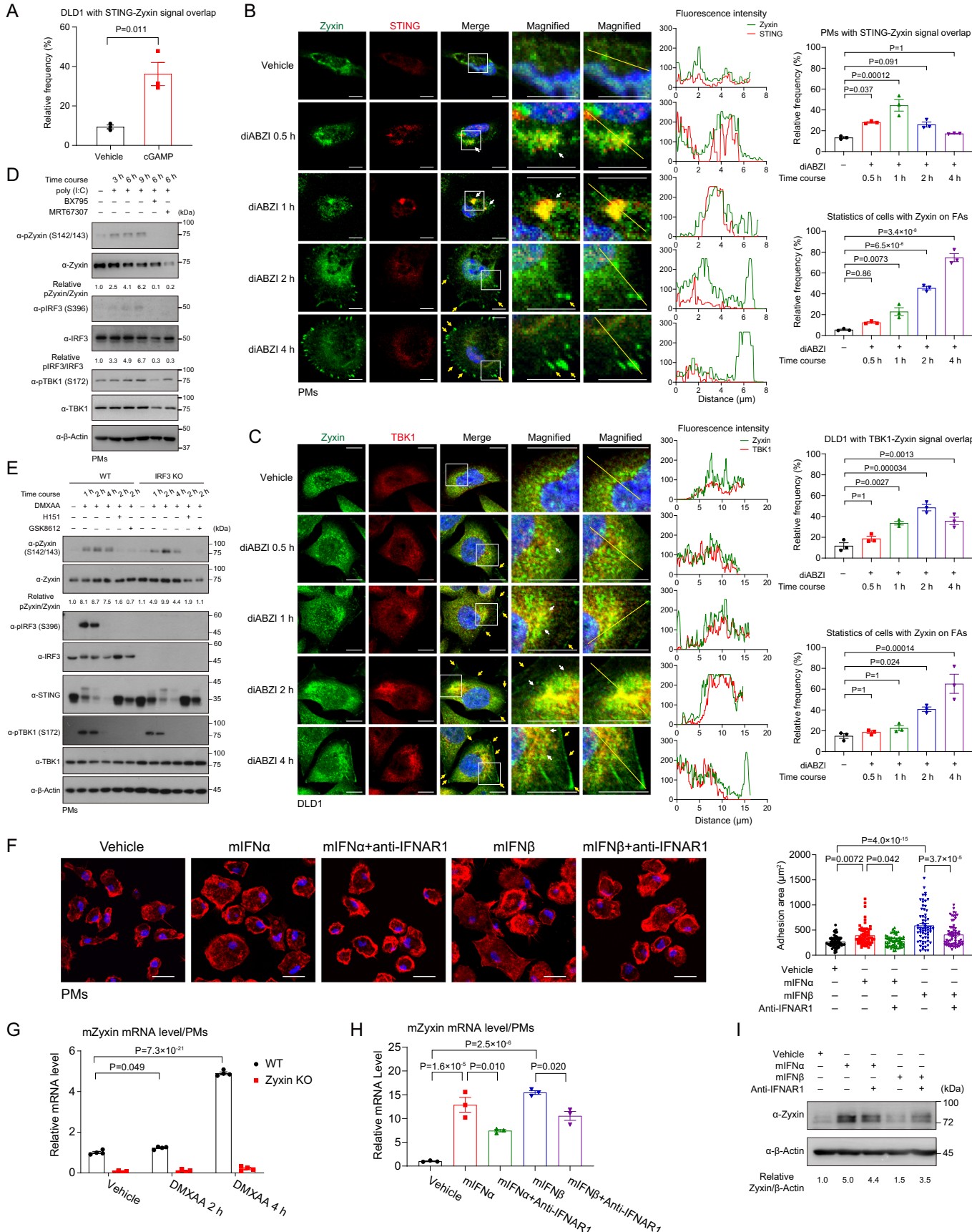

**Figure EV2.   STING or MAVS signalosomes recruit and phosphorylate Zyxin.**

(A) Statistics of cGAMP-induced Zyxin-STING signal overlap in Fig. 2B were shown. $n = 3$ per group. (B) Immunofluorescence imaging and statistics indicated an association of endogenous Zyxin proteins with STING signalosomes (white arrowed) in response to stimulation and its sequential distribution on focal adhesions (yellow arrowed) in primary macrophages. Scale bar, 5 μm. $n = 3$ per group. (C) Immunofluorescence and statistics showed diABZI induced an association of endogenous Zyxin with TBK1 signalosomes (white arrowed) and its sequential distribution on focal adhesions (yellow arrowed) in DLD1 cells. Scale bar, 5 μm. $n = 3$ per group. (D) Poly(I:C)-induced RNA sensing triggered Zyxin phosphorylation at S142/S143 residues in PMs. (E) Genetic ablation of IRF3 in primary macrophages failed to eliminate the DMXAA-induced phosphorylation of TBK1, STING, and Zyxin. (F) mIFNα (100 U, 6 h) or mIFNβ (100 U, 6 h) increased PM adhesion area, which was reversed upon treating anti-INFAR1 neutralizing. Scale bar, 20 μm. Vehicle group, $n = 68$; mIFNα group, $n = 63$; mIFNα+Anti-IFNAR1 group, $n = 59$; mIFNβ group, $n = 64$; mIFNβ+Anti-IFNAR1 group, $n = 65$. (G) An elevated level of Zyxin mRNA was detected upon the activation of cGAS-STING signaling at 4 h. $n = 4$ per group. (H, I) mIFNα (100 U, 6 h) and mIFNβ (100 U, 6 h) induced an increase in mRNA (H) and protein levels of Zyxin (I), which was reversed by an anti-IFNAR1 neutralizing antibody. (H) $n = 3$ per group. Data information: unless otherwise indicated, $n = 3$ independent biological experiments (mean ± SEM). $*P < 0.05$, $**P < 0.01$, and $***P < 0.001$, compared with the control condition (One-way ANOVA test and Bonferroni correction).

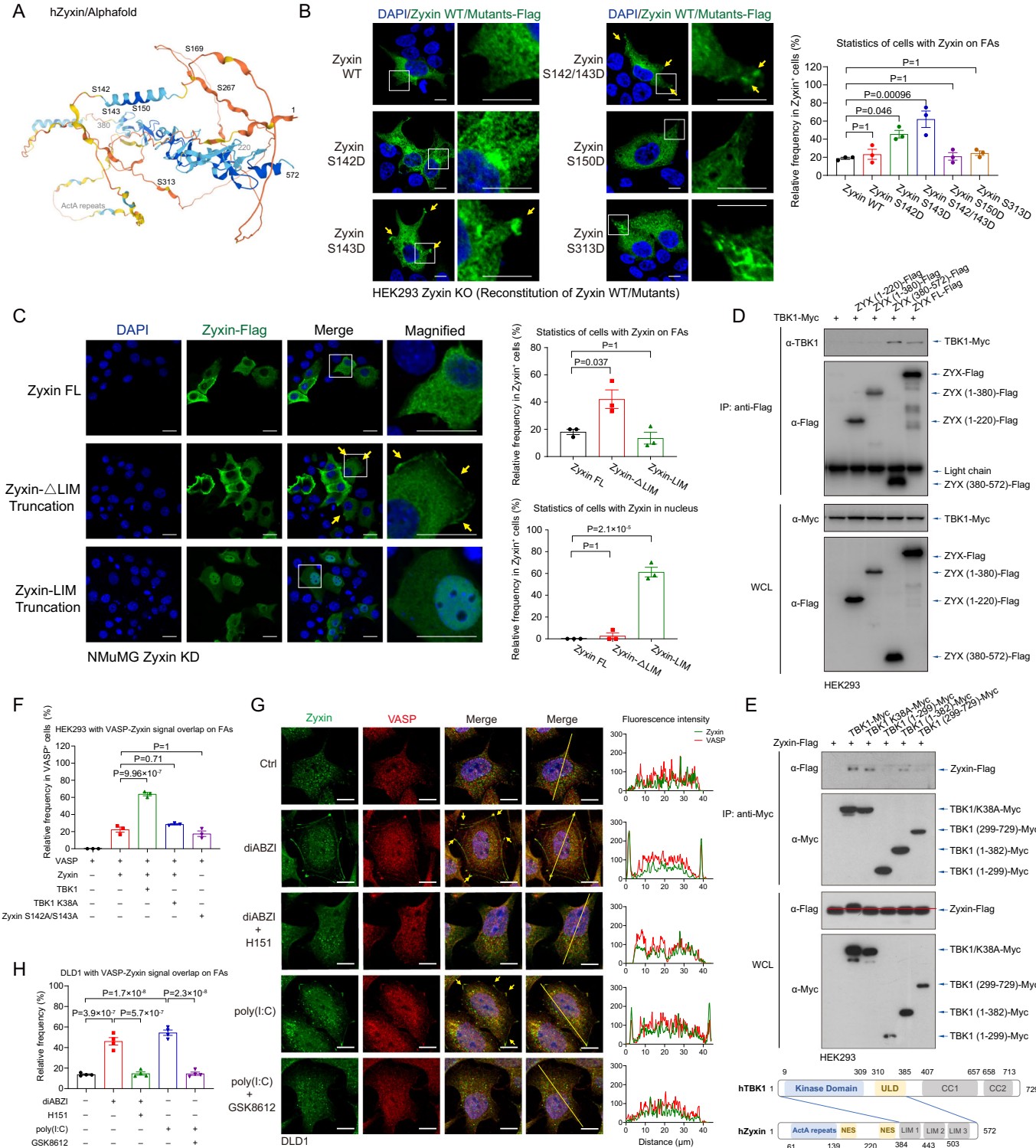

◀ **Figure EV3. TBK1 directly phosphorylates S143 residue to facilitate Zyxin focal adhesion localization.**

(A) Alphafold predicted the protein structure of human Zyxin proteins, implying an interaction between a long α-helix containing S143 and the C-terminal LIM domain. (B) Zyxin phosphomimetic mutants mimicking TBK1-mediated phosphorylation were individually reconstituted in Zyxin KO HEK293 cells; immunofluorescence imaging and statistics revealed a substantial subset of Zyxin S143D or S142D/S143D mutant localized on focal adhesions (yellow arrowed), in contradiction to wild-type Zyxin. Scale bar, 10 μm. $n = 3$ per group. (C) Immunofluorescence imaging and statistics indicated the cellular localization of Zyxin truncations in NMuMG cells. $n = 3$ per group. (D, E) Domain mapping by coimmunoprecipitations between TBK1 or Zyxin truncations showed that the kinase domain of TBK1 (a.a. 1–382) and the LIM domain of Zyxin (a.a. 380–572) were responsible for their mutual interaction. (F) Statistics of Zyxin-VASP signal overlap on focal adhesions in HEK293 cells are shown in Fig. 3K. $n = 3$ per group. (G, H) The signal overlap of endogenous Zyxin and VASP on focal adhesions (yellow arrowed) in DLD1 cells under TBK1 activation (diABZI or poly(I:C), 4 h) or inhibition (STING inhibitor H151 or TBK1 inhibitor GSK8612), respectively. Scale bar, 10 μm. (H) $n = 4$ per group. Data information: unless otherwise indicated, $n = 3$ independent biological experiments (mean ± SEM). *$P < 0.05$, **$P < 0.01$, and ***$P < 0.001$, compared with the control condition (One-way ANOVA test and Bonferroni correction).

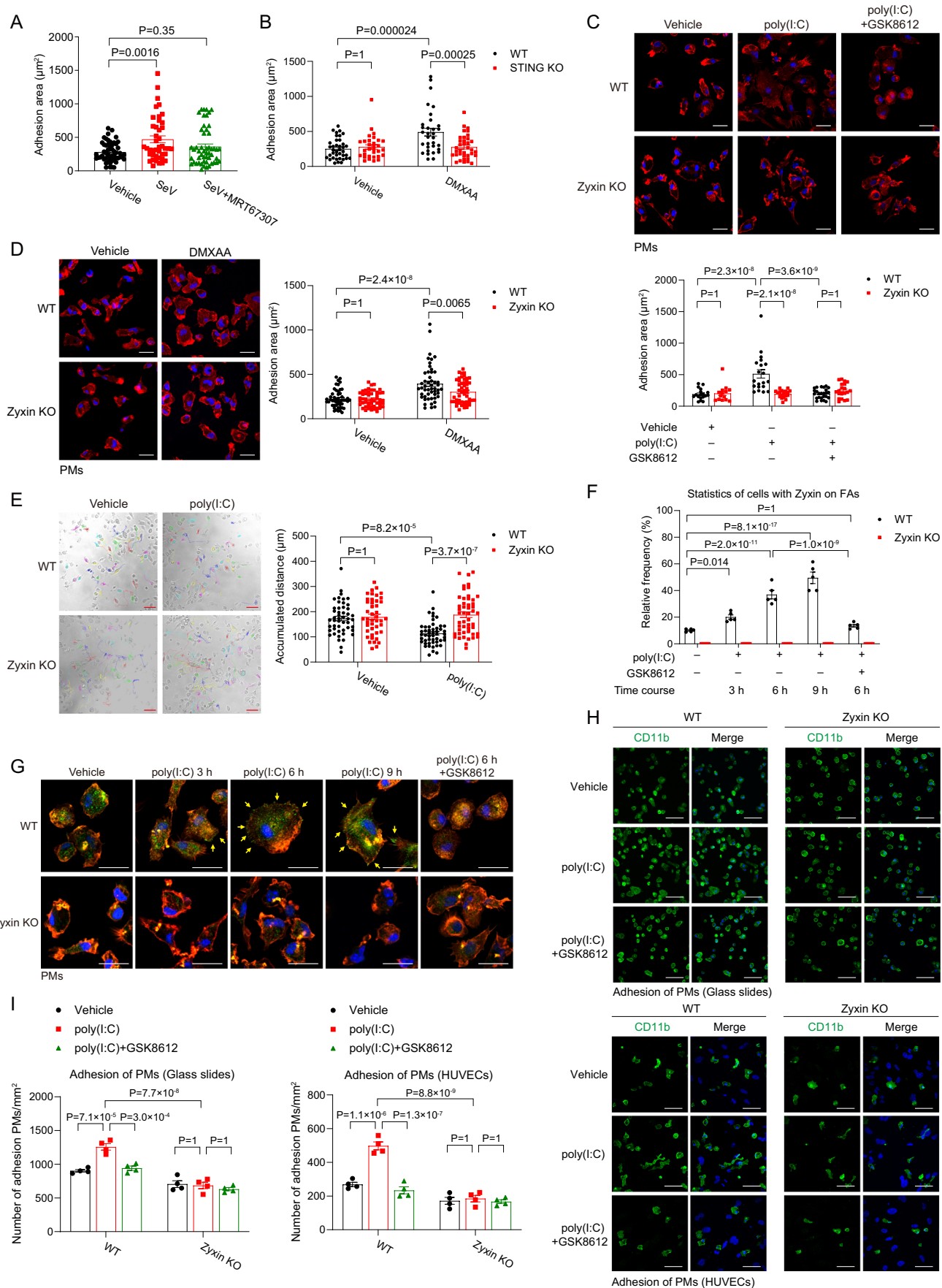

**Figure EV4.    The TBK1-Zyxin cascade restricts macrophage motility.**

(A, B) Statistics revealed that SeV infection upregulated the adhesion area of PMs, blocked by TBK1 inhibitor MRT67307 (A) and dependent on STING (B). (A) Vehicle group, $n = 52$; SeV group, $n = 42$; SeV+MRT67307 group, $n = 51$; (B) WT Vehicle group, $n = 38$; WT DMXAA group, $n = 32$; STING KO Vehicle group, $n = 31$; STING KO DMXAA group, $n = 36$. (C) Immunofluorescence and statistics revealed the adhesion of Zyxin KO macrophages upon the activation of innate RNA sensing by poly(I:C) transfection. The enhanced cellular adhesion induced by poly(I:C) was attenuated upon Zyxin deletion. Scale bar, 20 μm. WT Vehicle group, $n = 18$; WT poly(I:C) group, $n = 20$; WT poly(I:C) + GSK8612 group, $n = 25$; Zyxin KO Vehicle group, $n = 14$; Zyxin KO poly(I:C) group, $n = 19$; Zyxin KO poly(I:C) + GSK8612 group, $n = 20$. (D) Genetic ablation of Zyxin attenuated STING signaling-enhanced cellular adhesion, as revealed by immunofluorescence imaging and statistics of PMs from $Zyxin^{+/+}$ or $Zyxin^{-/-}$ mice treated with DMXAA (10 μg/mL, 4 h). Scale bar, 20 μm. $n = 50$ per group. (E) Macrophage motility inhibited by poly(I:C) transfection was relieved upon Zyxin deletion. Scale bar, 100 μm. WT Vehicle group, $n = 49$; WT poly(I:C) group, $n = 50$; Zyxin KO Vehicle group, $n = 44$; Zyxin KO poly(I:C) group, $n = 50$. (F, G) Immunofluorescence and statistics indicated Zyxin distribution at focal adhesions (indicated by the yellow arrows), induced by poly(I:C) transfection, was blocked by either TBK1 inhibitor GSK8612 or Zyxin deletion. Scale bar, 20 μm. (F) $n = 5$ per group. (H, I) Poly(I:C) transfection promoted the adhesion of murine PMs on glass slides and HUVEC, a process restored upon TBK1 inhibition or Zyxin deletion. Scale bar, 50 μm. (I) $n = 4$ per group. Data information: unless otherwise indicated, $n = 3$ independent biological experiments (mean ± SEM). $*P < 0.05$, $**P < 0.01$, and $***P < 0.001$, compared with the control condition (one-way ANOVA test and Bonferroni correction).

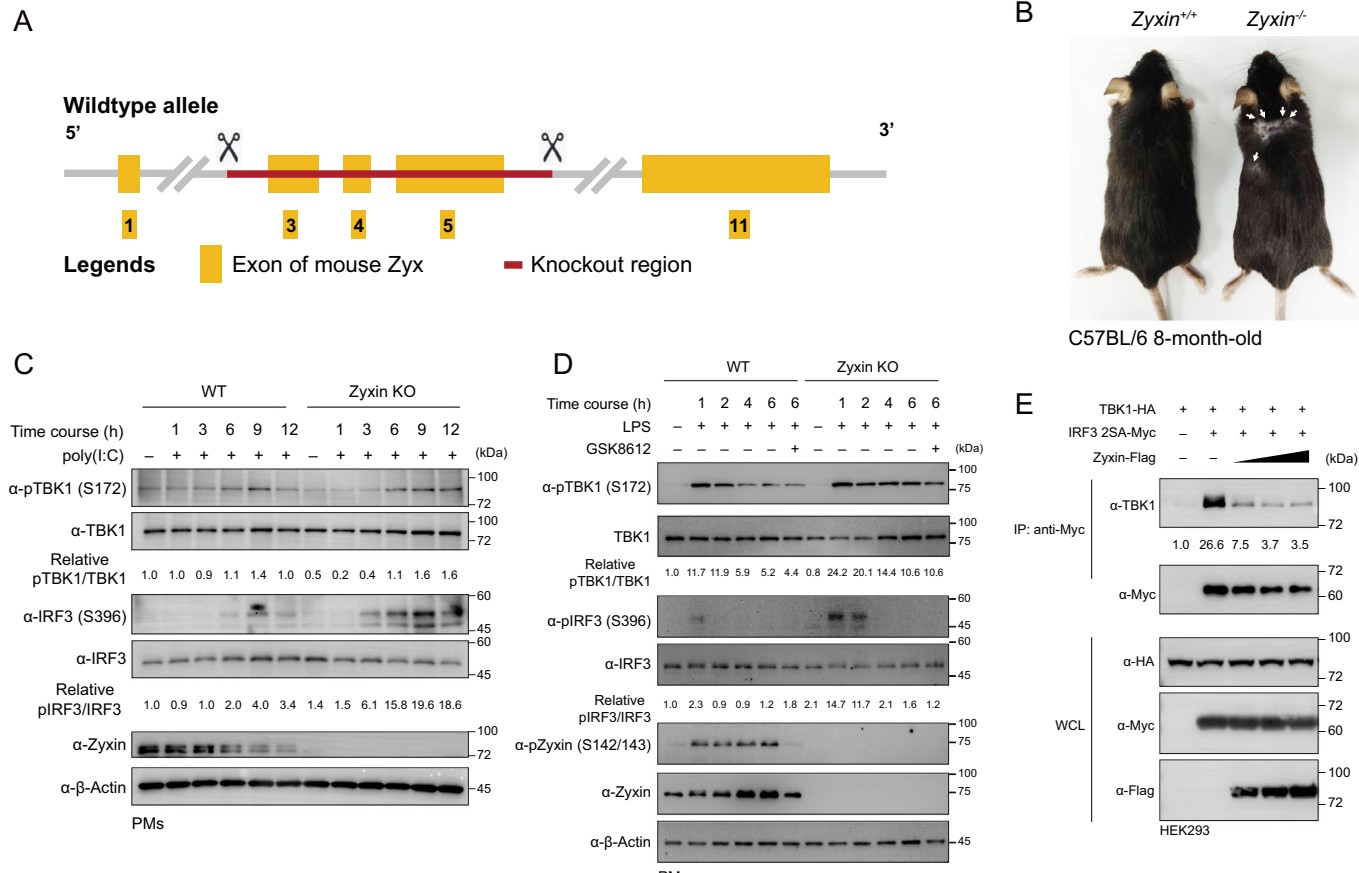

**Figure EV5. Zyxin deficiency enhances inflammatory responses.**

(A, B) The strategy of generating Zyxin KO mice by CRISPR-mediated genome editing was described (A), and spontaneous skin lesions in 8-month-old Zyxin knockout mice were found (B). (C, D) Poly(I:C)-induced RNA sensing and LPS-induced TLR4 signaling moderately increased pIRF3 (S396) levels in Zyxin-deficient cells. (E) Coimmunoprecipitation assays to detect the TBK1-IRF3 interaction in the absence or presence of Zyxin.

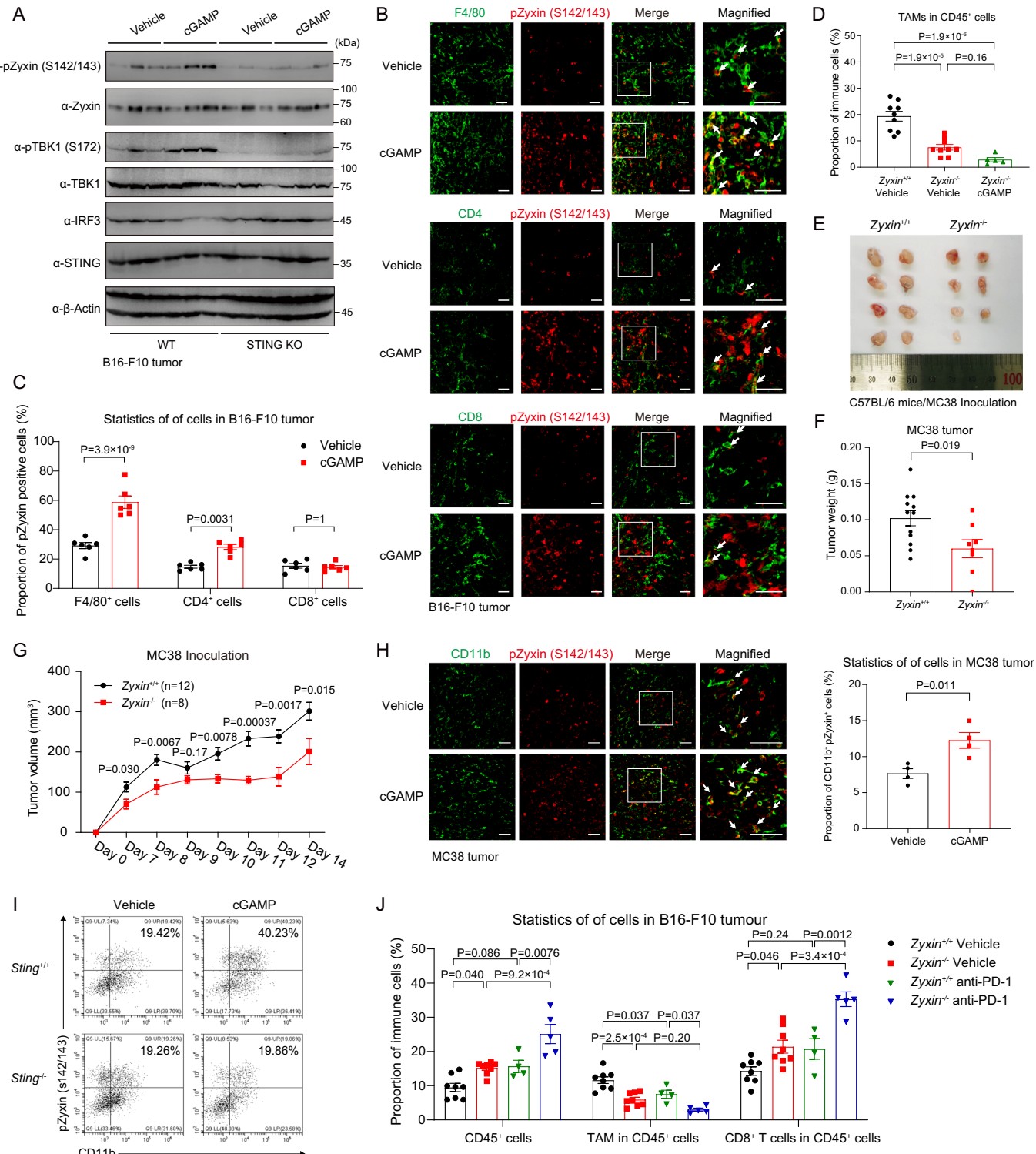

◀

**Figure EV6.  Interventing in STING-TBK1-Zyxin signaling improves antitumor immunity and synergizes with PD-1 immunotherapy.**

(A) Intratumoral injection of cGAMP induced the signal of phospho-Zyxin (S142/143) in melanoma specimens from WT mice but not STING KO mice. (B, C) Images and statistics showed cGAMP administration in mice induced the phospho-Zyxin mainly in F4/80$^+$ macrophages and, to a lesser extent, in CD4$^+$ T cells (indicated by the white arrows). Scale bars, 50 μm. (C) $n = 6$ per group. (D) Wild-type or Zyxin KO B16-F10 melanoma specimens in the absence or presence of cGAMP were analyzed by flow cytometry, and statistical results of TAMs (CD11b$^+$ F4/80$^+$) were displayed. *Zyxin$^{+/+}$* Vehicle group, $n = 9$; *Zyxin$^{-/-}$* Vehicle group, $n = 9$; *Zyxin$^{-/-}$* cGAMP group, $n = 5$. (E–G) Knockout of Zyxin compromised the growth of MC38 colon adenocarcinoma in the syngeneic tumor model. (F, G) *Zyxin$^{+/+}$* group, $n = 12$; *Zyxin$^{-/-}$* group, $n = 8$. (H, I) Immunofluorescence and flow cytometry showed that cGAMP increased pZyxin (S142/143) levels, majorly in myeloid cells (CD11b$^+$) of tumor tissues. STING KO mice showed a decreased proportion of CD11b$^+$pZyxin$^+$ cells. Scale bars, 50 μm. (H) $n = 4$ per group. (J) Statistics represented the individual abundance of TAMs (CD11b$^+$ F4/80$^+$) and CD8$^+$ T cells in B16-F10 tumors analyzed by flow cytometry, showing a substantially low proportion of TAMs and a high proportion of CD8$^+$ T cells under the combinational condition of Zyxin KO and anti-PD-1 administration. *Zyxin$^{+/+}$* Vehicle group, $n = 8$; *Zyxin$^{-/-}$* Vehicle group, $n = 8$; *Zyxin$^{+/+}$* anti-PD-1 group, $n = 4$; *Zyxin$^{-/-}$* anti-PD-1 group, $n = 5$. Data information: unless otherwise indicated, $n = 3$ independent biological experiments (mean ± SEM). *$P < 0.05$, **$P < 0.01$, and ***$P < 0.001$, compared with the control condition (One-way ANOVA test and Bonferroni correction).

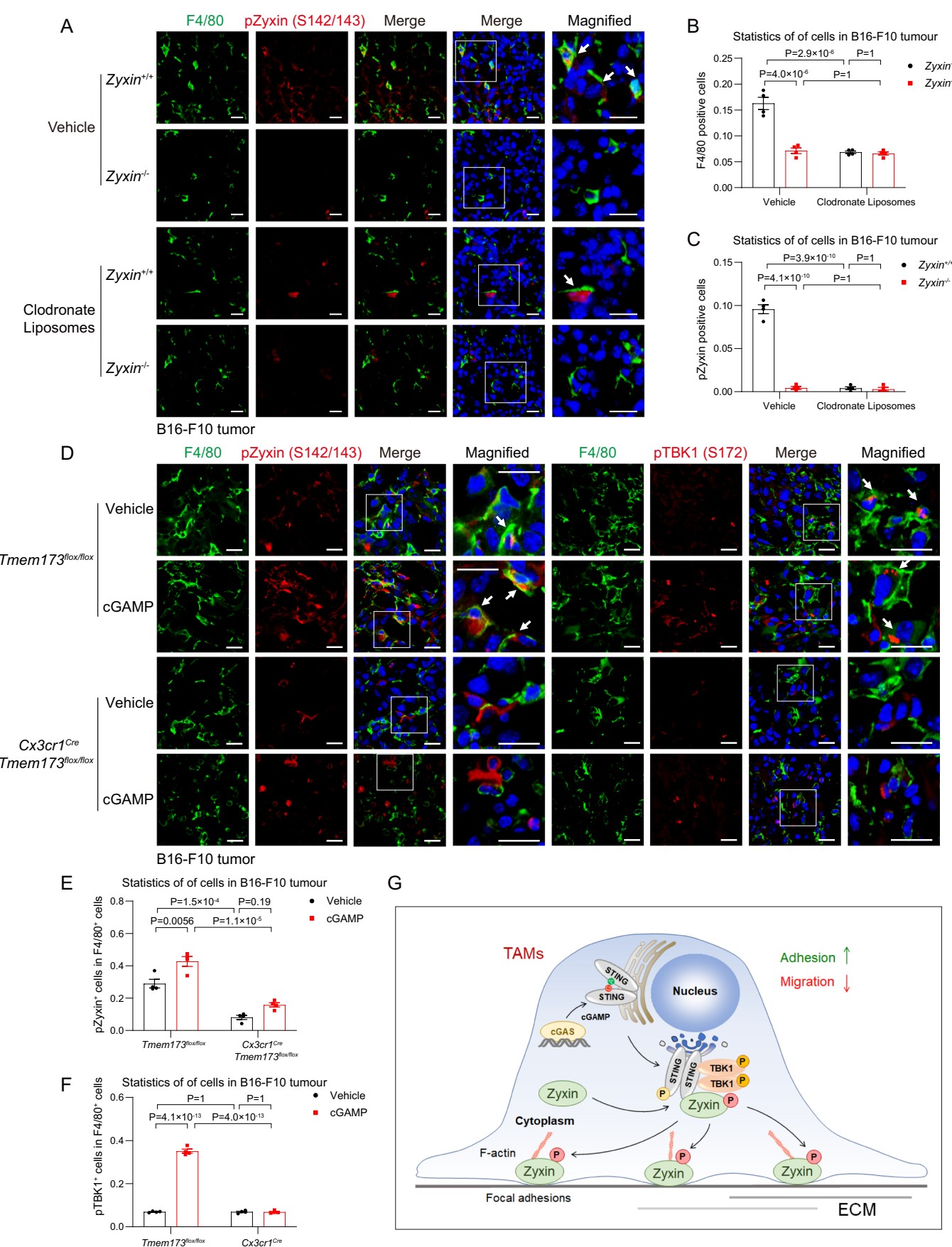

◀ **Figure EV7. STING-TBK1-Zyxin signaling retains TAM residency to suppress antitumor immunity.**

(A–C) Immunofluorescence imaging and statistics of F4/80$^+$ and pZyxin$^+$ cells in B16-F10 tumors. Administration of clodronate liposomes decreased F4/80$^+$ macrophages and pZyxin signals in B16-F10 tumors. Scale bars, 20 μm. White arrows indicated the pZyxin-positive TAMs. (B, C) $n = 4$ per group. (D–F) Immunofluorescence imaging and statistics of pZyxin (S142/143) and pTBK1 (S172) in B16-F10 tumors transplanted into *Sting1$^{flox/flox}$* or *Cx3cr1$^{cre}$Sting1$^{flox/flox}$* mice were shown, in the absence or presence of intratumoral injected cGAMP. cGAMP administration induced pZyxin and pTBK1 in TAMs, whereas tamoxifen-induced myeloid STING deficiency attenuated pZyxin and pTBK1 puncta in F4/80$^+$ macrophages. White arrows indicated the pTBK1-positive TAMs or pZyxin-positive TAMs. Scale bars, 20 μm. (E, F) $n = 4$ per group. (G) The diagram: cGAS-STING-Zyxin signaling converts biological signals into mechanical cues to reorganize the actin cytoskeleton that retains tumor-associated macrophages in the tumor microenvironment to mitigate antitumor immunity. Data information: unless otherwise indicated, $n = 3$ independent biological experiments (mean ± SEM). *$P < 0.05$, **$P < 0.01$, and ***$P < 0.001$, compared with the control condition (One-way ANOVA test and Bonferroni correction).

