## [Peer Review File · The EMBO Journal]

TBK1-Zyxin signaling controls tumor-associated macrophage recruitment to mitigate antitumor immunity

Ruyuan Zhou, Mengqiu Wang, Xiao Li, Yutong Liu, Yihan Yao, Ailian Wang, Chen Chen, Qian Zhang, Qirou Wu, Qi Zhang, Dante Neculai, Bing Xia, Jian-Zhong Shao, Xin-Hua Feng, Tingbo Liang, Jian Zou, Xiaojian Wang, and Pinglong Xu

Corresponding author: Pinglong Xu (xupl@zju.edu.cn)

Review Timeline:

Submission Date:	26th Nov 23
Editorial Decision:	14th Feb 24
Revision Received:	31st May 24
Editorial Decision:	22nd Jul 24
Revision Received:	7th Aug 24
Accepted:	19th Aug 24

Editor: Kelly Anderson

Transaction Report:

Dear Dr. Xu,

Thank you for submitting your manuscript for consideration by the EMBO Journal. It has now been seen by three referees whose comments are shown below.

Given the referees' positive recommendations, I would like to invite you to submit a revised version of the manuscript, addressing the comments of all three reviewers. I should add that it is EMBO Journal policy to allow only a single round of revision, and acceptance of your manuscript will therefore depend on the completeness of your responses in this revised version. It would be good to discuss you plan to address the referee concerns and I am available to do so in the coming weeks by email or zoom.

Thank you for the opportunity to consider your work for publication. I look forward to your revision.

Yours sincerely,

Kelly M Anderson, PhD
Editor, The EMBO Journal
k.anderson@embojournal.org

We realize that it is difficult to revise to a specific deadline. In the interest of protecting the conceptual advance provided by the work, we recommend a revision within 3 months (14th May 2024). Please discuss the revision progress ahead of this time with the editor if you require more time to complete the revisions.

Referee #1:

In their manuscript entitled "Mechanical control of tumor-associated macrophage residency by non-canonical cGAS-STING-Zyxin signaling", Zhou, Wang, Li et al. present evidence for a function of TBK1 in regulating cellular motility through phosphorylation of Zyxin, potentially specifically within macrophages, and a potential resulting function in suppressing anti-tumour immune responses. Overall, the findings are interesting and potentially represent an important advance in our understanding of the consequences of TBK1-mediated signalling. However, a number of items need to be addressed, such as a more thorough analysis of their microscopy data (especially the inclusion of proper quantifications), relevant controls for the interaction data, resolution of questions regarding the conclusions from the mouse experiments, and a delineation of cell-line specific effects. These points are described in more detail below.

Major points

1. Essentially all microscopy needs more detailed imaging and quantification. I would recommend going away from talking about STING and Zyxin "colocalization", as this is not clear in all experiments. I would prefer this to be called "some signal overlap". Quantifications are needed for all microscopy experiments. For example, the focal adhesion phenotype in Fig. 4 doesn't seem to manifest itself in all cells. The tumour analysis in Fig. 6/S6 is almost impossible to assess as very small images are provided and no overall quantification is provided. Fig. 3I should use microscopy just like the rest of the paper.
2. The diABZI experiment does not seem to completely recapitulate the DMXAA experiment, despite the authors' claims (no clear co-localisation between Zyxin and STING, but see regarding this point 1 above. However, diABZI does seem to induce changes in focal adhesions similar to DMXAA). This could be cell-type specificity or agonist specificity (also see point 5, below).
3. The interaction and phosphorylation data. Immunoprecipitations need negative controls (such as tubulin). The in vitro phosphorylation data description is misleading and needs changing. The reagents the authors use do not unambiguously show direct phosphorylation, as they used pulldown fractions from cells that could contain other proteins interacting with the proteins that were pulled down. Any possible effects could therefore potentially be indirect.
4. The mouse experiments have some ambiguity that should be resolved or at least discussed. For example, in Fig. 5B-D, the authors seem to make the conclusion that in the absence of Zyxin, tumours do not grow so well due to a less immunosuppressive environment. But then, why does cGAMP addition cause a positive effect in Zyxin wt? Shouldn't it drive Zyxin-dependent immunosuppression? And why is the Zyxin KO +cGAMP the same as either single? It should have less tumour growth than each single treatment. Obviously, tumour biology can be very complex, and I don't expect the authors to completely resolve this issue. But at least it should be discussed. An important experiment would be to repeat the tumour growth analysis from Fig. 5B in immunocompromised NSG mice (with the obvious caveats that needs to be kept in mind that these still have some macrophage functions intact). Furthermore, Fig. 5E-H should include wild-type plus cGAMP so that proper comparisons can be made.
5. It is not clear if the authors want to make a case for a macrophage-specific effect or not. Some of the cell biology seems to suggest macrophage specificity (for example, the HeLa cells in Fig. 2C do not seem to recapitulate the phenotype the authors describe, although the authors do suggest otherwise), and some of the tumour data on macrophage-depleted mice seem to suggest importance primarily in macrophages, but then some of the phenotypes do seem to be recapitulated in DLD1 and HEK293 cells. This should be clarified, at the very least by altering the wording of the manuscript, and inclusion in the discussion.
6. Following up from this, the macrophage depletion experiment is not entirely unambiguous. First, the authors don't show that their treatment in their specific experiment actually depleted macrophages (presumably that data exists within the microscopy, and could be resolved by quantification of the images, see also point 1, above). Second, Fig. 6B suggests that macrophage depletion actually improves the growth of tumours in zyxin^{-/-} mice. It is also unclear why there are effects in the absence of cGAMP treatment in Fig. 6. As before (point 4, above), tumour biology can be very complex, and I don't expect the authors to completely resolve this issue. But at least it should be discussed.
7. The Discussion seems a bit too much to repeat the findings from the results section, and could contextualise the results better with the published literature. For example, how would the authors put their findings of a function of cGAS-STING that potentially limits migratory activity with the published promotion of cell migration due to cGAS-STING (PMID 29342134 and 27225120). Furthermore, the question of potential cell type specificity (see also point 5) should be discussed.

Minor points

1. The authors use unscientific adjectives and adverbs such as "impressively", "dramatic", "profound", "critical", "profoundly" etc.

too much. Please remove them all.

2. The use of the term "non-canonical function" for the mechanism they have described should be removed from title, abstract and body of the manuscript. It is not defined unambiguously what a canonical function of cGAS is, although usually this term seems to be used to refer to cGAMP-dependent functions. However, the authors findings clearly do not describe functions that are obviously cGAMP-independent.
3. It is unclear from the text if the authors think that the Zyxin regulation is cGAS pathway specific. On the one hand, they have data indicating that other pathways that converge on TBK1 can also have this effect, but then they argue for an interaction between Zyxin and STING. I suppose this might be indirect as both proteins may bind TBK1? If so, this should be explained in the text.
4. The data from Fig. 4E-G seem to suggest that Zyxin is only partially responsible for the phenotypes. While the authors to say this, it isn't quite made clear enough.
5. Line 42, "cellular condensation", this is not a standard term, should be removed or changed.
6. Typos in lines 237 and 238 (should refer to the supplement).
7. Fig. S6 does not seem to be referenced in order.
8. Lines 343 and 344: "We also confirmed that combining Zyxin knockout with anti-PD-1 effectively reduces melanoma growth.". This should be changed, as the authors did not look at melanoma proper, but at a melanoma-derived cell line grown subcutaneously.

Referee #2:

In this manuscript the authors show that TBK1 - activated downstream of cytosolic DNA or RNA sensing - can modulate the migration/adhesion properties of macrophages. TBK1 promotes adhesion and limits migration in mouse peritoneal macrophages. The data supports a model where TBK1 directly phosphorylates the adhesion factor Zyxin at serine 142, promoting its localisation at focal adhesions.

The link between innate immune sensing and macrophage motility is interesting, and so far relatively under-studied. Most studies on DNA and RNA sensing focus on the TBK1-induced production of type I interferons etc, and so this is an interesting addition, particularly in the context of tumor immunology where STING agonists are promising anti-cancer agents (at least in mice). Thus, the significance and novelty of the findings are quite high.

My main comment is more about the description of the findings - the data clearly shows that this is a novel function of TBK1, not restricted to the cGAS-STING pathway. This should be made clearer in the title, abstract and discussion. The title currently states cGAS-STING-Zyxin as signalling axis (when TBK1-Zyxin is probably more accurate) and in the abstract (line 6) the statement that "cGAS-STING directly phosphorylates and mobilizes Zyxin" is also a little misleading. STING still activates TBK1, just as in canonical cGAS-STING signalling, so this is more an additional downstream function of TBK1 which should be made clearer.

Main issues:

1. Figure 1 nicely shows effects on macrophage adhesion and migration in mouse peritoneal macrophages - can similar experiments be performed with human macrophages?
2. For completion, the migration assays in 1B-E could also be shown with poly(I:C).
3. The STING-Zyxin co-localisation data in Fig. 2C is not very clear. Can the diABZI STING agonist be used like poly(I:C) in the TBK1-HA expressing cells (Fig. 3E-F)? Or in other responsive human cells (macrophages)?
4. Please quantify cell features in the microscopy images where possible, e.g. macrophage adhesion area in all experiments in Fig 4 etc.
5. It would be useful to show that Sendai virus or poly(I:C)-induced macrophage adhesion is also dependent on Zyxin (like shown for DMXAA in Fig. 4).
6. The finding that Zyxin deletion enhances STING-TBK1-IRF3 signaling is important, and might also be involved in the increased inflammatory phenotype in the deletion mice, rather than the effects on macrophage migration/adhesion. Thus, some of this data should go into the main figures. A similar experiment like Fig. S4C could also be carried out for poly(I:C)-induced RNA sensing.
7. Rather than affecting STING signalosomes as proposed, it is also possible that Zyxin competes with IRF3 for the interaction with TBK1 - this could be tested using co-immunoprecipitation with over-expression and deletion cells.
8. The observation that Zyxin deletion enhances STING activation confounds the tumor data in the mouse models. Given that STING-TBK1-IRF3 activation has potent anti-tumor effects in mice, the enhancement of canonical STING signaling could be responsible for the observed effects. A lot of weight rests on the results with IRF3 deletion mice (Fig. S6D-E) where no effect on B16 tumors is seen in the presence of absence of IRF3. However, this is different from many other studies where IRF3-dependent melanoma control has been observed (e.g. Woo et al, Immunity 2014) and so this will be controversial in the field. More substantial data would be needed to disentangle the role of IRF3- and Zyxin-dependent tumor control.

Minor suggestions:

1. Make sure acronyms are defined at first use, e.g. IRF3, NF-kB, TAM etc(p7)
2. Insert a few words introducing various cell types when used (human/mouse, cancer-derived/immortalized/primary), e.g. DLD1 cells (p8)
3. The layout of the data in Fig. 2H is unclear. Maybe separate into two graphs (e.g. pZyxin signal in individual cells, pZyxin in FAs separately) for clarity

Overall, I think the findings regarding TBK1-mediated control of macrophage adhesion are solid, novel and interesting, and cross-regulation with TBK1-IRF3 activation should be examined a little further. I am less convinced about the role of Zyxin in tumor control, given the observed enhancement in IRF3 signaling (which is interesting in itself). Some of the in vivo data (including the HSV-1 eye infection and the MC38 tumors are insufficiently controlled for potential enhanced STING-TBK1-IRF3 effects, and thus might need deeper investigation - potentially in a separate manuscript.

Referee #3:

Here the authors describe a STING-TBK1-dependent role of Zyxin in mediating macrophage tumour residency and migration. Overall the paper describes a novel pathway and phenotype of interest. I have some major and minor comments as below. Figure 1A - could the choices for different time points be explained in the text? It is unclear why some conditions are imaged at 9 hours and some at 1 hour post treatment. Inclusion of an interferon control would be important here as this seems to be a broad interferon response given the multiple DNA/RNA sensors which result in the same phenotype. Common feature of both pathways is TBK1.

Figure 5 - Clarify in the text that when referencing resistance to aPD-1 in melanoma the authors are specifically talking about the B16F10 model, not generally. Zyxin knockout results in resistance to cGAMP - attributing anti-tumour immune responses to a direct impact of the cGAS-STING-TBK-Zyxin axis. However, in contrast - Zyxin knockout sensitises to anti-PD-1. The response here is attributed to a reduction in TAMs overall mediated by anti-PD-1 - I would have thought this should be a T cell mediated response? Is it the case that a T-cell activation, mediated by anti-PD-1, is independent of Zyxin but the cGAMP response requires Zyxin-pathway activation in TAMs? I am not convinced by the authors explanation of these data.

Figure 6 - Treating Zyxin knockout with clodronate liposomes actually increases tumour growth - the stats here are not included, however there is clear separation in the curves. This is a slightly confusing result and the authors do not comment on it. If the PD-1 response (Figure 5) is mediated by TAMs, why is this not reflected in these data?

The FACS analysis of the tumours is quite broad - F4/80 is a broad macrophage marker and the potential repolarisation of the macrophages is not considered, rather that absolute numbers are quantified. This should be included as a limitation (that could address some of the confusing results above). The role of Zyxin in macrophage polarisation would be of interest, although is unfortunately not included in these data.

Too many references are included in a short piece

In Methods -

Following ethical permission, the maximum diameter of the tumors is limited to 15 mm in one dimension, by which the mice were sacrificed and defined as dead due to tumor burden. - please rephrase as "Mice were killed before tumors reached 15 mm in any one dimension in accordance with local ethical guidelines."

In Introduction

Signalingly - this is not a recognised expression. Please rephrase.

Figure 2H should be revised - the use of a striped section of bar is confusing.

Throughout - there are many instances in the text where the authors' meaning is not clear. Additional editing would be overall helpful to improve this manuscript. This is a particular issue in methods, although occurs throughout the manuscript.

Response to the reviewers' comments:

We sincerely thank the reviewers and editor for their outstanding efforts in evaluating our manuscript and their generally positive and very constructive comments. We appreciate reviewers considering this work original, influential, and significantly advanced in conception. This cross-disciplinary manuscript reports a TBK1-Zyxin axis underlying cGAS-STING or RLR-MAVS signaling and the mechanistic insight into innate immune control of macrophage residency and its significance in pathological contexts. We have made substantial efforts to perform a substantial set of experiments on new in vivo and cellular models, which addressed and clarified all the concerns raised by the reviewers and significantly expanded the innovative idea.

In particular, we provide additional evidence, including:

(1) Confirmed by immunodeficient NSG mice that Zyxin deletion in B16-F10 tumors did not affect tumor growth, supporting that the tumor-inhibitory effect observed in Zyxin knockout mice is mediated through an enhanced antitumor immunity.

(2) Validated in $Irf3^{-/-};Zyxin^{-/-}$ mice that are revealing that Zyxin deletion downregulated F4/80⁺ macrophage proportion and suppressed tumor growth in $Irf3^{-/-}$ mice, suggesting that the promoting antitumor immunity upon Zyxin deletion was at least partially independent of the classical TBK1-IRF3 axis.

(3) Discovered a potential effect of Zyxin in regulating macrophage polarization: Zyxin deletion in mice decreased CD206⁺ M2-type macrophages and increased CD86⁺ M1-type macrophages in tumors, which probably promoted antitumor immunity.

(4) Observed the signal overlap of endogenous Zyxin and VASP proteins at focal adhesions, increased upon the activation of cGAS-STING signaling but diminished upon the inhibition of either STING or TBK1; and the comparison of WT and TBK1-resistant Zyxin and VASP at focal adhesions.

(5) Revealed an increased activation of inflammatory responses upon Zyxin deletion in vivo: Zyxin-deficiency mice showed more severe symptoms of cGAMP-induced pulmonary inflammation.

(6) Demonstrated that activating RNA/DNA sensing pathways also regulate the adhesion of differentiated human macrophage (PMA-induced THP-1 cells), a process that could be prevented by TBK1 inhibition.

(7) Confirmed that this TBK1-Zyxin axis activated by innate RNA sensing also regulated macrophage adhesion and motility, similar to DNA sensing.

(8) Revealed by an elaborating time-course analysis to show the dynamic changes of signaling overlap between Zyxin and STING or TBK1, supporting their functional interactions during innate immune responses.

(9) Demonstrated that TBK1 directly phosphorylated Zyxin at S142/143 residues using an in vitro kinase assay with bacterial recombinant Zyxin proteins and kinase inhibitors.

(10) Confirmed a potential competition between IRF3 and Zyxin upon TBK1-mediated modification.

(11) Revealed that interferons, produced by the classic TBK-IRF3 signaling, could induce an increase in Zyxin mRNA and protein levels and aid in regulating macrophage adhesion.

(12) Strengthened the proposed mechanism and data quality with various improved and proper image quantifications, controls, delineation, and the interpretation of the observations.

*These substantial sets of new or updated data are now presented in Figs 2B, 2C, 2E, 2F, 2H, 2I, 3A, 3E, 3G, 3H, 3K, 4B, 4D, 4F, 5A, 5B, 5C, 5D, 5E, 6E, 6F, 6G, 6H, 7D, 7E, 7F, 7G, 7H, 7I, 7J, 7K, 7L, 7M, 7N, 7O, 7P, and Fig. EV1A-1C, EV2A-2H, EV3B, EV3D, EV3H, EV4A-4C, EV4E-4I, EV5C-5E, EV6K, EV7B-6C, EV7E-7F in the revised manuscript. Meanwhile, we have restructured data sets and revised the text carefully with necessary corrections and more precise descriptions, as well as the compliance of EMBO policies and formats. We believe that after these efforts, we have adequately addressed all concerns/questions raised by the reviewers, and the revised manuscript is substantially improved and suitable for publication in *The EMBO Journal*. Below are our point-by-point responses to the reviewers' comments:*

Remarks to the Author:

Referee #1:

In their manuscript entitled "Mechanical control of tumor-associated macrophage residency by non-canonical cGAS-STING-Zyxin signaling", Zhou, Wang, Li et al. present evidence for a function of TBK1 in regulating cellular motility through phosphorylation of Zyxin, potentially specifically within macrophages, and a potential resulting function in suppressing anti-tumour immune responses. Overall, the findings are interesting and potentially represent an important advance in our understanding of the consequences of TBK1-mediated signalling. However, a number of items need to be addressed, such as a more thorough analysis of their microscopy data (especially the inclusion of proper quantifications), relevant controls for the interaction data, resolution of questions regarding the conclusions from the mouse experiments, and a delineation of cell-line specific effects. These points are described in more detail below.

Major points

1. Essentially all microscopy needs more detailed imaging and quantification. I would recommend going away from talking about STING and Zyxin "colocalization", as this is not clear in all experiments. I would prefer this to be called "some signal overlap".

Quantifications are needed for all microscopy experiments. For example, the focal adhesion phenotype in Fig. 4 doesn't seem to manifest itself in all cells. The tumour analysis in Fig. 6/S6 is almost impossible to assess as very small images are provided and no overall quantification is provided. Fig. 3I should use microscopy just like the rest of the paper.

We sincerely thank the reviewer for these very positive comments and valuable suggestions. Following the reviewer's advice, we conducted quantitative fluorescence intensity analyses on the immunofluorescence images, including Figs. 2B, 2C, 2E, 2F, EV2A, EV2B, 3K, EV3H and quantified cell proportions of Figs. 2F, EV2B, 3E, EV3A, EV3C, 4A, 4C, 4E, 5C, 6E, 6G, 7L, 7P, EV7A, EV7D. These quantified graphs can now be found at Figs. 2H, 2I, 3E, EV3B, EV3D, 4B, 4D, 4F, 5C, 6F, 6H, 7K, 7O, EV7B, EV7C, EV7E, EV7F. Furthermore, we have improved the quality of representative immunofluorescence images used for analyses in Figs. 6E, 6G, 7L, 7P, EV7A, and EV7D, making the images more prominent and visually expressive of our perspectives. In addition, we have revised the description of the "colocalization" between Zyxin to more accurate "signal overlap". Following the reviewer's instruction, these changes significantly improved the data quality and further supported our proposed mechanism.

Besides, the subcellular localization in original Fig. 3I of wild-type Zyxin (Zyxin WT) and the S142A/143A mutant with VASP was now visualized by confocal microscopy (Fig. Figs. 3K and EV3H), more evidently supporting that TBK1 promoted a localization of Zyxin and VASP at focal adhesions, while the kinase-dead TBK1 failed to. Zyxin localization at focal adhesions was reduced upon preventing TBK1-mediated phosphorylation through S142A/143A mutation. We also observed the signal overlap of endogenous Zyxin and VASP in DLD1 cells (Fig. EV3H), regulated distinctly by TBK1 activation (diABZI or poly(I:C) treatment) and inhibition (treatment of STING inhibitor H151 or TBK1 inhibitor GSK8612). These visual observations further strengthened our proposition.

2. The diABZI experiment does not seem to completely recapitulate the DMXAA experiment, despite the authors' claims (no clear co-localisation between Zyxin and STING, but see regarding this point 1 above. However, diABZI does seem to induce changes in focal adhesions similar to DMXAA). This could be cell-type specificity or agonist specificity (also see point 5, below).

Thanks for this insightful observation and suggestion. We have conducted quantitative fluorescence intensity analyses (Figs 2B, 2C, 2E, 2F, and EV2B) and observed signal overlaps between Zyxin and STING in DLD1 and HeLa cells stimulated by either cGAMP or diABZI, as well as some of their signal overlap in DLD1 cells when stimulated by poly(I:C) or diABZI. Per the reviewer's advice, we performed the experiments in primary peritoneal macrophages using diABZI, from which diABZI induced the Zyxin aggregation on focal adhesions similar to DMXAA, as well as prominent signal overlaps

between Zyxin and STING in these primary macrophages (Fig EV2A). Therefore, we believe a cell-type-specific difference accounts for what we have seen between distinct STING agonists.

3. The interaction and phosphorylation data. Immunoprecipitations need negative controls (such as tubulin). The in vitro phosphorylation data description is misleading and needs changing. The reagents the authors use do not unambiguously show direct phosphorylation, as they used pulldown fractions from cells that could contain other proteins interacting with the proteins that were pulled down. Any possible effects could therefore potentially be indirect.

Thanks for the reviewer's positive comments and suggestions. We have now included negative controls (α - β -Actin) in co-IP analyses of endogenous complexes, such as in Fig 3A. We also changed the wording of Fig 3F and repeated the in vitro kinase assay with a bacterial recombinant GST-Zyxin. As shown in Fig. 3G, TBK1 could effectively phosphorylate recombinant GST-Zyxin at S142/143 residues, while kinase-dead TBK1 did not. Furthermore, TBK1 inhibitor GSK8612 was added in this in vitro assay, which blocked the signal of pZyxin, while AKT1 inhibitor GSK690693 did not. These consistent observations thus support direct phosphorylation of Zyxin proteins by TBK1.

4. The mouse experiments have some ambiguity that should be resolved or at least discussed. For example, in Fig. 5B-D, the authors seem to make the conclusion that in the absence of Zyxin, tumours do not grow so well due to a less immunosuppressive environment. But then, why does cGAMP addition cause a positive effect in Zyxin wt? Shouldn't it drive Zyxin-dependent immunosuppression? And why is the Zyxin KO +cGAMP the same as either single? It should have less tumour growth than each single treatment. Obviously, tumour biology can be very complex, and I don't expect the authors to completely resolve this issue. But at least it should be discussed. An important experiment would be to repeat the tumour growth analysis from Fig. 5B in immunocompromised NSG mice (with the obvious caveats that needs to be kept in mind that these still have some macrophage functions intact). Furthermore, Fig. 5E-H should include wild-type plus cGAMP so that proper comparisons can be made.

We thank the reviewer for the insightful suggestions. We agree that cGAS-STING signaling in tumor biology is complex, particularly with integrating antitumor immunity from NK/CD8+ T, TAM/CAF-mediated immune evasion, and effects of STING-induced senescence and microenvironmental modulations. Therefore, we have added a detailed discussion on tumor phenotypes observed in the Discussion section.

cGAS-STING signaling induces robust expression of type I interferons and ISGs, which recruits and activates CD8⁺T and NK-mediated antitumor immunity¹. We observed that the acute activation of cGAS-STING signaling by cGAMP changed the intratumoral macrophage residency through STING-TBK1-Zyxin signaling. Interestingly, an M2-to-M1 polarization was also detected (Fig 6G-6H), which increased the secretion of

interferons and pro-inflammatory cytokines and exhibited immunostimulatory functions. As a result, cGAMP addition positively affects the mice with WT Zyxin, as observed in the syngeneic B16-F10 model ^{2,3}.

We propose that this Zyxin-macrophage axis is critical in the immune-suppressive B16-F10 model. Zyxin knockout reduced the residency of tumor-suppressive macrophages, representing an ideal antitumor-endorsing microenvironment and promoting antitumor immunity at a level similar to the efficient recruitment of CD8⁺ T-mediated immunity. As a result, no significant changes will be seen upon adding cGAMP, which is supposed to recruit CD8⁺ T lymphocytes. In agreement with this opinion, we found that the Zyxin KO+anti-PD-1 group exhibits significantly more potent effects than any single condition (Fig 6I, 6J, 6K, EV6K), as anti-PD-1 further acts on the activation and efficacy of T lymphocytes. Following the reviewer's instruction, we acquired and conducted syngeneic B16-F10 models in the NSG mice. We found that little difference was seen in the growth curves and weights of WT and Zyxin KO B16-F10 tumors (Figs 7D, 7E, 7F), also suggesting that Zyxin KO does not affect tumor growth but antitumor immunity, from where CD8⁺ T-mediated antitumor effect was compromised. Expectedly, intratumoral injection of cGAMP only slightly reduced both growth curves and weights of B16-F10 tumors in NSG mice, much less pronounced than in C57BL/6 mice (Figs 6B-6D and Figs 7D-7I). All these consistent observations suggest a model that the STING-TBK1-Zyxin-mediated macrophage infiltration control modulates antitumor immunity to suppress the recruitment of cytotoxic CD8⁺ T cells in the syngeneic B16-F10 model. Besides, we have now supplemented the data of the cGAMP group in Figs 6E-F as recommended by the reviewer.

5. It is not clear if the authors want to make a case for a macrophage-specific effect or not. Some of the cell biology seems to suggest macrophage specificity (for example, the HeLa cells in Fig. 2C do not seem to recapitulate the phenotype the authors describe, although the authors do suggest otherwise), and some of the tumour data on macrophage-depleted mice seem to suggest importance primarily in macrophages, but then some of the phenotypes do seem to be recapitulated in DLD1 and HEK293 cells. This should be clarified, at the very least by altering the wording of the manuscript, and inclusion in the discussion.

Thanks for the instruction. We believe this phenomenon is present in various types of cells but might be more pronounced in macrophages (Figs 2B-2F, EV2A). As a prominent function of the STING-TBK1-Zyxin axis in tumor-associated macrophages was found, we insisted that, at least in TAMs, this non-canonical signaling has a biological relevance. The physiological relevance of the STING-TBK1-Zyxin-regulated cell motility in other types of cells is waiting to be explored. We added this discussion in the revised manuscript and changed the wording accordingly.

6. Following up from this, the macrophage deletion experiment is not entirely unambiguous. First, the authors don't show that their treatment in their specific

experiment actually depleted macrophages (presumably that data exists within the microscopy, and could be resolved by quantification of the images, see also point 1, above).

Second, Fig. 6B suggests that macrophage deletion actually improves the growth of tumours in zyxin^{-/-} mice. It is also unclear why there are effects in the absence of cGAMP treatment in Fig. 6. As before (point 4, above), tumour biology can be very complex, and I don't expect the authors to completely resolve this issue. But at least it should be discussed.

Thanks for the suggestion. We have now supplemented the statistical data of immunofluorescent analyses (Fig EV7B), confirming the substantial reduction of intratumoral macrophages by clodronate liposomes. The effects of TAMs in tumor biology are also complex^{4,5}. In the scenario of Zyxin KO mice, intratumoral TAMs were largely downregulated (Figs. 6E-6H), and the effect of slight tumor growth of macrophage deletion could have resulted from other secondary effects such as cytokine production⁶. Zyxin-mediated macrophage residency control is critical in this immune-suppressive B16-F10 model, and Zyxin knockout could represent an ideal antitumor-endorsing microenvironment and promote antitumor immunity similar to the efficient recruitment of CD8⁺ T-mediated immunity. Additionally, we observed that tumor-associated macrophages in Zyxin knockout mice exhibit a CD86⁺ phenotype, reporting as an M1-proinflammatory phenotype. The administration of Clodronate Liposomes in Zyxin^{-/-} mice might also partially eliminate these antitumor macrophages. Besides, chronic inflammation in the tumor microenvironment, such as chromosomal instability (CIN), activates the cGAS-STING pathway at a lower level (Fig EV6A). Thus, an evident phenotype was observed upon Zyxin deletion, suggesting its critical role in regulating immune responses.

7. The Discussion seems a bit too much to repeat the findings from the results section, and could contextualise the results better with the published literature. For example, how would the authors put their findings of a function of cGAS-STING that potentially limits migratory activity with the published promotion of cell migration due to cGAS-STING (PMID 29342134 and 27225120). Furthermore, the question of potential cell type specificity (see also point 5) should be discussed.

Thanks for the suggestions. We have revised the discussion section with the above discussions and added references such as PMID 29342134: "Chromosomal instability drives metastasis through a cytosolic DNA response", PMID 27225120: "Carcinoma-astrocyte gap junctions promote brain metastasis by cGAMP transfer", and others. The content of the discussion section has been expanded significantly.

Minor points

1. The authors use unscientific adjectives and adverbs such as "impressively", "dramatic", "profound", "critical", "profoundly" etc. too much. Please remove them all.

Thanks for the suggestions. We have reworded the text accordingly.

2. The use of the term "non-canonical function" for the mechanism they have described should be removed from title, abstract and body of the manuscript. It is not defined unambiguously what a canonical function of cGAS is, although usually this term seems to be used to refer to cGAMP-dependent functions. However, the authors findings clearly do not describe functions that are obviously cGAMP-independent.

Accordingly, we revised the title, abstract, and sentences and reworded "non-canonical function" with a more precise description.

3. It is unclear from the text if the authors think that the Zyxin regulation is cGAS pathway specific. On the one hand, they have data indicating that other pathways that converge on TBK1 can also have this effect, but then they argue for an interaction between Zyxin and STING. I suppose this might be indirect as both proteins may bind TBK1? If so, this should be explained in the text.

Thanks for the suggestion. TBK1, a critical stress kinase, is also activated in various cellular stress scenarios. We believe whether Zyxin could participate in the signalosome is the key, such as in the case of STING signalosomes and MAVS signalosomes. We have explained this point and changed the wording in the manuscript accordingly.

4. The data from Fig. 4E-G seem to suggest that Zyxin is only partially responsible for the phenotypes. While the authors to say this, it isn't quite made clear enough.

Thank you for this insightful suggestion. In the manuscript, we have discussed the potential regulation of other cytoskeleton proteins by cGAS-STING signaling. While Zyxin can partially rescue the phenotype, other scaffold and mechanosensing proteins could still play a role.

5. Line 42, "cellular condensation", this is not a standard term, should be removed or changed.

Thanks. This wording has been changed.

6. Typos in lines 237 and 238 (should refer to the supplement).

We appreciate your attention to detail. We have updated the labeling of the relevant figures in the manuscript.

7. Fig. S6 does not seem to be referenced in order.

Many thanks. We have carefully reviewed the order of Figure S6 and corrected the mislabeling in the manuscript and figure captions.

8. Lines 343 and 344: "We also confirmed that combining Zyxin knockout with anti-PD-1 effectively reduces melanoma growth." This should be changed, as the authors did not look at melanoma proper, but at a melanoma-derived cell line grown subcutaneously.

Thanks, and we have revised the text to say, "We also confirmed that combining Zyxin knockout with anti-PD-1 effectively reduces a subcutaneously grown melanoma-derived cell line." This modification indeed adds rigor to the manuscript.

Referee #2:

In this manuscript the authors show that TBK1 - activated downstream of cytosolic DNA or RNA sensing - can modulate the migration/adhesion properties of macrophages. TBK1 promotes adhesion and limits migration in mouse peritoneal macrophages. The data supports a model where TBK1 directly phosphorylates the adhesion factor Zyxin at serine 142, promoting its localisation at focal adhesions.

The link between innate immune sensing and macrophage motility is interesting, and so far relatively under-studied. Most studies on DNA and RNA sensing focus on the TBK1-induced production of type I interferons etc, and so this is an interesting addition, particularly in the context of tumor immunology where STING agonists are promising anti-cancer agents (at least in mice). Thus, the significance and novelty of the findings are quite high.

My main comment is more about the description of the findings - the data clearly shows that this is a novel function of TBK1, not restricted to the cGAS-STING pathway. This should be made clearer in the title, abstract and discussion. The title currently states cGAS-STING-Zyxin as signalling axis (when TBK1-Zyxin is probably more accurate) and in the abstract (line 6) the statement that "cGAS-STING directly phosphorylates and mobilizes Zyxin" is also a little misleading. STING still activates TBK1, just as in canonical cGAS-STING signalling, so this is more an additional downstream function of TBK1 which should be made clearer.

Main issues:

1. Figure 1 nicely shows effects on macrophage adhesion and migration in mouse peritoneal macrophages - can similar experiments be performed with human macrophages?

Thanks for the reviewer's very positive comments about the manuscript and insightful wording instructions. Following the reviewer's suggestion, we have examined the effect of innate DNA/RNA sensing in human macrophages. Human monocyte THP-1 cells were differentiated into macrophages by PMA induction, and their adhesion and migration were analyzed under the effects of SeV (RNA virus), poly(I:C) (RNA analog), HSV-1 (DNA virus), and diABZI (STING agonist). Similar to murine macrophages, diABZI or poly(I:C) promoted THP-1 adhesion to HUVECs, by which TBK1 inhibitor GSK8612 compromised it, and similar but somewhat minor effects of SeV/HSV-1 on macrophage adhesions were seen (Figs EV1A, EV1C), suggesting conservation of TBK1-Zyxin-mediated regulation of macrophage adhesion and migration between mice and humans.

2. For completion, the migration assays in 1B-E could also be shown with poly(I:C).

Per the suggestions, we analyzed the motility of macrophages under the treatment of poly(I:C) transfection, which similarly inhibited the motility of PMs (Figs EV1B, EV4E) and promoted the adhesion of murine PMs on glass slides and HUVEC (Figs EV4H, EV4I).

3. The STING-Zyxin co-localisation data in Fig. 2C is not very clear. Can the diABZI STING agonist be used like poly(I:C) in the TBK1-HA expressing cells (Fig. 3E-F)? Or in other responsive human cells (macrophages)?

Thanks for the suggestion. We have quantified the fluorescence intensity of the data in Fig 2C, and also repeated these experiments on PMs and human cells, which showed the signal overlap between Zyxin and TBK1 in DLD1 cells (Figs 2I, EV2B), followed by an aggregation of Zyxin at focal adhesions.

4. Please quantify cell features in the microscopy images where possible, e.g. macrophage adhesion area in all experiments in Fig 4 etc.

Thanks for the reviewer's suggestion. The statistics of macrophage adhesion area in Fig 4A, 4C, 4E can be found in Fig EV4A-EV4B. In addition, we conducted quantitative fluorescence intensity analyses on the immunofluorescence images, including Figs 2B, 2C, 2E, 2F, EV2A, EV2B, 3K, EV3H and quantified cell proportions of Figs 2F, EV2B, 3E, EV3A, EV3C, 4A, 4C, 4E, 5C,6E, 6G, 7L, 7P, EV7A, EV7D. These quantified graphs can now be found at Figs 2H, 2I, 3E, EV3B, EV3D, 4B, 4D, 4F, 5C, 6F, 6H, 7K, 7O, EV7B, EV7C, EV7E, EV7F, which improved the solidity of the observations.

5. It would be useful to show that Sendai virus or poly(I:C)-induced macrophage adhesion is also dependent on Zyxin (like shown for DMXAA in Fig. 4).

Thanks for the reviewer's instruction. We have examined the adhesion of Zyxin KO macrophages upon the activation of innate RNA sensing by poly(I:C) transfection (Fig EV4C). Noticeably, Zyxin distribution at focal adhesions induced by poly(I:C)

transfection almost disappeared in Zyxin KO macrophages (Figs EV4G, EV4F). Besides, poly(I:C) transfection promoted PM adhesion on both glass slide and HUVEC, by which TBK1 inhibitor GSK8612 or Zyxin deletion reversed it (Figs EV4H, EV4I). These observations suggest the presence of RNA-sensing-induced control of macrophage adhesion.

6. The finding that Zyxin deletion enhances STING-TBK1-IRF3 signaling is important, and might also be involved in the increased inflammatory phenotype in the deletion mice, rather than the effects on macrophage migration/adhesion. Thus, some of this data should go into the main figures. A similar experiment like Fig. S4C could also be carried out for poly(I:C)-induced RNA sensing.

Thanks for the reviewer's suggestion. Per the reviewer's suggestion, we have rearranged the original Fig S4 accordingly. We also examined the effect of poly(I:C)-induced RNA sensing and LPS-induced TLR4 signaling in Zyxin KO cells (Figs EV5C, EV5D), which showed a moderate effect of enhanced pIRF3 (S396) levels in Zyxin-deficient cells activated by cGAMP, diABZI, poly (I:C), or LPS (Figs 5A, 5B, EV5C, EV5D). However, PMs from global IRF3 knockout mice retained the signal of DMXAA-induced pZyxin, indicating that the TBK1-Zyxin axis is independent of the classical TBK1-IRF3 axis (Fig EV2D).

7. Rather than affecting STING signalosomes as proposed, it is also possible that Zyxin competes with IRF3 for the interaction with TBK1 - this could be tested using co-immunoprecipitation with over-expression and deletion cells.

Thanks for the suggestion. We employed the IRF3 2SA mutation, a well-documented IRF3 tool to detect low-affinity TBK1-IRF3 interaction (the endogenous TBK1-IRF3 complex is beyond detection as its subtle kinase-substrate nature), to analyze a possible competition between Zyxin and IRF3 to TBK1-mediated phosphorylation. We found that Zyxin interfered with the TBK1-IRF3 interaction, suggesting the presence of a possible substrate competition between Zyxin and IRF3 (Fig 5D and EV5E), which might explain the somewhat enhanced pIRF3 levels in cells without Zyxin.

8. The observation that Zyxin deletion enhances STING activation confounds the tumor data in the mouse models. Given that STING-TBK1-IRF3 activation has potent antitumor effects in mice, the enhancement of canonical STING signaling could be responsible for the observed effects. A lot of weight rests on the results with IRF3 deletion mice (Fig. S6D-E) where no effect on B16 tumors is seen in the presence or absence of IRF3. However, this is different from many other studies where IRF3-dependent melanoma control has been observed (e.g. Woo et al, Immunity 2014) and so this will be controversial in the field. More substantial data would be needed to disentangle the role of IRF3- and Zyxin-dependent tumor control.

Thanks for the reviewer's insightful suggestion on this interesting point. The classical STING-TBK1-IRF3 axis is critical in antitumor immunity by controlling the production of interferons/chemokines/inflammatory factors and modulating the functionality of CD8⁺ T/NK-mediated cytotoxic effects^{7,8}. However, IRF3-independent functions of STING have been documented by several lines of evidence^{9,10}. As illustrated in Fig EV6A, we observed the activation of pTBK1 and pZyxin in a resting state but did not observe a significant activation of pIRF3. We suggested that Zyxin functions independent of IRF3, supported by two lines of evidence: 1) DMXAA robustly induced pZyxin in IRF3 knockout KO PMs, dependent on TBK1 activity (Fig EV2D). 2) Global Zyxin deletion in IRF3 knockout KO mice still compromised the intratumoral residency of TAMs (Figs 7O, 7P) and the growth of B16-F10 tumors (Figs 7M and 6N), suggesting a critical role of the STING-TBK1-Zyxin axis in regulating antitumor immunity through an IRF3-independent mechanism, at least partially.

We also repeated our syngeneic B16-F10 models in STING KO mice and reviewed those observations compared to the report from Woo et al. (Immunity 2014). Without cGAMP injection, the tumors in STING KO mice were slightly but not significantly larger than those in the WT group, consistent with those observations in IRF3 KO mice. However, in the cGAMP group, B16-F10 tumors in STING KO mice were significantly larger than those in the WT group, as shown in Figs 7G, 7H, 7I, 7M, and 7N. We speculate that these contradictory observations may be related to the differences between B16-F10 and B16.SIY tumor cells, the initial seeding quantity of tumors, and the possible side effects of cGAMP. We recorded these distinct but interesting observations in STING KO mice, which did not go against our conclusions on STING-TBK1-Zyxin control of TAM residency and its potential role in antitumor immunity.

Minor suggestions:

1. Make sure acronyms are defined at first use, e.g. IRF3, NF-kB, TAM etc(p7)

Thanks, and we have followed the suggestion now.

2. Insert a few words introducing various cell types when used (human/mouse, cancer-derived/immortalized/primary), e.g. DLD1 cells (p8)

Thanks, and we have followed the suggestion.

3. The layout of the data in Fig. 2H is unclear. Maybe separate into two graphs (e.g. pZyxin signal in individual cells, pZyxin in FAs separately) for clarity.

Thanks for the suggestion. We have rearranged this figure, making it clearer.

Overall, I think the findings regarding TBK1-mediated control of macrophage adhesion are solid, novel and interesting, and cross-regulation with TBK1-IRF3 activation should be examined a little further. I am less convinced about the role of Zyxin in tumor

control, given the observed enhancement in IRF3 signaling (which is interesting in itself). Some of the in vivo data (including the HSV-1 eye infection and the MC38 tumors are insufficiently controlled for potential enhanced STING-TBK1-IRF3 effects, and thus might need deeper investigation - potentially in a separate manuscript.

Thanks for the reviewer's insightful and positive feedback. On the one hand, we have strengthened our opinion of Zyxin-mediated modulation of antitumor immunity by utilizing IRF3 KO mice. However, we also acknowledge that the roles of Zyxin in inflammatory regulation might be more complicated and could go beyond macrophage movement control. According to the reviewer's instructions, we have not included HSV-1 corneal infection data.

Referee #3:

Here the authors describe a STING-TBK1-dependent role of Zyxin in mediating macrophage tumour residency and migration. Overall the paper describes a novel pathway and phenotype of interest. I have some major and minor comments as below.

Figure 1A - could the choices for different time points be explained in the text? It is unclear why some conditions are imaged at 9 hours and some at 1 hour post treatment. Inclusion of an interferon control would be important here as this seems to be a broad interferon response given the multiple DNA/RNA sensors which result in the same phenotype. Common feature of both pathways is TBK1.

Thanks very much for the reviewer's positive and constructive suggestions. We have examined macrophage adhesion at a distinct time frame, mainly as the dynamics of cellular responses to innate immune stimulations, such as infection of SeV and HSV-1, poly(I:C) transfection, or treatment of STING agonist, are significantly different. For example, DMXAA quickly activated STING signaling and exhibited a decline of pTBK1 and pIRF3 levels upon 1 hour (Figs 2J, 2K, 2L, EV2C), while a slow but sustained TBK1/IRF3 activation was seen during RNA/DNA viruses infection. Per the reviewer's suggestion, a description of the time point selection was added.

To check the involvement of interferons in macrophage adhesion, we analyzed IRF3 KO cells, which had a deficiency in interferon production¹¹, but showed somewhat regular levels of Zyxin phosphorylation upon STING signaling. This observation suggests a potential role of the TBK1-Zyxin axis independent of IRF3 and interferon. On the other hand, we found that mIFN α and mIFN β can indeed induce an increase in Zyxin mRNA and protein levels (Figs EV2G-EV2H) and increased PM adhesion area (Fig EV2E). Therefore, we proposed that interferons, produced by the classic TBK-IRF3 signaling, could aid in regulating macrophage adhesion, possibly through the transcriptional regulation of Zyxin.

Figure 5 - Clarify in the text that when referencing resistance to anti-PD-1 in melanoma the authors are specifically talking about the B16F10 model, not generally. Zyxin knockout results in resistance to cGAMP - attributing anti-tumour immune responses to a direct impact of the cGAS-STING-TBK-Zyxin axis. However, in contrast - Zyxin knockout sensitises to anti-PD-1. The response here is attributed to a reduction in TAMs overall mediated by anti-PD-1 - I would have thought this should be a T cell mediated response? Is it the case that a T-cell activation, mediated by anti-PD-1, is independent of Zyxin but the cGAMP response requires Zyxin-pathway activation in TAMs? I am not convinced by the authors explanation of these data.

Thanks for these insightful suggestions. We have clarified that the resistance to anti-PD-1 in melanoma is specific to this model, not a general phenomenon. cGAS-STING signaling induces robust expression of type I interferons and ISGs, which recruits and activates CD8⁺ T-mediated antitumor immunity¹. We propose that this Zyxin-macrophage axis is critical in the immune-suppressive B16-F10 model. Zyxin knockout reduces the residency of tumor-suppressive macrophages, representing an ideal CD8⁺ T recruiting and antitumor-endorsing microenvironment. As a result, no significant changes will be seen upon adding cGAMP, which is also supposed to recruit CD8⁺ T lymphocytes. In agreement with this opinion, we found that the Zyxin KO+anti-PD-1 group exhibits significantly more potent effects than any single condition (Fig 6I, 6J, 6K, EV6K) as anti-PD-1 further acts on the activation and efficacy of T lymphocytes, and a decreased proportion of TAMs and increased infiltration of CD8⁺ T lymphocytes was expectedly seen in the combination of anti-PD-1 with Zyxin deficiency. Besides, we have conducted the syngeneic B16-F10 model in the NSG mice and found little difference was seen in the growth curves and weights of WT and Zyxin KO B16-F10 tumors (Figs 7D, 7E, 7F). These observations suggest that Zyxin KO relieves TAM-mediated immune suppression, from which the CD8⁺ T-mediated antitumor effect was compromised.

Figure 6 - Treating Zyxin knockout with clodronate liposomes actually increases tumour growth - the stats here are not included, however there is clear separation in the curves. This is a slightly confusing result and the authors do not comment on it. If the PD-1 response (Figure 5) is mediated by TAMs, why is this not reflected in these data?

Thanks for the suggestion. We have provided additional explanations regarding clodronate liposomes and statistical results (Figs S7B, S7C). Clodronate liposomes have a complex regulatory effect on the tumor microenvironment^{12,13}. In the scenario of Zyxin KO mice, intratumoral TAMs were largely downregulated (Figs. 6E-6H), and the effect of slight tumor growth of macrophage deletion could have resulted from other secondary effects such as cytokine production⁶. Additionally, we observed that tumor-associated macrophages in Zyxin knockout mice exhibit a CD86⁺ phenotype, reporting as an M1-proinflammatory phenotype. The administration of Clodronate Liposomes in Zyxin^{-/-} mice might also partially eliminate these antitumor macrophages, and we have supplemented data in Figs EV7A-7C. We agree with your insightful point that the

effectiveness of PD-1 and Zyxin KO is achieved through the reduction of TAMs and an increase in the proportion of CD8⁺ T cells (Fig. EV6K).

The FACS analysis of the tumours is quite broad - F4/80 is a broad macrophage marker and the potential repolarisation of the macrophages is not considered, rather that absolute numbers are quantified. This should be included as a limitation (that could address some of the confusing results above). The role of Zyxin in macrophage polarisation would be of interest, although is unfortunately not included in these data.

Thanks for the reviewer's instruction. We added a limitation statement in the text and examined macrophage polarization preliminarily by using markers such as CD206 and CD86 (see Figs 6G, 6H). These observations showed that most intratumoral macrophages exhibit M2-type markers in the resting state, while Zyxin KO reduced the proportion of CD206-positive cells but somewhat increased CD86-positive cells (M1-type marker). These observations have been included in the revised manuscript and discussed. It appeared that Zyxin deletion could decrease the mRNA level of Arg1, one of the key regulators of M2 macrophage physiology; however, we did not include this data as it was somewhat irrelevant to the current topic and preliminary.

Too many references are included in a short piece

Thanks, and we have appropriately reduced the number of references in the manuscript.

In Methods -

Following ethical permission, the maximum diameter of the tumors is limited to 15 mm in one dimension, by which the mice were sacrificed and defined as dead due to tumor burden. - please rephrase as "Mice were killed before tumors reached 15 mm in any one dimension in accordance with local ethical guidelines."

Thanks, and we have accordingly rephrased.

In Introduction

Signalingly - this is not a recognised expression. Please rephrase.

Thanks, and we have reworded.

Figure 2H should be revised - the use of a striped section of bar is confusing.

Thanks for the suggestion. We have generated a figure depicting the pZyxin signal in individual cells in Fig 2H.

Throughout - there are many instances in the text where the authors' meaning is not clear. Additional editing would be overall helpful to improve this manuscript. This is a particular issue in methods, although occurs throughout the manuscript.

Thanks for your suggestion. We have now carefully revised the sentences and descriptions, and we believe this revision can help us to convey our scientific conclusions to readers better.

- 1 Wu, S. *et al.* HER2 recruits AKT1 to disrupt STING signalling and suppress antiviral defence and antitumour immunity. *Nat Cell Biol* **21**, 1027-1040 (2019). <https://doi.org/10.1038/s41556-019-0352-z>
- 2 Wang, H. *et al.* cGAS is essential for the antitumor effect of immune checkpoint blockade. *P Natl Acad Sci USA* **114**, 1637-1642 (2017). <https://doi.org/10.1073/pnas.1621363114>
- 3 Uslu, U. *et al.* The STING agonist IMSA101 enhances chimeric antigen receptor T cell function by inducing IL-18 secretion. *Nat Commun* **15**, 3933 (2024). <https://doi.org/10.1038/s41467-024-47692-9>
- 4 Colegio, O. R. *et al.* Functional polarization of tumour-associated macrophages by tumour-derived lactic acid. *Nature* **513**, 559-563 (2014). <https://doi.org/10.1038/nature13490>
- 5 Pittet, M. J., Michielin, O. & Migliorini, D. Clinical relevance of tumour-associated macrophages. *Nat Rev Clin Oncol* **19**, 402-421 (2022). <https://doi.org/10.1038/s41571-022-00620-6>
- 6 Pan, Y., Yu, Y., Wang, X. & Zhang, T. Tumor-Associated Macrophages in Tumor Immunity. *Front Immunol* **11**, 583084 (2020). <https://doi.org/10.3389/fimmu.2020.583084>
- 7 Deng, L. *et al.* STING-Dependent Cytosolic DNA Sensing Promotes Radiation-Induced Type I Interferon-Dependent Antitumor Immunity in Immunogenic Tumors. *Immunity* **41**, 843-852 (2014). <https://doi.org/10.1016/j.immuni.2014.10.019>
- 8 Kwon, J. & Bakhoun, S. F. The Cytosolic DNA-Sensing cGAS-STING Pathway in Cancer. *Cancer Discov* **10**, 26-39 (2020). <https://doi.org/10.1158/2159-8290.CD-19-0761>
- 9 Gui, X. *et al.* Autophagy induction via STING trafficking is a primordial function of the cGAS pathway. *Nature* **567**, 262-266 (2019). <https://doi.org/10.1038/s41586-019-1006-9>
- 10 Chu, T. T. *et al.* Tonic prime-boost of STING signalling mediates Niemann-Pick disease type C. *Nature* **596**, 570-575 (2021). <https://doi.org/10.1038/s41586-021-03762-2>
- 11 Roers, A., Hiller, B. & Hornung, V. Recognition of Endogenous Nucleic Acids by the Innate Immune System. *Immunity* **44**, 739-754 (2016). <https://doi.org/10.1016/j.immuni.2016.04.002>
- 12 Zeisberger, S. M. *et al.* Clodronate-liposome-mediated deletion of tumour-associated macrophages: a new and highly effective antiangiogenic therapy approach. *Br J Cancer* **95**, 272-281 (2006). <https://doi.org/10.1038/sj.bjc.6603240>
- 13 Wu, L. *et al.* Defining and targeting tumor-associated macrophages in malignant mesothelioma. *Proc Natl Acad Sci U S A* **120**, e2210836120 (2023). <https://doi.org/10.1073/pnas.2210836120>

Dear Dr. Xu,

Congratulations on a great revision! Overall, the referees have been positive. However two referees have a few more suggestions that we ask you to (non-experimentally if you choose) address in a new revision, including to please have your manuscript thoroughly edited. When you submit your revised version, please also take care of the following editorial items, and add this also to your point response:

1. Please add the following funding information into eJP: NSFC Projects (31830052, 31725017, 82201920, and 81902915).
2. Please reduce the number of keywords to 5.
3. Please update the format of references to be alphabetical and list only the first ten authors followed by et al.
4. Please remove the author contribution section from the main manuscript.
5. Please review our new policy on conflict of interests on the EMBO author guide website and update the title of this section to: Disclosure and competing interests statement.
6. In the author checklist, please change the general info table from "Journal Submitted to: Kelly M Anderson, PhD" should be "Journal Submitted to: The EMBO Journal"
7. We include a synopsis of the paper (see <http://emboj.embopress.org/>). Please provide me with a general summary statement and 3-5 bullet points that capture the key findings of the paper.
8. We also need a summary figure for the synopsis. The size should be 550 wide by 200-440 high (pixels). You can also use something from the figures if that is easier.
9. For the appendix file, please include a title page with title and table of contents with page numbers.
10. Please include source data for Figure 4G.
11. Please note that the accession ID for the BioStudies database should be provided in the data availability statement.
12. Please define the annotated p values ***/**/* as well as provide the exact p-values for the same in the legend of figure EV 6f as appropriate.
13. Please note that the exact p values are not provided in the legends of figures 1a-e; 4b, d, f, j; 6h; EV 1a-b; EV 2e-g; EV 3d; EV 4c-e, f-g; EV 6c, k; EV 7b-c, e-f.
14. Please indicate the statistical test used for data analysis in the legends of figures 3e; 4b, d, f, h, j; 5c; 6b-c, f, h, j-k; 7b-c, e-f, h-k, n-o; EV 1a-c; EV 2e-g; EV 3b, d; EV 4a-e, g, i; EV 6c-d, f-g, j-k; EV 7b-c, e-f.
15. Please note that information related to n is missing in the legends of figures 2g-i; 3e; 4b, d, f; 6b-c, f, h, j-k; 7b-c, e-f, h-k, n-o; EV 1a, c; EV 2e-g; EV 3b, d; EV 4a-c, e, g, i; EV 6d, f-g, j-k; EV 7b-c, e-f.
16. Although 'n' is provided, please describe the nature of entity for 'n' in the legends of figures 4h, j; EV 1b.
17. Please note that the error bars are not defined in the legends of figures 2g-i; 3e; 4b, d, f, h, j; 6b-c, f, h, j-k; 7b-c, e-f, h-k, n-o; EV 1a-c; EV 2e-g; EV 3b, d; EV 4a-e, g, i; EV 6c-d, f-g, j-k; EV 7b-c, e-f."
18. Please note that the yellow arrows are not defined in the legend of figure 3k; 4a; 7l, p; EV 3h; EV 4f; EV 7a. This needs to be rectified.
19. title page with complete author information, abstract, keywords, introduction, results, discussion, materials & methods, data availability section, acknowledgements, disclosure and competing interests statement, references, main figure legends, tables, expanded figure legends.
20. The following author email bounced back to us, please provide an updated email address: li_xiao@zju.edu.cn

Thank you for the opportunity to consider your work for publication, I look forward to your revision!

Warm regards,
Kelly

Kelly M Anderson, PhD
Editor, The EMBO Journal
k.anderson@embojournal.org

Referee #1:

In their manuscript entitled "Mechanical control of tumor-associated macrophage residency by non-canonical cGAS-STING-Zyxin signaling", Zhou, Wang, Li et al. present evidence for a function of TBK1 in regulating cellular motility through phosphorylation of Zyxin, and a potential resulting function in suppressing anti-tumour immune responses. Overall, the findings are interesting and potentially represent an important advance in our understanding of the consequences of TBK1-mediated signalling. Once the remaining points have been addressed, I am in favour of publication.

1. The microscopy has not fully been associated with quantifications. All the signal overlap analyses need quantifications: Fig. 2B, C (Fig. 2D doesn't indicate how often the experiment was repeated); Fig. 2E, F; Fig. 3K; Fig. S2A, B; Fig. S3H.
2. The relative impacts of cGAMP addition and Zyxin knockout (Fig. 6B) need to be explained better in the text. In their tissue culture experiments, they clearly need STING agonism or RNA sensing to drive Zyxin relocalisation (Fig. 2I). Yet in their tumour model it is not clear where cGAMP is coming from, especially since cGAMP addition has an effect that suppresses tumour growth rather than promoting it (as one might think, given the authors model that cGAMP drives oncogenic TAMs into tumours), and confusingly, there is no additive effect between cGAMP addition and Zyxin KO. For example, this might mean (and actually quite likely, given its described impacts on cell biology) that Zyxin has cGAMP-independent roles. It is fine to have somewhat confusing data, but all this needs to be acknowledged and discussed in a better way.
3. The confusing fact that Fig. 7B shows that depletion of macrophages improves the growth of tumours in zyxin KO mice is still not properly discussed. Again, it is fine to have somewhat confusing data in a biological study, but this needs to be acknowledged and discussed.
4. The authors still use a number of unscientific adverbs and adjectives such as "impressively" and "dramatic". Please remove them all.
5. Figure panels are arranged in a confusing manner and are referenced in a somewhat chaotic manner. Please streamline this. The manuscript contains a huge amount of data, and the authors need to help readers by being clearer and more systematic in their descriptions of the data.
6. Line 54/55: The authors need to make clear that they talk about mouse experiments. To my knowledge not human study has yet shown positive effects of STING agonism.
7. Line 210, "implying" is too strong a word and should be replaced with 'suggesting'.
8. Lines 292-295: From the text it seems like Fig. S6H-J should show Zyxin KO but it doesn't, which is confusing, as the reader needs to work out what the authors mean with this sentence. Please make this clearer.

Referee #2:

The authors have revised the manuscript to convincingly show a novel function of TBK1 in Zyxin-mediated cell motility. The authors have provided additional experimental data, quantifications and controls to support their findings and have addressed my main concerns.

The manuscript would benefit from minor text edits (for spelling, grammar and clarity on occasion).

Referee #3:

The authors have added a number of experiments and revised their manuscript in line with the reviewers comments. The quality of the figures and text has significantly improved. The authors provide a point-by-point response. The detailed response and

improved manuscript are appreciated.

For the reviewers' comments:

Referee #1:

In their manuscript entitled "Mechanical control of tumor-associated macrophage residency by non-canonical cGAS-STING-Zyxin signaling", Zhou, Wang, Li et al. present evidence for a function of TBK1 in regulating cellular motility through phosphorylation of Zyxin, and a potential resulting function in suppressing anti-tumour immune responses. Overall, the findings are interesting and potentially represent an important advance in our understanding of the consequences of TBK1-mediated signalling. Once the remaining points have been addressed, I am in favour of publication.

1. The microscopy has not fully been associated with quantifications. All the signal overlap analyses need quantifications: Fig. 2B, C (Fig. 2D doesn't indicate how often the experiment was repeated); Fig. 2E, F; Fig. 3K; Fig. S2A, B; Fig. S3H.

We appreciate your instructions, which have guided us in substantially improving the manuscript. Signal overlap analyses have now been quantified, including Fig. 2B, 2C, Fig. 2E, F; Fig. 3K; Fig. EV2B, C; Fig. EV3G, and the missing information on experimental repetitions of all Figure 2 panels has been complemented. These new quantification analyses are presented in Figs 2E-F, EV2B-C, and EV 3F, 3H upon the rearrangement of figure panels.

2. The relative impacts of cGAMP addition and Zyxin knockout (Fig. 6B) need to be explained better in the text. In their tissue culture experiments, they clearly need STING agonism or RNA sensing to drive Zyxin relocalisation (Fig. 2I). Yet in their tumour model it is not clear where cGAMP is coming from, especially since cGAMP addition has an effect that suppresses tumour growth rather than promoting it (as one might think, given the authors model that cGAMP drives oncogenic TAMs into tumours), and confusingly, there is no additive effect between cGAMP addition and Zyxin KO. For example, this might mean (and actually quite likely, given its described impacts on cell biology) that Zyxin has cGAMP-independent roles. It is fine to have

somewhat confusing data, but all this needs to be acknowledged and discussed in a better way.

We thank you for clarifying this intricate point. The roles of cGAS-STING signaling in tumors are currently somewhat controversial¹. Rapid activation of cGAS-STING signaling using agonists like cGAMP or DMXAA generally robustly inhibits tumor growth^{2,3}. However, tumor-derived cGAMP and chronic inflammation caused by DNA-damage response (DDR), chromosomal instability (CIN), mtDNA release, and dying cells may promote tumor progression^{4,5}. These discrepancies could be due to different target cells and downstream pathways of cGAS-STING. In our study, we suggest that regulating the residency of tumor-associated macrophages (TAMs) is likely to play a crucial role in antitumor immunity. We propose that cGAMP-induced STING-TBK1-Zyxin signaling maintains TAM residency within tumors that suppress antitumor immunity, while the induction of robust IFN responses by STING-TBK1-IRF3 signaling promotes CD8+ and NK functionality and TAM pro-antitumoral polarization. Though these contradictory effects look paradoxical, they fine-tune the complex interactions of tumors and TME. Lacking additive effects upon cGAMP addition on Zyxin KO mice somehow indicates the importance of TAM residency control in antitumor immunity, which has obtained a state that could barely further increase. However, we agree with the reviewer's opinion that Zyxin may have other effects on a variety of immune and tumor cells. We have incorporated these explanations into the Discussion section of the manuscript.

3. The confusing fact that Fig. 7B shows that depletion of macrophages improves the growth of tumours in Zyxin KO mice is still not properly discussed. Again, it is fine to have somewhat confusing data in a biological study, but this needs to be acknowledged and discussed.

Levels of intratumoral TAMs in Zyxin KO mice were largely downregulated. As the reviewer referred, we have observed that macrophage deletion in these mice somewhat mildly improved the growth of tumors, although the difference was not significant ($P=0.47$) (Fig. 7B-C). We interpreted this observation in more detail in this updated revision. Clodronate liposomes have a complex and context-dependent regulatory effect on the tumor microenvironment^{6,7}. We observed that remaining tumor-associated macrophages in Zyxin KO mice exhibited CD86⁺, i.e., an M1-proinflammatory phenotype (Fig. 6G-H). Therefore, eliminating these antitumor macrophages by clodronate liposomes in Zyxin^{-/-} mice might contribute to the downregulation of antitumor immunity. We have included these interpretations in the Result section.

4. The authors still use a number of unscientific adverbs and adjectives such as "impressively" and "dramatic". Please remove them all.

Thanks for the suggestions. We have removed all these adverbs.

5. Figure panels are arranged in a confusing manner and are referenced in a somewhat chaotic manner. Please streamline this. The manuscript contains a huge amount of data, and the authors need to help readers by being clearer and more systematic in their descriptions of the data.

Many thanks for the suggestion. We have attempted to streamline the panels logically by rearranging current panels and placing quantification panels into the supplementary information, which clarified and improved the conveyance of these findings. The rearranged images are Figure 2, 7, and Figures EV2, EV3, EV6.

6. Line 54/55: The authors need to make clear that they talk about mouse experiments. To my knowledge not human study has yet shown positive effects of STING agonism.

Thanks, and we have reworded it accordingly.

7. Line 210, "implying" is too strong a word and should be replaced with 'suggesting'.

Thanks, and we have reworded it.

8. Lines 292-295: From the text it seems like Fig. S6H-J should show Zyxin KO but it doesn't, which is confusing, as the reader needs to work out what the authors mean with this sentence. Please make this clearer.

Thanks for the clarification. We have added a description in EV 6I: "STING KO mice show a decreased proportion of CD11b⁺pZyxin⁺ cells".

Referee #2:

The authors have revised the manuscript to convincingly show a novel function of TBK1 in Zyxin-mediated cell motility. The authors have provided additional experimental data, quantifications and controls to support their findings and have addressed my main concerns.

The manuscript would benefit from minor text edits (for spelling, grammar and clarity on occasion).

We appreciate your instructions and suggestions, which have guided us in improving the manuscript. We carefully edited the text in this revision to enhance its precision and readability.

Referee #3:

The authors have added a number of experiments and revised their manuscript in line with the reviewers comments. The quality of the figures and text has

significantly improved. The authors provide a point-by-point response. The detailed response and improved manuscript are appreciated.

We appreciate your instructions and suggestions, which have guided us in improving the manuscript.

- 1 Samson, N. & Ablasser, A. The cGAS-STING pathway and cancer. *Nat Cancer* **3**, 1452-1463 (2022). <https://doi.org/10.1038/s43018-022-00468-w>
- 2 Demaria, O. *et al.* STING activation of tumor endothelial cells initiates spontaneous and therapeutic antitumor immunity. *Proc Natl Acad Sci U S A* **112**, 15408-15413 (2015). <https://doi.org/10.1073/pnas.1512832112>
- 3 Marcus, A. *et al.* Tumor-Derived cGAMP Triggers a STING-Mediated Interferon Response in Non-tumor Cells to Activate the NK Cell Response. *Immunity* **49**, 754-763 e754 (2018). <https://doi.org/10.1016/j.immuni.2018.09.016>
- 4 Bakhoun, S. F. *et al.* Chromosomal instability drives metastasis through a cytosolic DNA response. *Nature* **553**, 467-+ (2018). <https://doi.org/10.1038/nature25432>
- 5 Hou, Y. *et al.* Non-canonical NF-kappaB Antagonizes STING Sensor-Mediated DNA Sensing in Radiotherapy. *Immunity* **49**, 490-503 e494 (2018). <https://doi.org/10.1016/j.immuni.2018.07.008>
- 6 Zeisberger, S. M. *et al.* Clodronate-liposome-mediated depletion of tumour-associated macrophages: a new and highly effective antiangiogenic therapy approach. *Br J Cancer* **95**, 272-281 (2006). <https://doi.org/10.1038/sj.bjc.6603240>
- 7 Wu, L. *et al.* Defining and targeting tumor-associated macrophages in malignant mesothelioma. *Proc Natl Acad Sci U S A* **120**, e2210836120 (2023). <https://doi.org/10.1073/pnas.2210836120>

Dear Dr. Xu,

Congratulations on an excellent manuscript, I am pleased to inform you that your manuscript has been accepted for publication in The EMBO Journal. Thank you for your comprehensive response to the referee concerns and for providing detailed source data. It has been a pleasure to work with you to get this to the acceptance stage.

I will begin the final checks on your manuscript before submitting to the publisher next week. Once at the publisher, it will take about 3 weeks for your manuscript to be published online. As a reminder, the entire review process, including referee concerns and your point-by-point response, will be available to readers.

I will be in touch throughout the final editorial process until publication. In the meantime, I hope you find time to celebrate!

Warm wishes,
Kelly

Kelly M Anderson, PhD
Editor, The EMBO Journal
k.anderson@embojournal.org
